# *Fusobacterium nucleatum* interacts with cancer-associated fibroblasts to promote colorectal cancer

Jessica Karta[1,10], Marianne Meyers [1,10], Fabien Rodriguez[1], Eric Koncina[1], Cedric Gilson [1], Eliane Klein [1], Monica Gabola [1], Mohaned Benzarti [1], Pau Pérez Escriva [1], Jose Alberto Molina Tijeras [1], Catarina Correia Tavares Bernardino [1], Falk Ponath[2], Anais Carpentier[3], Mònica Aguilera Pujabet[4], Maryse Schmoetten[1], Mina Tsenkova[1], Perla Saoud [1], Anthoula Gaigneaux[1], Dominik Ternes[1], Lidia Alonso[4], Nikolaus Zügel[5], Eric Willemssen[6], Philippe Koppes[6], Daniel Léonard [6], Luis Perez Casanova[3], Serge Haan[1], Michel Mittelbronn [1,3], Johannes Meiser [7], Vitaly I Pozdeev [1], Jörg Vogel [2,8], Paolo G Nuciforo[4], Paul Wilmes[1,9] & Elisabeth Letellier [1✉]

## Abstract

**Gut microbial species contribute to colorectal cancer (CRC) by interacting with tumor or immune cells, however if CRC-associated bacteria engage with stromal components of the tumor microenvironment remains unclear. Here, we report interaction between the CRC-associated bacterium *Fusobacterium nucleatum* and cancer-associated fibroblasts (CAFs), and show that *F. nucleatum* is present in the stromal compartment in murine CRC models in vivo and can attach to and invade CAFs. *F. nucleatum*-exposed CAFs exhibit a pronounced inflammatory-CAF (iCAF) phenotype, marked by elevated expression of established iCAF markers, secretion of pro-inflammatory cytokines such as CXCL1, IL-6 and IL-8, generation of reactive oxygen species (ROS), and an increased metabolic activity. In co-culture experiments, the interaction of cancer cells with *F. nucleatum*-stimulated CAFs enhances invasion, a finding further validated in vivo. Altogether, our results point to a role for the tumor microbiome in CRC progression by remodeling the tumor microenvironment through its influence on cancer-associated fibroblasts, suggesting novel therapeutic strategies for targeting CRC.**

**Keywords** Cancer-associated Fibroblasts (CAFs); Colorectal Cancer; *Fusobacterium nucleatum*; Inflammatory CAF (iCAF); Invasion
**Subject Categories** Cancer; Immunology; Microbiology, Virology & Host Pathogen Interaction

## Introduction

Colorectal cancer (CRC) is the third most common cancer type with the second highest rate of cancer-related deaths worldwide (Lichtenstern et al, 2020; Xi and Xu, 2021). Cancer-associated fibroblasts (CAFs) are the most dominant stromal cell type within the tumor microenvironment (TME) and have been associated with worse clinical outcomes and chemotherapy failure in many cancers, including CRC (Tsujino et al, 2007; Chen et al, 2021). CAFs are involved in many oncogenic processes recognized as cancer hallmarks, including the support of tumor cells for evasion from immunosurveillance (Zhang et al, 2023), metabolic rewiring (Karta et al, 2021), sustained proliferation, and metastasis (Norton et al, 2020; Joshi et al, 2021). CAFs are heterogenous and can generally be divided into myofibroblastic CAFs (myCAFs) and inflammatory CAFs (iCAFs), characterized by the increased expression of collagens and inflammatory genes, respectively (Joshi et al, 2021; Khaliq et al, 2022; Elyada et al, 2019; Öhlund et al, 2017; Jenkins et al, 2025; Li et al, 2024). The secretome of CAFs, including cytokines, metabolites, and growth factors, has been widely reported to support tumorigenesis. CAF-derived cytokines such as IL-6 and IL-8, have been shown to suppress anti-tumor natural killer (NK) cell functions in CRC (Zhang et al, 2019) and promote metastasis (Zhong et al, 2021). Colorectal CAFs have also been described to kill CD8[+] T cells in an antigen-dependent manner via PD-L2 and FASL (Lakins et al, 2018; Takahashi et al, 2015). In addition, CAF-derived pro-invasive or pro-angiogenic factors, including CXCL12 (Kojima et al, 2010), VEGF or EGF, directly support tumor growth (Mishra et al, 2010). CAFs are also the main producers of collagens (Kharaishvili et al, 2014; Erdogan and Webb, 2017) and laminins (Fullár et al, 2015), which enhance extracellular matrix (ECM) stiffness and contribute to desmoplastic reactions.

[1]Department of Life Sciences and Medicine (DLSM), Faculty of Science, Technology and Medicine, University of Luxembourg, Esch-sur-Alzette, Luxembourg. [2]Institute of Molecular Infection Biology, University of Würzburg, Würzburg, Germany. [3]Laboratoire National de Santé (LNS), National Center of Pathology (NCP), Dudelange, Luxembourg. [4]Molecular Oncology Group, Vall d'Hebron Institute of Oncology, Barcelona, Spain. [5]Centre Hospitalier Emile Mayrisch, Esch-sur-Alzette, Luxembourg. [6]Groupe Chirurgical Zitha, Hôpitaux Robert Schuman, Luxembourg, Luxembourg. [7]Department of Cancer Research (DOCR), Luxembourg Institute of Health, Luxembourg, Luxembourg. [8]Helmholtz Institute for RNA-based Infection Research (HIRI), Helmholtz Centre for Infection Research (HZI), Würzburg D-97080, Germany. [9]Luxembourg Centre for Systems Biomedicine, University of Luxembourg, Esch-sur-Alzette, Luxembourg. [10]These authors contributed equally: Jessica Karta, Marianne Meyers. ✉E-mail: elisabeth.letellier@uni.lu

*Fusobacterium nucleatum* (*Fn*), a Gram-negative anaerobe often involved in the pathogenesis of periodontal disease (Park and Eslick, 2019), is the most reported and best-studied CRC-associated bacterium (Ternes et al, 2020). An overabundance of *Fn* is often observed in CRC patients (Kostic et al, 2012) and generally associated with poor prognosis and a higher mortality rate (Tahara et al, 2014; Mima et al, 2016). Mechanistically, expression of the adhesion protein FadA on the surface of *Fn* has been shown to bind to E-cadherin on host epithelial cells (Rubinstein et al, 2013). This interaction leads to oncogenic and inflammatory responses via activation of the WNT/β-catenin signaling pathway (Rubinstein et al, 2013). Likewise, the fusobacterial Fap2 autotransporter protein can also bind to a host polysaccharide D-galactose-β-(1-3)-N-acetyl-D-galactosamine (Gal-GalNAc), which is highly expressed in human CRC and metastatic tissues (Abed et al, 2016). Casasanta et al reported that Fap2-mediated binding of *Fn* on tumor cells induced the secretion of IL-8 and CXCL1, which triggered tumor cell migration in vitro (Casasanta et al, 2020). In addition, *Fn* has been reported to suppress the host's immune response when binding via Fap2 to the TIGIT and CEACAM1 inhibitory receptors of NK and T cells in the TME (Gur et al, 2015, 2019) or lead to the recruitment of myeloid-derived suppressor cells (MDSCs) (Liang et al, 2022; Hayashi et al, 2023). While interactions between *Fn* and epithelial or immune cells have been explored, elucidating its engagement with the stromal compartment, particularly CAFs, is critical to fully understanding *Fn*'s impact on the TME and its potential as a therapeutic target.

Previous studies on the cross-talk between *Fn* and fibroblasts have been limited to non-cancer contexts. *Fn* was shown to induce expression of inflammatory and apoptosis-related genes in healthy human gingival fibroblasts, via the activation of Toll-like receptor (TLR)-dependent NF-κB/MAPK/AKT pathways (Kang et al, 2019). *Fn* enhanced the attachment of another pathogenic bacterium, *Porphyromonas gingivalis*, to oral human fibroblasts (Metzger et al, 2009). However, how *Fn* and CAFs interact within the TME and how this cross-talk impacts tumorigenesis in CRC presently remains unclear. In this study, we demonstrate that *Fn* contributes to the formation of a pro-invasive TME. We show that *Fn* is present in the tumor stroma and can attach to and invade CAFs. Furthermore, *Fn*-exposed CAFs present a strong iCAF phenotype, highlighted by the increased secretion of CXCL1, IL-6 and IL-8 as well as reactive oxygen species. Thereby, *Fn*-exposed CAFs contribute to driving cancer cell migration and invasion. Altogether, this study suggests a multifaceted role for *Fn* in modulating the TME, ultimately enhancing cancer cell invasiveness and revealing a previously unrecognized function of *Fn*, particularly through its interactions with CAFs.

# Results

## *Fusobacterium nucleatum* is present in the stromal compartment of CRC, where it can bind to and invade CAFs

The TME plays a significant role in cancer progression. However, the impact of the tumor microbiome within the stromal compartment remains far less explored compared to its interactions with the epithelial or immune cell compartments. We leveraged the CSI-Microbes analysis of the Pelka2021 scRNA-Seq dataset (Robinson et al, 2024; Pelka et al, 2021), to investigate the presence of *Fusobacterium* across different cellular types of the CRC TME. Strikingly, the data show a preferential presence of *Fusobacterium* in the stromal/fibroblast compartment (Fig. 1A). We subsequently investigated the presence of *Fn* in human CRC tissue via in situ hybridization, alongside immunostaining for the stromal markers αSMA and podoplanin (PDPN). We observed areas where *Fn* resides in close proximity to αSMA and PDPN-positive cells, demonstrating that *Fn* is present in vivo within stromal-rich areas (Fig. 1B,C; Appendix Fig. S1A). Next, we performed an in vivo experiment where tumorigenesis was induced by tamoxifen in germ-free *CDX2-CreER^{T2}Apc^{fl/fl}* mice, followed by oral gavage with *Fn* for a span of 3 weeks (Fig. 1D; Appendix Fig. S1B for validation of *Fn* colonization in stool samples, and Appendix Fig. S1C for a PBS-gavaged control mouse). Using an anti-OMP *Fn* reactive antibody, and co-staining mice colons with the stromal marker PDPN, we were again able to localize *Fn* within the stromal compartment (Fig. 1E). While CAF-specific markers remain limited, with many lacking strong specificity (Nurmik et al, 2020), our data indicates that *Fn* is present in vivo in the stromal compartment of tumors, and suggests that bacteria may influence and remodel the stroma, including CAFs.

While the binding of different CRC-associated bacteria, including *Fn*, to epithelial cells is well-documented, less is known about their interactions with other cells of the TME, in particular CAFs. To explore this further, we performed a binding assay directly on human CRC tissue. Using labeled *Fn*, we observed significant binding of *Fn* to CRC tissue, including stromal cell-rich regions. In contrast, the control bacterium *Escherichia coli* MG1655 (*Ec*), exhibited only minimal binding (Fig. 1F), highlighting the selective binding specificity of *Fn* and suggesting that only certain bacteria, including *Fn*, are capable of localizing within the TME. Next, we assessed the invasion capacity of *Fn* to CAFs, by co-culturing CAFs with *Fn* in vitro and observed, via immunofluorescent staining, intracellular *Fn* (Fig. 1G), further supporting the CSI-Microbes analysis (Fig. 1A). To quantify the binding and invasion of *Fn* to CAFs, we performed flow cytometry analysis. *Fn* binding/invasion to CAFs and cancer cells increased with higher multiplicity of infection (MOI), unlike with the control *Ec* (Fig. 1H). Moreover, we observed *Fn* subsp. *animalis* (*Fn* 71), isolated from the human gastrointestinal tract (Strauss et al, 2011) and a representative of the *Fn* subsp. animalis clade C2 (Zepeda-Rivera et al, 2024) reported to have pro-tumorigenic functions in CRC (Kostic et al, 2013), to exhibit the highest affinity (Fig. 1I; Appendix Fig. S1D). Accordingly, transmission electron microscopy (TEM) confirmed that *Fn* was found adhering to the cell surface and residing in the cytoplasm of CAFs (Fig. 1J). To assess the viability of intracellular *Fn* in CAFs, we performed differential staining where we identified both dead and viable bacteria within CAFs (Appendix Fig. S1E). Altogether, our data demonstrates that *Fn* is capable of binding to and invading CAFs.

## *Fn* binds to CAFs through Gal-GalNAc

*Fn* is known to bind to tumor cells through several mechanisms, notably through Gal-GalNAc via its outer membrane protein Fap2 (Abed et al, 2016), as well as to E-cadherin via its adhesin FadA (Rubinstein et al, 2013). Yet, it is unclear whether *Fn* is using

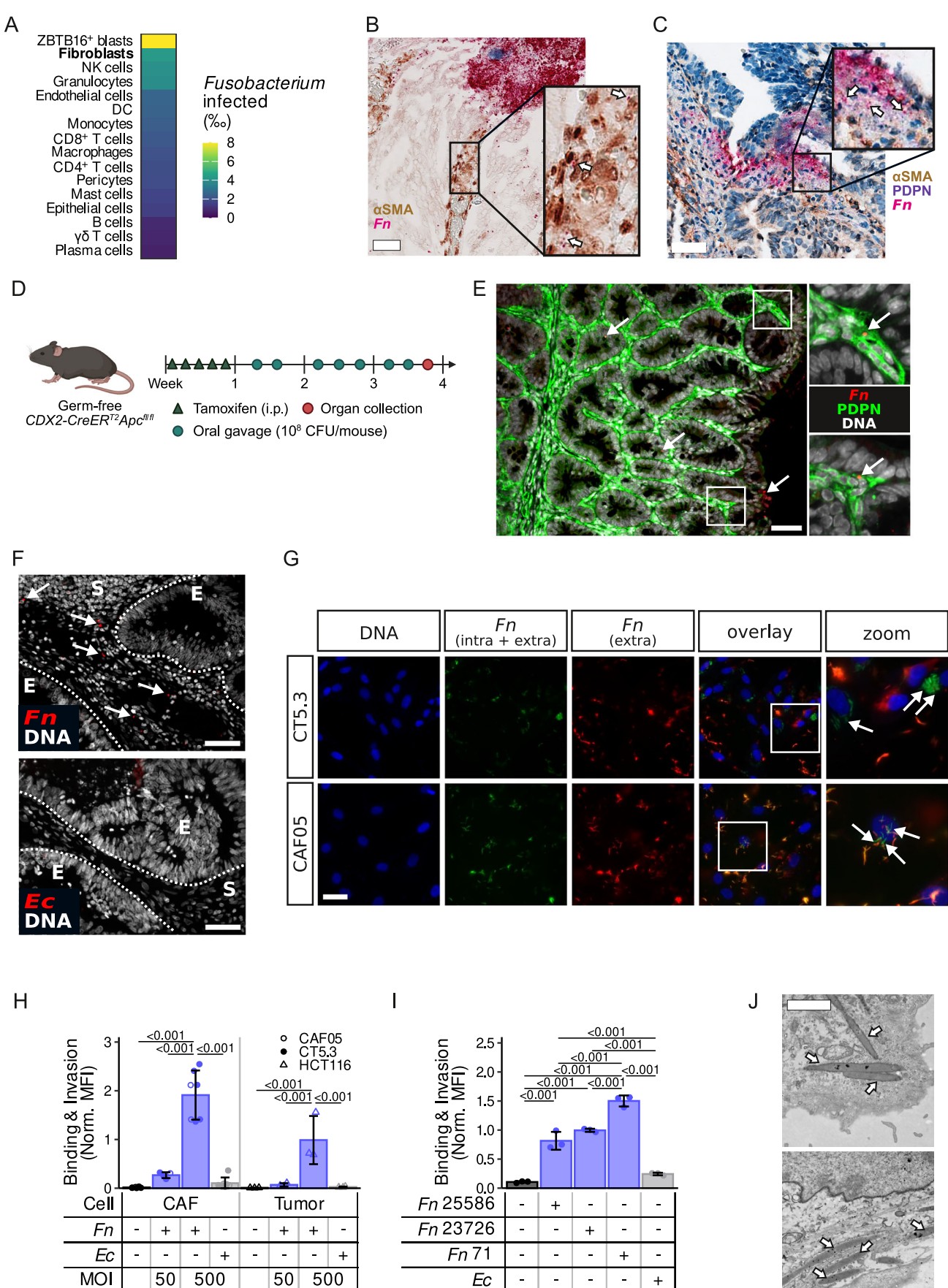

**Figure 1.** ***Fn*** **is present in the stromal compartment of CRC where it binds to and invades CAFs.**

(A) Re-analysis of the CSI-Microbe Pelka2021 dataset (Robinson et al, 2024; Pelka et al, 2021) showing *Fusobacterium*-infected cells per mille (‰) for each cell type in the tumor. (B, C) In situ hybridization staining of human CRC tissue with αSMA in brown (B, C), PDPN in purple (C), and *Fn* in pink (B, C; indicated by white arrows) in two independent stage III, microsatellite stable CRC patients. Scale bar = 50 µm. The image in (C) was visualized using a gamma correction of 0.5 to enhance visibility. (D) Schematic representation of the germ-free *CDX2-CreER$^{T2}$Apc$^{fl/fl}$* experiment. Briefly, tumorigenesis was induced by i.p. injections of tamoxifen (75 mg per kg of body weight) for five daily injections, followed by three times a week oral gavage with $10^8$ CFU/mouse of *Fn* 71 or PBS control. (E) Representative immunofluorescent images showing *Fn* 71 (red, white arrows), stained with an OMP *Fn*-specific antibody, colocalized with the stromal marker PDPN (green), and DAPI (gray) in a dysplastic region of the colon from the mouse germ-free *CDX2-CreER$^{T2}$Apc$^{fl/fl}$* experiment. Images are representative of two mice from one experiment. Scale bar = 50 µm. (F) Attachment and binding assay of *Fn* 71 and *Ec* (red, indicated by white arrows, DAPI in gray) on human CRC tissue. Epithelium (E) and stromal (S) stromal compartments separated by white dashed line. Images are representative of one patient and two independent experiments. Scale bar = 50 µm. (G) Fluorescence images of CAFs (CT5.3 and CAF05) following co-culture with *Fn* 25586 for 2 h (MOI 50). Nuclei were stained with DAPI (blue) and bacteria with both CFSE (green, staining intra- and extracellular bacteria) and FMAS (red, staining only extracellular bacteria). Bacteria that potentially invaded CAFs appear as CFSE$^+$ and FMAS$^-$ and are highlighted by white arrows on the zoom area of the overlay. Pictures are representative of two independent experiments, with two independent cell lines. Scale bar = 20 µm. (H) Quantification of the binding and invasion of CFSE-labeled *Fn* 25586 (MOI 50 and 500) or *Ec* (MOI 500) on CAFs (CAF05 and CT5.3) or HCT116 CRC cells after a 2-h co-culture by flow cytometry ($n = 5$, $n = 2$ and $n = 3$ for CT5.3, CAF05 and HCT116, respectively). (I) Quantification of the binding and invasion of *Fn* 25586, *Fn* 23726, *Fn* 71 and *Ec* (MOI 50) on CT5.3 CAFs after a 4-h co-culture ($n = 3$ biological replicates). (J) Representative TEM images of *Fn* 71 invasion (indicated by white arrows) after a 4-h co-culture with CT5.3 CAFs (MOI 50, $n = 1$ experiment). Scale bar = 5 µm. PDPN podoplanin, αSMA alpha smooth muscle actin, i.p. intraperitoneal. Bar and error bars in (H, I) show the mean ± SD and data points the values from each biologically independent experiment. Statistically significant differences were determined using a repeated measure ANOVA followed by Tukey's HSD post hoc test. Source data are available online for this figure.

similar routes to attach and bind to CAFs. To gain insights into the binding potential of *Fn* to CAFs, based on our knowledge of its binding on epithelial cells, we analyzed the expression of N- or O-glycan biosynthesis-related gene sets, as well as cadherins, using our RNA-Seq dataset of patient-derived fibroblasts and tumor cells (Fig. 2A). CAFs were found to express genes related to N- and O-glycan biosynthesis, as well as cadherins (Fig. 2B,C). Importantly, we observed a reduced attachment of *Fn* to CAFs when using the *fap2 Fn* mutant compared to wild-type *Fn* (Fig. 2D). Yet, we did not observe a reduction in binding with the *fadA* mutant *Fn* (Fig. 2D), suggesting that the binding to CAFs is preferentially mediated through the Gal-GalNAc-Fap2 axis. Accordingly, we were able to reduce the binding capacity of *Fn* to CAFs by pre-incubating the CAFs with N-acetyl-D-galactosamine (GalNAc) (Fig. 2E). Our data indicate that *Fn*'s adhesin Fap2 binds to CAFs, most likely via Gal-GalNAc motifs. However, the partial reduction in binding observed when treating cells with GalNac or the mutants, indicates that this represents only one of the potential mechanisms involved in bacterial attachment and invasion to CAFs.

## *Fn* polarizes fibroblasts to inflammatory CAFs

We next aimed to explore the impact of *Fn* presence in the stromal compartment, particularly its effect on CAFs. In CRC, two major CAF subtypes have been described and are now commonly accepted, namely myCAFs and iCAFs (Karta et al, 2021; Öhlund et al, 2017; Chhabra and Weeraratna, 2023; Nurmik et al, 2020). While we and others have shown that iCAFs are associated with poor prognosis in CRC (Koncina et al, 2023; Nicolas et al, 2022), the prognostic significance of myCAFs remains less clear (Jenkins et al, 2025; Choi et al, 2013). To explore how *Fn* affects the CAF subtype in human CRC patients, we leveraged the TCGA CRC dataset and performed a PathSeq analysis on the transcriptomics data to characterize the bacterial presence in patients' tumor samples. We used the calculated scores to define the 25% patients exhibiting the highest *Fn* abundance as *Fn$^{hi}$* and the remaining ones as *Fn$^{lo}$* (Fig. 3A), as has been done previously (Salvucci et al, 2022). When further looking at the gene expression levels in both group of patients, we observed that iCAF markers, represented by the pro-

inflammatory genes CXCL1, CXCL2, CXCL8, and IL-6 as well as the ECM remodeling gene MMP1, were higher in *Fn$^{hi}$* patients (Fig. 3B). On the other hand, myCAF markers were decreased in *Fn$^{hi}$* patients (Fig. 3B). To assess whether *Fn* affects the switch between both CAF subtypes, we analyzed the expression changes of the iCAF marker *IL-6* and the myCAF marker *ACTA2* at the RNA level in *Fn*-treated CAFs. The presence of *Fn* increased the expression of *IL-6* while it decreased the expression of *ACTA2* (Fig. 3C). Next, we used flow cytometry to analyze the expression of additional markers previously reported to characterize myCAFs and iCAFs (Nurmik et al, 2020; Elyada et al, 2019) (Appendix Fig. S2A). Exposure of CAFs to *Fn* did not alter their viability (Appendix Fig. S2B). While we did not detect a difference in the expression of the myCAF marker αSMA on the protein level, we observed that the myCAF marker PDGFRβ (Elyada et al, 2019) was decreased in CAF cultures upon *Fn* and *Ec* exposures (Fig. 3D). Interestingly, the iCAF markers PDGFRα, Lamin A/C and PDPN showed an increasing trend following exposure to *Fn* (Fig. 3D; Appendix Fig. S2C–F), a response not observed with *Ec* treatment (Fig. 3D; Appendix Fig. S2C,D).

Individual markers for CAFs, including iCAF and myCAF subsets, do exist, but have limitations (Nurmik et al, 2020; Kalluri, 2016). To better identify *Fn*-induced CAF phenotypes by considering a larger gene signature panel, we generated a *Fn*-treated CAF RNA-seq dataset (Fig. 3E). Consistently, we noted an increased expression of iCAF marker genes (Elyada et al, 2019) and a decreasing trend of the myCAF ones (Elyada et al, 2019; Chen et al, 2023) (Fig. 3F). To link our observations with clinical data, we analyzed-survival data from the TCGA cohort and observed reduced survival in patients with high *Fn* abundance within the CMS4 or CRIS-B subtypes—characterized by a prominent stromal component (Guinney et al, 2015; Isella et al, 2017)—consistent with findings from Salvucci et al (Salvucci et al, 2022). Notably, *Fn*-high patients belonging to the CMS4/CRIS-B subtypes exhibited increased iCAF and reduced myCAF scores, potentially suggesting a shift of CAFs toward an iCAF phenotype (Fig. 3G,H). Further analyses in independent patient cohorts are required to clarify *Fn*'s capacity to modulate CAF phenotype switching from myCAFs to iCAFs.

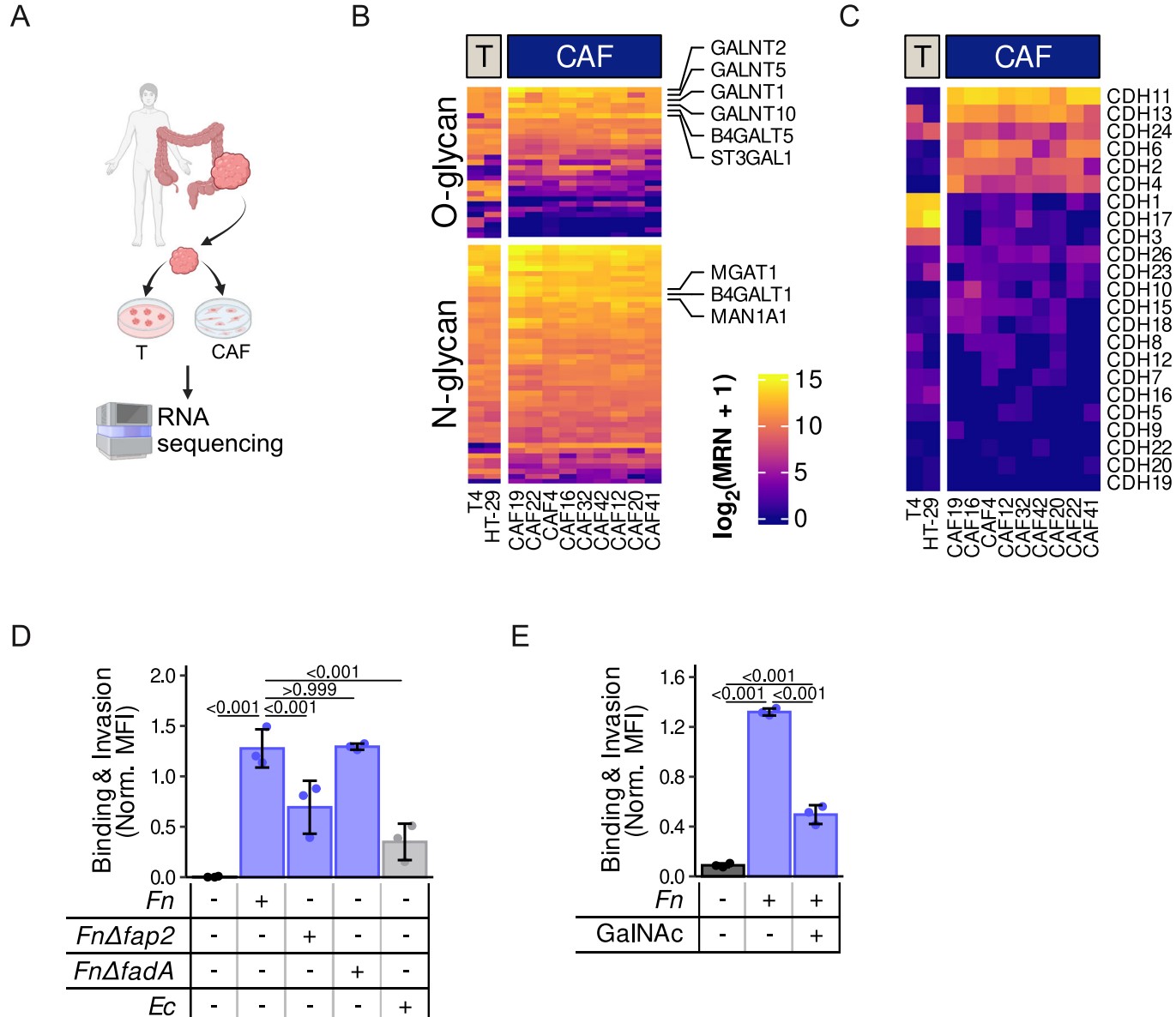

**Figure 2.** *Fn* **binds to and invades CAFs, potentially through the Gal-GalNAc-Fap2 axis.**

(A) Schematic representation of the generated RNA-seq dataset. CAFs and tumor spheroids (T) were isolated from fresh human CRC tumor biopsies, cultured and sent for sequencing. (B, C) Heatmap showing the expression of genes (log₂ of median ratio-normalized expression values) responsible for N- and O-glycan biosynthesis (KEGG_N_GLYCAN_BIOSYNTHESIS and KEGG_O_GLYCAN_BIOSYNTHESIS gene sets, respectively) (B) and cadherins (C) in the generated RNA-seq dataset. Columns show the expression assessed in independent experiments of patient-derived (T4) and HT-29 tumor spheroids as well as patient-derived CAFs (n = 9 patients: CAF4, CAF12, CAF16, CAF19, CAF20, CAF22, CAF32, CAF41 and CAF42). Expression values are median ratio-normalized counts on a log₂ scale. (D) Binding and invasion of wild-type *Fn* 23726, *fap2* and *fadA* mutant on CT5.3 cells after a 4-h co-culture (MOI 50) assessed using the CFSE dye by flow cytometry (n = 3 independent experiments). (E) Binding and invasion of *Fn* 71 on CT5.3 (4-h co-culture, MOI 50) cells in the presence of GalNAc assessed using the CFSE by flow cytometry (n = 3 independent experiments). T tumor. Bar and error bars in (D, E) show the mean ± SD and datapoints the values from each biologically independent experiment. Statistically significant differences were determined using a repeated measure ANOVA followed by Tukey's HSD post hoc test. Source data are available online for this figure.

## *Fn*-exposed CAFs exhibit enhanced production of inflammatory cytokines and increased metabolic activity

iCAFs have been characterized by the secretion of inflammatory cytokines, chemokines and tumor-promoting growth factors (Caligiuri and Tuveson, 2023). Therefore, to validate the observed iCAF-like phenotype, we performed cytokine profiling to characterize the expression of pro-inflammatory cytokines and chemokines in *Fn*-treated CAFs. We observed higher levels of IL-6, CCL20 and CXCL1 in *Fn*-treated fibroblasts as compared to non- or *Ec*-treated CAFs (Fig. 4A). This observation is in line with studies that showed that *Fn* increased IL-6, IL-8 (CXCL8), and CXCL1 expressions in both human normal gingival fibroblast (Rath-Deschner et al, 2020) and CRC cells (Rubinstein et al, 2013;

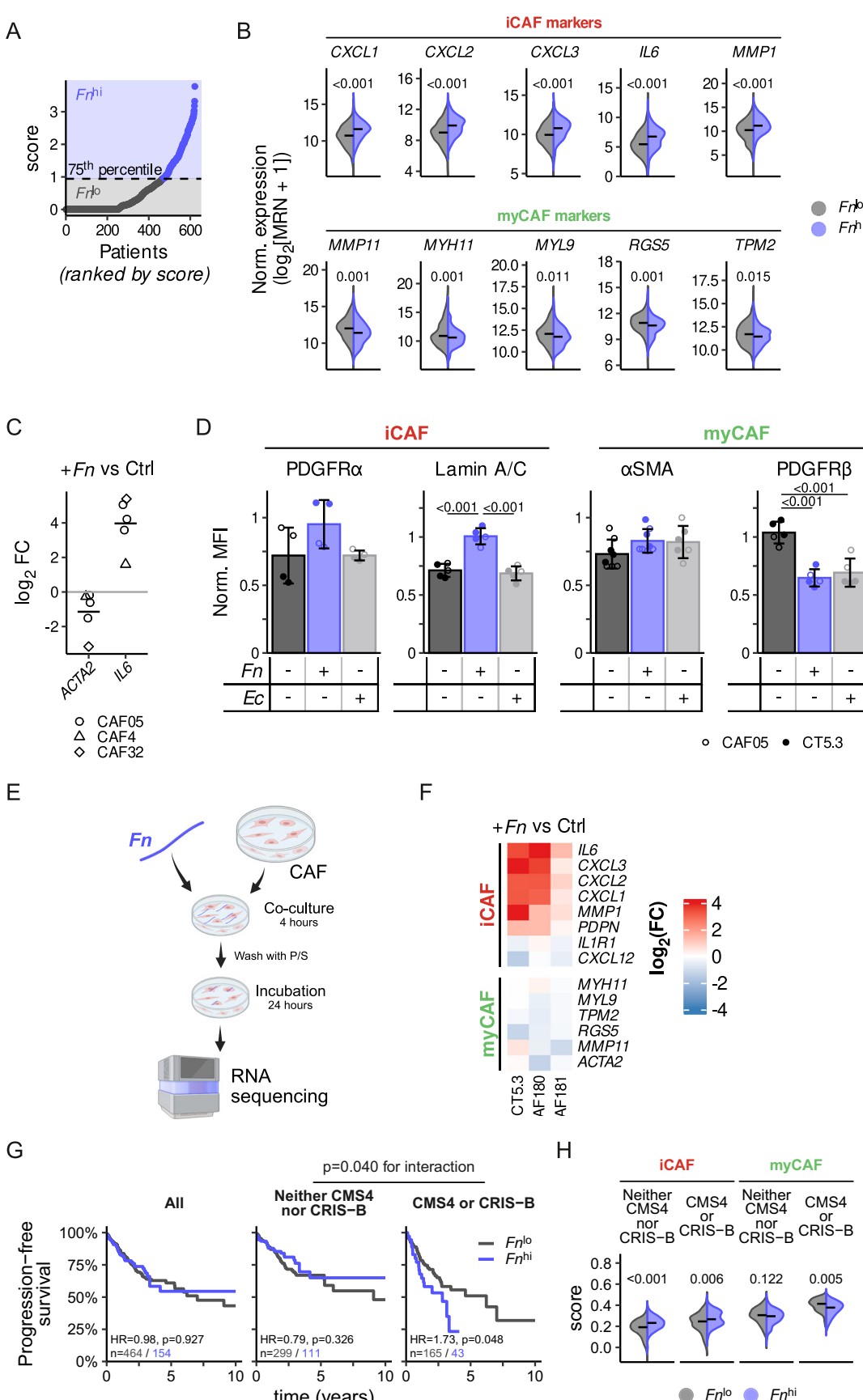

**Figure 3.** *Fn* treatment polarizes CAFs to an inflammatory phenotype.

(A) ComBat corrected *Fn* PathSeq scores in TCGA COAD and READ patients ($n = 622$ patients), ranked by score; the 75th percentile was used to define $Fn^{lo}$ and $Fn^{hi}$ group. (B) Gene expression of iCAF and myCAF markers in $Fn^{lo}$ and $Fn^{hi}$ patients, shown as $\log_2$ median ratio-normalized values. (C) $\log_2$ fold change in *IL-6* and *ACTA2* expression in *Fn*-treated versus untreated CAF05 cells (2-h co-culture, MOI 500) ($n = 3$ biologically independent experiments) as well as in patient-derived CAFs (CAF4 and CAF32, $n = 1$ for each). (D) Expression of the myCAF markers αSMA ($n = 8$) and PDGFRβ ($n = 5$), and the iCAF markers PDGFRα ($n = 4$) and Lamin A/C ($n = 5$) 24 h after a 2-h co-culture with either *Fn* 25586 or *Ec* (MOI 500) as measured by flow cytometry. (E) Schematic representation of the generated RNA-seq dataset. CT5.3, or the patient-derived CAF180 and CAF181 CAFs were exposed to *Fn* 71 (MOI 50) for 4 h, washed with penicillin/streptomycin (P/S)-containing medium and incubated for a further 24 h before RNA sequencing ($n = 3$ independent cell lines). (F) Heatmap of the RNA-Seq expression $\log_2$ fold change in CT5.3 and patient-derived CAFs exposed to *Fn* 71 compared to untreated CAFs. (G) Kaplan–Meier curve of progression-free survival by *Fn* load in either the complete CRC cohort or by CMS4/CRIS-B classification. (H) iCAF and myCAF scores (calculated using the *R* package singscore and the gene sets from (F)) by *Fn* load and CMS4/CRIS-B classification. Bar and error bars in (D) show the mean ± SD, and datapoints the values from each biologically independent experiment. Statistically significant differences were determined in (A, H) using a two-tailed *t* test with Holm's method adjusted *P* values, in (D) using a repeated measure ANOVA followed by Tukey's HSD post hoc test and in (G) using an ANOVA on the Cox proportional hazard models to assess the interaction significance. Source data are available online for this figure.

Casasanta et al, 2020; Rossano et al, 1993). Besides the detected increase of inflammatory cytokines in the supernatant of *Fn*-treated CAFs, we also observed higher levels of several shed membrane-associated proteins such as extracellular matrix metalloproteinase inducer (Emmprin/CD147), urokinase-type plasminogen activator receptor (uPAR) and dipeptidyl peptidase IV (DPPIV/CD26) (Fig. 4A), all described to be involved in tumorigenesis and poor prognosis (Lam et al, 2014; Lescaille et al, 2012; Lv et al, 2021). ELISA measurements further confirmed the significant increase in CXCL1, IL-6 and IL-8 secretion from CAFs following *Fn* stimulation (Fig. 4B–D). *Fn* triggered CXCL1 expression in CAFs to a higher extent than *Ec* and untreated control (Appendix Fig. S3A). One possible mechanism by which Gram-negative bacteria, including *Fn*, may induce an iCAF phenotype is through lipopolysaccharides (LPS)-mediated activation of Toll-like receptor 4 (TLR4) signaling (Wang et al, 2025). Interestingly, the use of the TLR4 inhibitor TAK-242, partially reversed the *Fn*-induced cytokine secretion (Appendix Fig. S3B), an effect which is shared with *Ec* and stronger in CT5.3 CAFs compared to primary CAFs. Of note, *Ec* proliferates much faster than *Fn*, resulting in different bacterial loads towards the end of co-culture with CAFs, which may account for some of the observed effects. Yet, this data suggests that LPS plays a role in the activation of iCAFs in the context of bacterial exposure (Wang et al, 2025).

Given the observed increase in pro-inflammatory cytokines, it is plausible that reactive oxygen species (ROS), which can be triggered by LPS (Cheng et al, 2015), may be involved in driving the CAF response, as ROS are known to activate transcription factors like NF-κB and STAT3 that regulate cytokine expression (Li et al, 2013). First, leveraging on a publicly available scRNA-Seq dataset (Lee et al, 2020) in which we previously labeled myCAFs and iCAFs (Koncina et al, 2023), we calculated a ROS gene signature score (using the HALLMARK_REACTIVE_OXYGEN_SPECIES_PATH-WAY geneset) and observed it to be increased in the iCAF cluster (Fig. 4E). Next, using the same geneset, we calculated a ROS gene signature score in the TCGA CRC dataset and observed that it was slightly but significantly increased in $Fn^{hi}$ patients (Fig. 4F), however, it is important to note that this data reflects the whole tumor and not only the stromal compartment. Nonetheless, experimentally, while *Fn* induced only a modest increase in cytoplasmic ROS levels, it led to a significantly higher increase in mitochondrial ROS levels in CAFs (Fig. 4G). In addition, as we did not observe a difference in cell viability (Appendix Fig. S3C), this increase in mitochondrial ROS might reflect mitochondrial

dysfunction and potentially metabolic reprogramming of CAFs. Mitochondrial ROS is linked to increased activity of the TCA cycle and oxidative phosphorylation. As such, we measured the relative enrichment in [U-$^{13}$C]glutamine fed into the TCA cycle and observed a statistically significant increase in enrichment into glutamate and alpha-ketoglutarate (Appendix Fig. S3D,E). Glutamine is one of the main substrates feeding into the TCA cycle, and an increase in its relative flux indicates higher activity of the cycle, supporting the notion of *Fn*-exposed CAFs being more active as characterized by an increased ROS level and increased TCA cycle activity. Altogether, our data suggest that *Fn*-exposed CAFs are metabolically more active.

## *Fn*-exposed CAFs increase the migration and invasion of CRC tumor cells in vitro and in vivo

In order to characterize how *Fn*-reprogrammed CAFs affect tumorigenesis, we treated CRC cells with conditioned media (CM) of CAFs previously co-cultured with bacteria, followed by RNA sequencing analysis (Fig. 5A). We identified 115 genes that were differentially expressed in tumor cells stimulated with *Fn*-CAF-CM as compared to tumor cells stimulated with the control CAF-CM (Fig. 5B). Furthermore, Ingenuity Pathway Analysis (IPA) predicted a higher migration activation score for *Fn*-CAF-CM-treated tumor cells (Fig. 5C), suggesting that the *Fn*-induced iCAF secretome may play a significant role in the migratory and invasion phenotype of tumor cells.

To confirm that *Fn*-activated CAFs are involved in tumor cell migration, we used a Transwell-based co-culture system (Fig. 5D). We observed an increase in HCT116 cell migration upon co-culture with CAFs, an effect that was further exacerbated in an MOI-dependent manner upon *Fn*, but not *Ec*, pretreatment of CAFs (Fig. 5E,F). Similar effects were observed in the invasion of another CRC cell line, SW480 (Appendix Fig. S4A). In addition, the migration of the CAFs themselves was not influenced by *Fn* or *Ec* (Appendix Fig. S4B,C), ruling out any direct effect of *Fn* on CAF migration. To validate these findings, we moved to a more complex 3D spheroid model in which we cultured mCherry-labeled CRC cells as 3D spheroids and co-incubated them with GFP-labeled CAFs. These complex 3D spheroid models were then treated with CM of CAFs co-cultured either with a blank medium control, *Ec* or *Fn*. Tumor cell outgrowth from the spheroids into the collagen matrix was monitored as a proxy for invasion and aggressiveness (Fig. 5G). Using this model, we demonstrated that supernatants

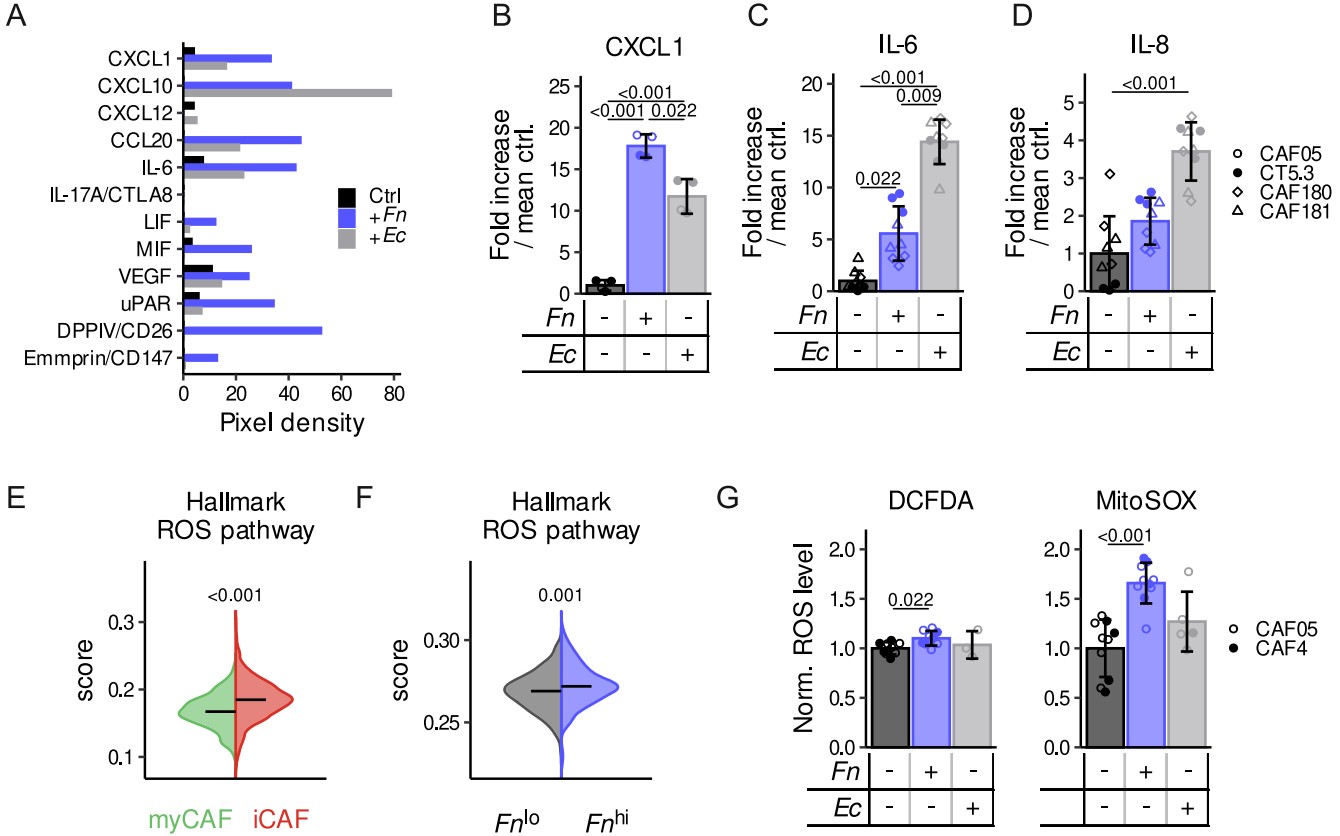

**Figure 4. *Fn* exposure induces CAFs to secret pro-inflammatory cytokines and increases their metabolic activity.**

(A) Cytokine profiling of medium collected 24 h after a 2-h co-culture of *Fn* 25586 with CT5.3 CAFs (MOI 500) using the Proteome Profiler Human XL Cytokine Array Kit (R&D Systems, $n = 1$ experiment). (B) CXCL1 concentration in CM 24 h after a 2-h co-culture of *Fn* 25586 and *Ec* (MOI 500) with CAF05 and CT5.3 CAFs, as determined by ELISA ($n = 3$ independent experiments with two independent cell lines). (C, D) IL-6 (C) and IL-8 (D) levels in CM 24 h after a 4-h co-culture of CT5.3, or the patient-derived CAF180 and CAF181 CAFs with *Fn* 71 and *Ec* (MOI 50), measured by ELISA ($n = 3$ independent experiments with three independent cell lines). (E) ROS geneset scores in iCAF and myCAF in the Lee CRC scRNA-Seq dataset. Scores were calculated using UCell on previously labeled CAFs (Koncina et al, 2023). (F) ROS geneset scores in the TCGA CRC dataset segregated into *Fn*lo and *Fn*hi (Fig. 3A). Scores were calculated using singscore. (G) Cytoplasmic ($n = 5$ for CAF05 and $n = 4$ for CAF4, H2DCFDA) and mitochondrial ($n = 6$ for CAF05 and $n = 4$ for CAF4, MitoSOX) ROS levels as analyzed by flow cytometry, in CAFs 24 h after a 2-h treatment with *Fn* 25586 or *Ec* (MOI 500). Data in (B–D, G) is shown as the mean ± SD, and the datapoints represent the values from each biologically independent experiment. The horizontal lines in (E, F) show the median. Statistically significant differences in (B–D, G) were determined using a nested ANOVA followed by Tukey's HSD post hoc test and in (E, F) using a two-tailed *t* test. Source data are available online for this figure.

from *Fn*-treated CAFs significantly enhanced the invasive potential of tumor spheroids compared to controls (Fig. 5H,I; Appendix Fig. S4D). Given our earlier observation of increased ROS in *Fn*-exposed CAFs—as well as the link of ROS to the iCAF phenotype (Jain et al, 2013) and the association of ROS with cancer cell invasion (Li et al, 2014)—we decided to investigate its role by inhibiting ROS using NAC, a well-established ROS scavenger. Strikingly, when CAFs were treated with NAC and the resulting supernatant was used (Fig. 5H,I), the *Fn*-induced increase in invasion was effectively reversed (Fig. 5H,I). Of note, the invasion of GFP-labeled CAFs was also assessed in the 3D culture system (Appendix Fig. S4E), revealing only minor differences in their invasive capacity, similar to our previous observation (Appendix Fig. S4C).

Finally, we used an in vivo tail vein metastasis assay to determine the metastatic potential of cancer cells, and assess their ability to colonize distant organs, most commonly the lungs. To this end, we performed tail vein injections of the CRC cell line HT-29

pre-stimulated with CM of CAFs previously co-cultured with bacteria (Fig. 6A). Our observations showed that CRC cells treated with CM from CAFs pre-stimulated with *Fn*, exhibited significantly enhanced metastatic outgrowth in the lungs compared to those treated with control CAF-CM or CM from *Ec*-stimulated CAFs (Fig. 6B,C). Altogether, our results suggest that exposure to *Fn* increases CAFs capacity to induce cancer cell migration and invasion.

## Discussion

In recent years, *Fn* has garnered considerable attention due to the increasing evidence of its overabundance and pathogenic activity in CRC. While previous studies have primarily focused on the direct interactions of *Fn* with tumor cells to characterize its carcinogenic role in CRC, we show in the current study that *Fn* also interacts with other cells within the TME, in particular CAFs. We demonstrate that *Fn* is localized within the stromal compartment

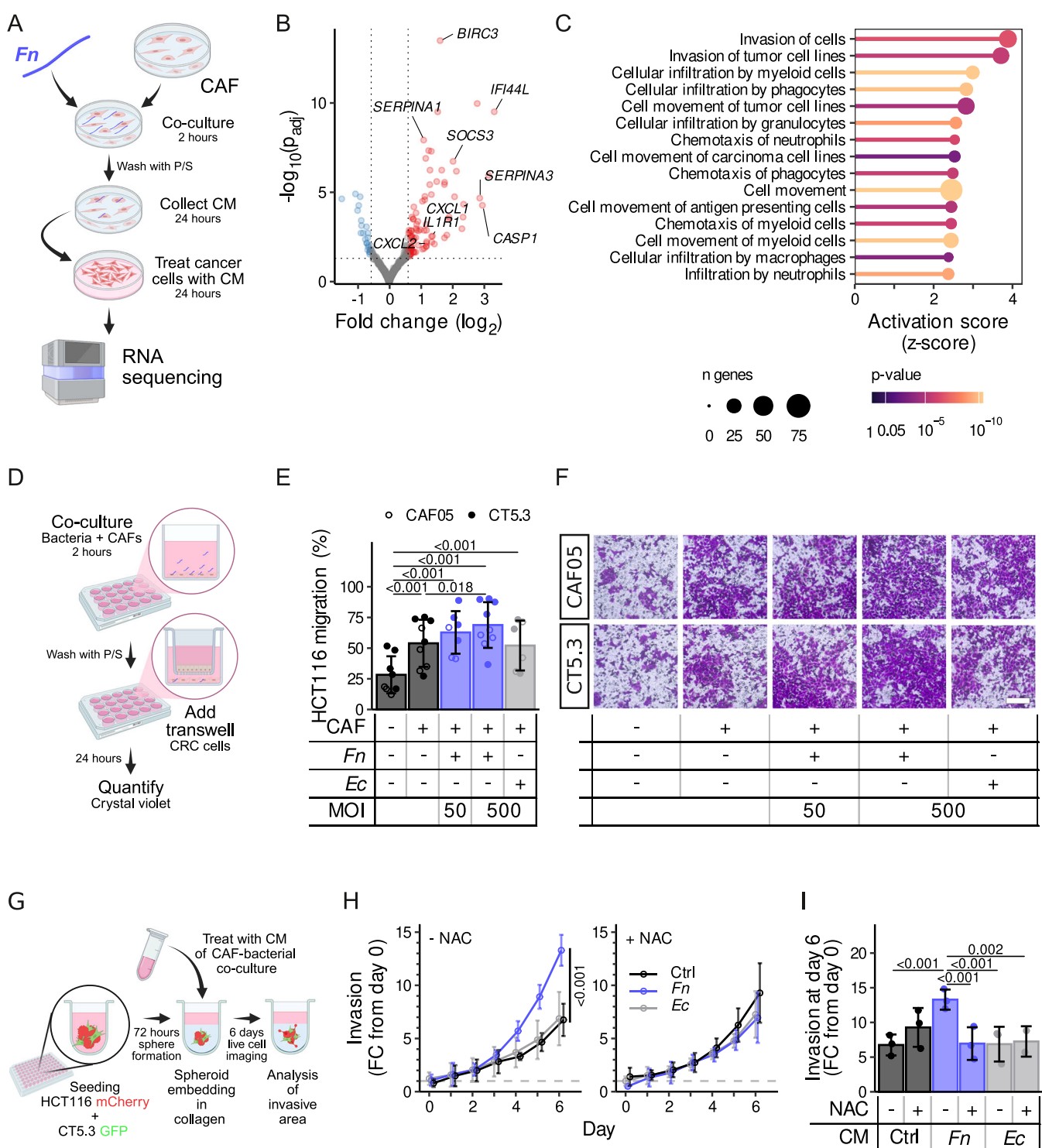

in vivo and has the potential to bind to and invade CAFs. We further show that *Fn*-exposed CAFs alter their phenotype, notably shifting towards iCAFs, characterized by increased secretion of CXCL1, IL-6, and IL-8, partially mediated by TLR4 signaling. The increased iCAF functionality, potentially linked to ROS, supports the migratory and invasive potential of tumor cells in vitro and in vivo. These findings suggest that apart from the direct effect of

*Fn* on cancer cells, *Fn* also plays a role in promoting tumor cell dissemination by activating CAFs within the TME.

*Fn* is an adhesive bacterium which possesses several adhesins on its surface, including Aid1 and FomA (Liu et al, 2010; Kaplan et al, 2014). Some of these adhesins, such as RadD, FadA and Fap2, have previously been recognized as virulence factors as they facilitate the bacterium's binding and invasion into tumor cells (Brennan and

◄ **Figure 5.  *Fn* promotes CAF-induced migration and invasion of tumor cells.**

(A) Schematic representation of the generated RNA-seq dataset. Tumor spheroids (HT-29) were treated with CM collected 24 h after a 2-h co-culture of *Fn* 25886, MOI 500 with CAF05 or the patient-derived CAF32 and CAF20 CAFs (*n* = 3 independent experiments with three different cell lines) compared to non-treated CAF-CM. (B) Volcano plot showing genes being differentially expressed (DESeq2 ashr shrunken |fold change| >1.5 and Benjamini and Hochberg method adjusted Wald test *P* value < 0.05) in *Fn*-CAF-CM- vs ctr-CAF-CM-treated tumor cells as outlined in (A). (C) Ingenuity pathway analysis of diseases and biological functions of the differentially expressed genes. Activation z-scores, *P* values and the number of genes are shown for migration and infiltration-related pathways. *P* values were obtained from the IPA implemented right-tailed Fisher's exact test. (D) Schematic representation of the Transwell assay. (E) Migration of HCT116 cells after 24 h of co-culture in a Transwell assay with CT5.3 CAFs that had been pretreated for 2 h with either no bacteria or with bacteria (*Fn* 25586 and *Ec*, MOI 50 or 500) (*n* = 9 independent experiments). (F) Representative images of crystal violet-stained HCT116 cells that migrated in the Transwell setup at endpoint. Scale bar = 500 μm. (G) Schematic representation of the in vitro complex 3D spheroid co-culture experimental setup. mCherry-labeled HCT116 CRC cells are co-embedded in collagen with GFP-labeled CAFs (CT5.3). These multi-component spheroids are then treated with CM from CT5.3 CAFs cultured alone (blank medium control) or CM from CT5.3 CAFs co-cultured with *Fn* 71 or *Ec* (4-h co-culture, MOI 50), in the presence or absence of NAC (5 mM). (H) Quantification of invasion in the 3D complex spheroid model measured by HCT116 outgrowth (mCherry) over 6 days (*n* = 3 independent experiments, 2–5 technical replicates included per experiment). (I) Quantified invasion of HCT116 (mCherry) outgrowth at endpoint (day 6 from the data shown in panel 5H). CM conditioned medium, NAC N-acetyl-l-cysteine. The bar chart and error bars in (E, I) show the mean ± SD. Statistically significant differences were determined using a nested ANOVA followed by Tukey's HSD post hoc test. The error bars in (H) show the mean ± SD and statistically significant differences determined using pairwise two-way repeated measure ANOVAs (time × treatment). The interaction term *P* values were adjusted using Holm's method. Source data are available online for this figure.

Garrett, 2019; Slade, 2021; Abed et al, 2016; Zhang et al, 2024). We now demonstrate that one of the mechanisms by which *Fn* binds to and invade CAFs, involves the Gal-GalNAc-Fap2 axis. Importantly, we were not able to detect any stained bacteria upon co-culture of *Ec* with fibroblasts, suggesting that not all bacteria are able to bind nor invade CAFs and that the binding/invasion is specific to pathogenic species. Additionally, we observed differences in CAF binding affinity among *Fusobacterium nucleatum* subspecies and strains, which may be attributed to variation in the expression of virulence factors, particularly Fap2, as recently reported (Ma et al, 2023; Ponath et al, 2021).

Exposure of CAFs to *Fn* altered several CAF markers, including αSMA, PDPN, PDGFRs, and Lamin A/C. Intriguingly, we observed a significant increase in Lamin A/C expression in *Fn*-treated CAFs. Lamin A/C has been proposed as an iCAF marker, its upregulation may promote gene expression profiles, mediated through the PI3K/AKT/PTEN signaling pathway (Kong et al, 2012), that favor tumor–CAF interactions and CRC progression (Kong et al, 2012; Willis et al, 2008; Urciuoli et al, 2021). In pancreatic cancer, iCAFs also show high Lamin A/C and pro-inflammatory cytokines such as IL-6, IL-8, CXCL1, and CXCL12 (Elyada et al, 2019). Likewise, we noted several cytokines, among which IL-6, IL-8, and CXCL1, to be secreted by *Fn*-treated CAFs.

We observed increased CRC cell migration and invasion following co-culture with *Fn*-exposed CAFs both in vitro and in vivo. Further studies are required to elucidate the mechanisms underlying the strain-specific variability in *Fn* binding to CAFs and the resulting downstream effects on tumor cells. While *Ec* also induces the expression of cytokines such as CXCL1, IL-6, and IL-8 in exposed CAFs, potentially through LPS-induced TLR4 signaling, in agreement with prior findings (Wang et al, 2025), it fails to promote invasion in vitro—whether in Transwell assays or complex 3D models—, or in vivo, and does not enhance direct cancer cell invasiveness, as previously shown by us (Ternes et al, 2022). Our findings suggest that *Fn* contributes to CRC progression by modulating CAFs through multiple mechanisms. First, *Fn* is physiologically localized within the stromal compartment in vivo as demonstrated in this study. Second, *Fn* has the unique ability to bind to and invade not only cancer and immune cells, but also CAFs, a capability not shared by all bacteria. Third, this invasive capacity is combined with *Fn*'s ability to switch CAFs into an iCAF

phenotype, partially through TLR4 signaling, and associated with increased metabolic activity and ROS production. The elevated ROS levels and metabolic activity observed in *Fn*-exposed CAFs are associated with an increased invasion of tumor cells, as this phenotype was reversed by the antioxidant NAC. Further studies are needed to determine whether the observed metastatic phenotype is driven exclusively by ROS-induced invasion or whether enhanced proliferation also contributes to increased metastatic burden. Moreover, while ROS appears to be a key driver, it is unlikely to be the only mechanism involved. Future work should investigate additional pathways by which *Fn*-exposed CAFs influence tumor cell invasion.

Altogether, we demonstrate here a multifactorial role by which *Fn* shapes the TME and favors cancer cell invasiveness. Additionally, within the physiological disease context, *Fn* may exert an additive effect as many previous studies have already demonstrated a direct effect of *Fn* on tumor cells. Our results now add a new layer of complexity to the initial understanding of *Fn*-interaction in CRC by highlighting the role of the TME, in particular CAFs, as an important mediator of *Fn*-induced tumor-promoting effects. However, to fully delineate the specific contributions of *Fn* across different cellular compartments, further complex analyses such as spatial transcriptomics combined with fluorescent in situ hybridization (FISH), are required to dissect how these compartment-specific roles influence tumor progression. Additionally, in-depth studies are now necessary to link the *Fn*-induced CAF remodeling to clinical data, including patient outcome. Finally, as *Fn* exists within a broader microbial community in the TME, future studies should explore how microbial interactions influence CAF phenotypes and collectively shape the stromal compartment.

## Methods

### Patient sample collection and preparation

All patient samples were obtained through our established CRC cohort and handled according to all institutional guidelines and regulations (ERP-16-032), based on previously described biospecimen handling standards. All patient samples were donated

## A

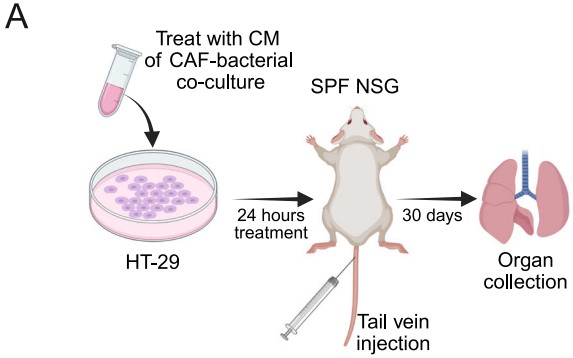

## B

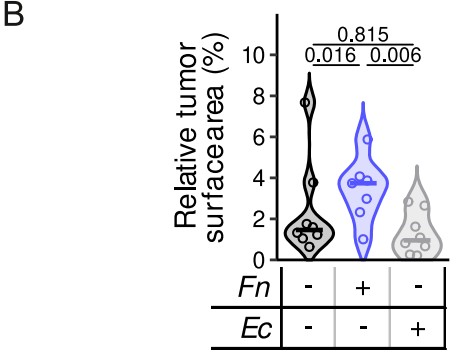

## C

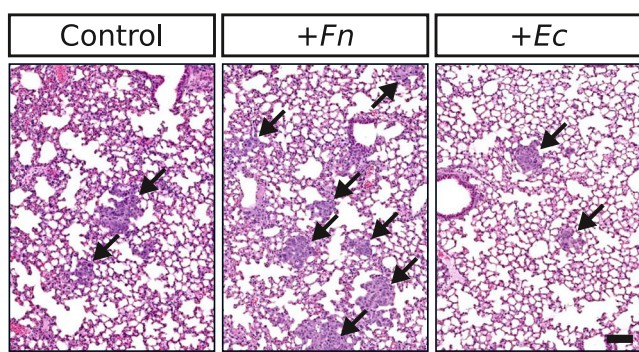

**Figure 6.** *Fn* promotes CAF-induced metastatic spreading of tumor cells in vivo.

(**A**) Schematic representation of the in vivo experimental setup. SPF NSG mice were tail vein injected with $1 \times 10^6$ HT-29 cells, which were pretreated with CM of CT5.3 CAFs cultured alone or CM of CT5.3 CAFs co-cultured with *Fn* 71 or *Ec* (4-h co-culture, followed by a 24-h incubation, MOI 50). Lungs were harvested after 30 days ($n = 8$ mice for the Ctrl and *Ec* conditions and $n = 7$ for the *Fn* condition). (**B**) Quantification of the relative tumor surface area in percentage per imaged lobe (four lobes were analyzed per mouse). (**C**) Representative H&E images of tumors (indicated by black arrows) in the lungs on day 30. CM conditioned medium. Scale bar = 100 μm. Statistically significant differences were determined using an ANOVA on a robust linear model fit (using the lmRob function in the R package robust) followed by Tukey's multiple comparison method (using the R package multcomp). Source data are available online for this figure.

freely, with informed written consent. Ethical approvals were obtained for samples collected in Luxembourg from the Comité National d'Ethique de Recherche (CNER) (Reference 201009/09), and in Spain approved by the Institutional Ethics Committee of Vall d'Hebron University Hospital (PR(AG)210/2019).

## Cell cultures

All cell lines used in this study are listed in Appendix Table S1. Patient-derived tumor fibroblast cultures were prepared as previously described (Koncina et al, 2023). Briefly, tissue specimens were carefully washed with DMEM-F12 (DMEM-F-12, GlutaMAX™ Supplement from Gibco, 31331028) supplemented with penicillin-streptomycin (P/S, 1% (v/v) Life Technologies, 15140122) and antibiotic and antimycotic agents (1× of Antibiotic–Antimycotic Solution, Life Technologies, A5955-100ML). Tissue samples were then minced into 1–2 mm³ pieces and dissociated into a single-cell suspension using the human tumor dissociation kit (Miltenyi Biotec, 130-095-929), according to the manufacturer's specifications. The mixture was then filtered through a 70 μm pore-sized cell strainer, and the single-cell suspension was transferred to cell culture-treated Petri dishes (∅10 cm) containing DMEM-F12, supplemented with 10% fetal bovine serum (FBS), 1% P/S. After a period of 30 min to allow for enrichment in fibroblasts through differential attachment, the single-cell suspension was removed, and the Petri dishes were gently washed with PBS to remove floating cells. Attached cells were then cultured and amplified in Endothelial Growth Medium 2-MV (Promocell, C-22010) supplemented with 1% P/S and 1% antibiotic/antimycotic (Gibco, 15240062). Non-transduced primary fibroblasts were transferred to DMEM-F12 (10% FBS, 1% P/S) for experimental assays. CAF05 were purchased from Neuromics and maintained in Mesenchymal stem cells (MSCs) medium (Vitro Biopharma, PC00B1-500) with 1% P/S but transferred to DMEM-F12 (10% FBS, 1% P/S) for experimental assays. Immortalized primary human colon fibroblasts (CT5.3) were generously donated by Prof. Olivier de Wever, and were cultivated according to previously established protocols (Stadler et al, 2017).

The commercial HCT116, HT-29 and SW480 CRC cell lines were purchased from the American Type Culture Collection (ATCC) and maintained in DMEM-F12 supplemented with 10% FBS and 1 U/ml P/S at 37 °C, 5% $CO_2$. To generate the mCherry-labeled HCT116 and GFP-labeled CT5.3, the respective cell lines were transduced with the mCherry virus rLV.EF1.mCherry-9 (Takara Bio Europe, 631281) and the GFP virus, pLV[Exp]-Bsd-CMV > MaxGFP (Bio Connect, ECOLI-VB210908-1092nqh), according to the manufacturer's protocol. 3D epithelial tumor spheroids were cultivated in spheroid culture medium (nSCM, DMEM-F12 with 25 mM glucose, B-27, insulin (4 U/l), heparin (4 μg/ml), EGF (20 ng/ml), FGF (20 ng/ml) and 1% P/S) at 37 °C, 5% $CO_2$ in ultra-low-attachment plates, as previously described (Qureshi-Baig et al, 2016, 2017). SW480 were acquired recently, and all other used cell lines were authenticated in November 2024.

## Bacterial cell culture

Bacterial strains used in the study are listed in Appendix Table S2. *Fusobacterium nucleatum* (ATCC 23726, referred to as *Fn* 23726) and *Fusobacterium nucleatum* subsp. *animalis* 7_1 (referred to as *Fn* 71) strains were a kind gift from Prof. Dr. Daniel Slade, VirginiaTech. *Fusobacterium nucleatum* (ATCC 25586, referred to as *Fn* 25586) was obtained from the DSMZ and *Escherichia coli* MG1655 (referred to as *Ec*) from the laboratory of Prof. Dr. Paul Wilmes (Luxembourg). The *fadA* and *fap2 Fn* mutants were kindly provided by Prof. Dr. Jörg Vogel (Ponath et al, 2022) (for detailed

generation, please see the next section). Bacteria were cultivated in an anaerobic chamber at 37 °C with 95% $N_2$ and 5% $CO_2$ using BD DIFCO™ Columbia Broth (Biosc. USA) or Anaerobic fluid (AF) media (Weiss et al, 2022).

## Generation of Δ*fap2 F. nucleatum* subsp. *nucleatum* ATCC 23726

Electro-competent *F. nucleatum* subsp. *nucleatum* ATCC 23726 cells were prepared and transformed as previously described (Ponath et al, 2022). In brief, the vector pVoPo-04 was used to construct a fusobacterial suicide plasmid containing flanking regions of *fap2* (see Appendix Table S3). Approximately 5 μg of desalted plasmid (see Appendix Table S3) were transformed into *Fn* and plated on thiamphenicol-containing BHI-S plates for initial selection of integration. A single colony was inoculated in Columbia broth with the addition of 100 ng/ml anhydro-tetracycline (ATc) and allowed to grow for 4 h upon which the culture was plated on ATc-containing BHI-C plates (200 ng/ml). Plasmid loss was confirmed by re-streaking colonies on BHI-C plates with and without thiamphenicol, followed by PCR verification.

## Direct co-culture of fibroblasts and bacteria

Cells were pretreated for 1 h at 37 °C with 1× Brefeldin (eBiosience™, 00-4506-51) or 100 mM N-Acetyl-D-galactosamine (GalNac, Sigma-Aldrich, A2795) or 5 mM N-Acetyl-L-cysteine (NAC, Sigma-Aldrich, A9165) or 1 μM TAK-242 (Tocris Bioscience, 6587/5). Fibroblasts (primary established or commercial cell lines) were co-incubated with *Fn* or *Ec* at multiplicity of infections (MOIs) of 50 or 500. After 2 or 4 h, cells were washed three times for 20 min, each with 2% P/S containing medium. Fibroblasts were then cultivated for an additional 24 h in fresh medium at 37 °C, 5% $CO_2$ containing 1% P/S. Co-culture specifics are specified in the respective figure legends.

## Immunofluorescence imaging for *Fn* binding and invasion

CAF05 and CT5.3 CAFs were seeded in μ-Slide eight-well chamber (Ibidi, 80826) at 40,000 cells/well. Before co-culture, *Fn* was stained with CFSE (CellTrace™ CFSE Cell Proliferation Kit, for flow cytometry from Invitrogen, C34554) incubated for 20 min to label at RT on a shaker. CAFs were then co-incubated with the labeled *Fn* according to the co-culture protocol described above. After the assay, the cells were fixed with a 4% formaldehyde solution for 15 min at room temperature (RT), followed by two PBS wash steps. Fixed cells were blocked with 1% BSA (1 h) and further incubated with a pan-Fusobacterium membrane antiserum (Casasanta et al, 2020) (DJSVT_MAS1, a kind gift from Prof. Dr. Daniel Slade, VirginiaTech) to label extracellular bacteria used at a 1:40 dilution overnight at 4 °C. After two washing steps with PBS, the pan-Fusobacterium membrane antiserum was labeled with the secondary antibody (Alexa Fluor 594 goat anti-rabbit antibody) for 1 h at RT. Host-cell nuclei were stained with 4',6-diamidino-2-phenylindole (DAPI) for 15 min at RT in PBS. Immunofluorescence images were taken with an Olympus IX83 inverted microscope.

## Binding assay on human CRC tissue

*Fn* binding assay was performed according to a previously described protocol (Abed et al, 2016). Briefly, patient tissue

sections were deparaffinized and hydrated, washed in TBS and then blocked in TBS supplemented with 20% BSA, 20% FBS and 5% Triton X-100 for 7 h at RT. The slides were subsequently incubated with labeled Cy-5-Fusobacteria ($3 \times 10^7$ bacteria/ml in blocking solution) overnight at 4 °C. The next day, slides were washed once in PBS with 0.5% Tween followed by two PBS washing steps. Slides were counterstained with DAPI (1:1000, stock 1 mg/mL) diluted in PBS for 30 min, washed in PBS and then mounted with Fluoromount mounting media. Images were acquired at the Cytation 10 confocal microscope.

## Immunofluorescent analysis of paraffin-embedded mouse colons

Colons were rolled lengthwise and fixed in 10% formalin for 24 h, then embedded in paraffin. For immunostaining, tissue sections were deparaffinized and rehydrated sequentially in xylene ($2 \times 5$ min), 100% ethanol (3 min), 96% ethanol (3 min), 70% ethanol (3 min), and finally distilled water (5 min). Antigen retrieval was performed by incubating slides in 1× Citrate Buffer (Abcam, ab93678), pH 6.0, at 93 °C for 15 min. After two washes in distilled water, the sections were blocked with blocking buffer (5% BSA, 5% Horse serum and 0.3% Triton X-100) for 30 min at RT, followed by overnight incubation at 4 °C with primary antibodies. The anti-OMP-antibody (Strauss et al, 2011), generously provided by Prof. Emma Allen-Vercoe, was used at a 1:750 dilution to detect *Fn*. Simultaneously, an anti-podoplanin (PDPN) antibody was applied at a 1:200 dilution to label the stromal compartment (Invitrogen, 14-5381-82). After incubation, slides were washed twice with PBS containing 0.5% Tween, followed by one wash in PBS. Secondary antibodies were applied as follows: Alexa Fluor 647 donkey anti-rabbit (1:1000; Invitrogen, A31573) and Alexa Fluor 488 anti-hamster (1:1000; Invitrogen, A78958), and nuclei were counterstained with DAPI (1:1000, stock 1 mg/mL) diluted in PBS for 40 min at RT. Slides were then washed once with PBS containing 0.5% Tween and twice with PBS before being mounted using Fluoromount mounting medium. Images were captured using the Cytation 10 confocal microscope.

## In situ hybridization staining of human CRC sample

Human CRC samples were fixed overnight at 4 °C with neutral buffered formalin 10%. Paraffin-embedded tissue sections (2–3 μm) were air-dried and further dried at 60 °C for 2 h. All stainings were performed using a Ventana Discovery Ultra Kit. For ISH, ready-to-use (RTU) reagents from RNAscope® VS Universal AP Detection Reagents (ACD bio-techne, 323260) were loaded onto the DISCOVERY mRNA RED Probe Amplification Kit (Roche, 760-236,) or the DISCOVERY mRNA Sample Prep Kit (Roche, 760-248,) containers according to the user manual (Doc. No. 323250-USM-ULT). FFPE tissue sections were deparaffinized, and antigen retrieval was initiated using RNAscope® VS Sample Prep Reagents v2 (ACD bio-techne, 323740), followed by the epitope retrieval Cell Conditioning 1 (CC1) buffer for 24 min at 97 °C (Roche, 6414575001) and protease treatment (16 min at 37 °C). Probe hybridization, signal amplification, colorimetric detection with DISCOVERY mRNA RED Detection kit (Roche, 760-234) and counterstaining were subsequently performed following the manufacturer's recommendations. Hybridization against 23S rRNA

from *Fusobacterium nucleatum* subsp. *animalis* was performed with the RNAscope® 2.5 VS Probe - B-Fusobacterium-23S-3zz (ACD bio-techne, 486419). Thereafter, a sequential immunohistochemistry was performed by adding the primary mouse monoclonal IgG1 anti-PDPN [clone D2-40] antibody (Roche, 05463645001) RTU, for 60 min at 37 °C; signal was detected using the OptiView DAB IHC Detection Kit (Roche, 06396500001); and subsequently, adding the primary mouse monoclonal IgG/k anti-Actin, Smooth Muscle [clone 1A4] antibody (Roche, 05268303001) RTU, incubated for 40 min at 37 °C; signal was detected using with the DISCOVERY UltraMap anti-Ms HRP (Roche, 760-4313) and chromogenic detection was done with the DISCOVERY Purple kit (Roche, 7053983001). Sections were counterstained with hematoxylin II (Roche, 790-2208) and mounted with Pertex (MEDITE, 41-4011-00) using a MEDITE RCM 9000 coverslipping machine. Brightfield images were acquired with a NanoZoomer S360 digital scanner (Hamamatsu) equipped with a ×40 objective. All images were visualized with a gamma correction set at 1.8 in the image control panel of the NDP.view 2 U12388-01 software (Hamamatsu, Photonics, France).

## Flow cytometry of *Fn* binding and invasion

Before co-culture, bacteria were pre-labeled with CFSE (CellTrace™ CFSE Cell Proliferation Kit, for flow cytometry from Invitrogen, C34554) for 20 min on a shaker in PBS at RT. After 4 h of co-culture with CFSE-labeled *Fn*, cells were washed with PBS, trypsinized (0.05%) and centrifuged at 350 rcf for three minutes at RT. Cells were then stained with LIVE/DEAD™ Fixable Near-IR Dead Cell stain (Invitrogen, L34975) for 20 min at 37 °C, further washed one time with FACS buffer (DPBS supplemented with 2% FBS and 2 mM EDTA), and resuspended in 150 μl of FACS buffer. The fluorescence was measured by flow cytometry, and data analysis was performed using FlowJo v10.10.0.

## Flow cytometry analysis

Cells were dissociated into a single-cell suspension using trypsin-EDTA (1 mM EDTA, Life Technologies, 25300062), resuspended in FACS buffer and incubated with the fluorescent-labeled antibodies FITC-anti-Lamin A/C (Santa Cruz Biotechnology, SC-376248), APC anti-αSMA (Bio-Techne, IC420A), PerCP-Cy-5.5-anti-PGDFRα (Bio-Techne, FAB1264P), BV421-anti-PGDFRβ (BD Biosc., USA564124), PE-anti-CXCL1 (Bio-Techne, IC275P), AF488 anti-αSMA (eBioscience, 53-9760-82), APC anti-FAP (Bio-Techne, FAB3715A) or PE-Cy7 anti-PDPN (Biolegend, 337014). Intracellular proteins were labeled following an additional fixation and permeabilization step (Cytofix/Cytoperm Solution Kit, BD Bioscience). Intracellular and mitochondrial reactive oxygen species (ROS) were detected with 2',7'-dichlorodihydrofluorescein diacetate ($H_2DCFDA$, Sigma, D6883) and MitoSOX (Sigma, M36008), respectively, incubated in DMEM-F12 for 30 min at 37 °C. Dead cells were stained prior to fixation using LIVE/DEAD® Fixable Near-IR Dead Cell stain (Invitrogen) and subsequently excluded from the analysis. Once stained, cells were washed twice with FACS buffer, resuspended in 150 μl of FACS buffer and the fluorescence measured using a BD FACSCanto II or BD LSRFortessa flow cytometer. Results were further analyzed using

FlowJo v10.10.0 and figures rendered in *R* using the packages ggcyto (version 1.36.0), flowCore (version 2.20.0) and CytoML (version 2.20.0).

Accordingly, for the Annexin V assay, cells were stained according to the manufacturer's protocol (BD Biosc, 556422), with the sole modification being the inclusion of DAPI at a 1:1000 dilution.

## Transmission electron microscopy (TEM)

In total, 50,000 CT5.3 CAF cells were seeded into an eight-well chamber μ-Slide (Ibidi, 80826). CAFs were then treated for 4 h at an MOI 50 with *Fn* 71. After incubation, the cells were fixed with 2.5% glutaraldehyde containing 0.1 M sodium cacodylate and subsequently washed three times with cacodylate buffer. Samples were post-stained with osmium tetroxyde, then dehydrated, embedded in epoxy resin, and sectioned at a thickness of 60 nm. Imaging was performed at 30 kV using a Zeiss Gemini scanning transmission electron microscope (STEM).

## Intracellular bacteria viability imaging

In total, 20,000 CT5.3 CAF cells were seeded per well in eight-well chamber μ-Slide (Ibidi, 80826). CAFs were then treated for 4 h at an MOI 50 with *Fn* 71. To remove extracellular bacteria, the cells were washed thrice for 20 min each with 2% P/S containing medium. Cells were then washed three times with DPBS and stained with anti-OMP antibody (1:750 dilution) for 1 h at 4 °C. After three more washes in DPBS, cells were incubated for 1 h with AF488-conjugated goat anti-rabbit secondary antibody (1:1000, Abcam, ab150077). Cells were washed three times with DPBS and selectively permeabilized for 5 min with a 0.1% saponin solution in DPBS, then incubated for 30 min in a 0.1% saponin solution containing 30 μM propidium iodide and 1 μg/ml Hoechst (33342) at RT. Finally, the cells were washed twice in DPBS and imaged on a Cytation 10 Imaging reader (Biotek).

## Podoplanin in vitro CAF staining

In total, 20,000 CT5.3, CAF180 or CAF181 CAFs were seeded per well in an eight-well chamber μ-Slide (Ibidi, 80826). CAFs were pretreated for 1 h with 1 μM TAK-242 or appropriate DMSO vehicle control, then infected for 4 h at an MOI 50 with *Fn* 71. To remove extracellular bacteria, cells were then washed thrice for 20 min each with 2% P/S containing medium. The cells were cultured a further 24 h in DMEM-F12 supplemented with 10% FBS and 1% P/S in the presence of 1 μM TAK-242 or appropriate DMSO vehicle control. The CAFs were then washed twice using FACS buffer and stained with 1:100 dilution of APC-conjugated anti-PDPN antibody (clone NC-08, Biolegend) for 1 h at 4 °C. Cells were then washed thrice in FACS buffer and fixed for 15 min at RT using 2% PFA. CAFs were washed twice and stained with 1 μg/ml Hoechst (33342) in DPBS, followed by two more DPBS washes. Finally, the cells were imaged using a Cytation 10 Imaging reader (Biotek). For image analysis, automated cell detection was performed on at least three 20× fields using QuPath software (Bankhead et al, 2017) and the mean cytoplasmic PDPN fluorescence intensity per cell was plotted.

## Transcriptome analysis

### RNA extraction and quality control

RNA from cell cultures was isolated using the miRNeasy Mini Kit (Qiagen) according to the manufacturer's protocol. RNA concentration was determined at an absorbance of 260 nm using a NanoDrop 2000c Spectrophotometer (Thermo Fisher Scientific) and its purity assessed from the 260/280 nm and 260/230 nm absorbance ratios. RNA samples were desalted by ethanol precipitation and the RNA integrity number (RIN) was calculated using the Agilent 21000 Bioanalyzer platform (Agilent Technologies) with the Agilent RNA 6000 Nano Kit. Only samples with a RIN ≥ 9 were subsequently used for RNA sequencing.

### RNA sequencing and data analysis

RNA libraries were sequenced as 100 bp paired-end runs on a HiSeq2500 platform (Illumina). Differential expression analysis was performed using the statistical software R (version 4.4.0) and DESeq2 (version 1.44.0), and the R package ashr (version 2.2.63) (Stephens, 2017) was used to perform $\log_2$ fold change shrinkage. Genes with a fold change higher than 1.5 and adjusted p value < 0.05 were considered as significantly differentially expressed. Median ratio-normalized (MRN) count values were obtained using the "counts" method implemented in DESeq2 and the "normalized" argument set to "TRUE".

### cDNA synthesis and real-time (RT) qPCR

cDNA was synthesized using the high-capacity cDNA reverse transcription kit (Applied Biosystems) following the manufacturer's protocol. The qPCR reaction was prepared using the Absolute Blue qPCR SYBR Green Low ROX Mix (Thermo Scientific) with 5 ng of cDNA and 2.5 pmol of the forward and reverse primers. The PCR amplification was performed over 40 cycles (95 °C for 30 s, 60 °C for 30 s and 72 °C for 30 s) on a 7500 FAST Real-time PCR Detection System (Applied Biosystems). Gene expression levels (Ct) were then normalized against the geometric mean of the house-keeping genes HPRT1 and YWHAZ using qBase+ (Biogazelle) based on the MIQE guidelines. Primer sequences can be found in Appendix Table S4.

## Cytokine array assay (dot blot)

CAFs were treated with or without bacteria for 2 h and, following washing steps, further incubated for 24 h. On the following day, media from non- and bacteria-treated CAFs were collected, and cellular debris were removed by centrifugation at 350 rcf for five minutes at 4 °C. Cytokine arrays were performed using the Proteome Profiler Human XL Cytokine Array Kit (R&D Systems, ARY022B). A Fusion FX imaging platform was used to detect the signals from the dot blot, and the signal densities were further quantified using the ImageJ 1.52p software.

## ELISA

The concentrations of CXCL1, IL-6, and IL-8 in cell culture supernatants were measured using the Human CXCL1 DuoSet ELISA (R&D Systems, #DY275), Human IL-6 DuoSet ELISA (R&D Systems, #DY206) and Human IL-8/CXCL8 DuoSet ELISA (R&D Systems, #DY208) kits according to the manufacturer's protocol.

Cytation 5 cell imaging multi-mode reader (Biotek) was used to measure the absorbances at 450 nm and 570 nm.

## Migration and invasion assay

In total, 50,000 HCT116 were seeded on the surface of a Transwell insert (8 µm pore size, Greiner) filled with DMEM-F12 containing 1% FBS, while 80,000 CAFs were cultured in 24-well plates in DMEM-F12 containing 10% FBS. Fibroblasts were further treated with bacteria for 2 h and extensively washed with P/S containing medium to remove residual bacteria. The co-culture was performed by placing the Transwell insert on top of the fibroblasts. To evaluate the migration capacity of HCT116 cells, fibroblasts (either untreated or treated with Fn) were cultivated in DMEM-F12 with 10% FBS, while HCT116 cells were maintained in DMEM-F12 with 1% FBS. After 24 h, tumor cells on the membranes were fixed with a 4% formaldehyde solution and stained using crystal violet (0.05%). Cells that did not migrate (located on the upper/inner part of the insert) were gently removed with a cotton swab. Images of the migrated cells were captured using a brightfield microscope, and the number of migrating cells was quantified using MATLAB.

For the SW480 transwell invasion assays, the Transwell inserts (8 µm pore size, Greiner) were first pre-coated with a 0.05 mg/ml Collagen Type I (Sigma-Aldrich) and 0.5 mg/ml of ECM gel (Sigma-Aldrich) in a 1:1 PBS-Culture medium mixture for 2 h at 37 °C. Then 80,000 cells were seeded on top and the same protocol as above applied.

## 3D collagen invasion assay

HCT116-mCherry and CT5.3-GFP cells were co-cultured in ultra-low attachment BIOFLOAT™ 96-well plates (Facellitate, Mannheim, Germany) at densities of $1 \times 10^3$ and $0.5 \times 10^3$ cells, respectively, maintaining a 2:1 ratio in DMEM-F12 supplemented with 10% FBS and 1 U/ml P/S. After 72 h of spheroid formation on a shaking platform, the spheroids were embedded in a collagen solution composed of 2 mg/ml collagen type I (MerckMillipore, Darmstadt, Germany), 1× DMEM-F12 (Gibco, Thermo Fisher Scientific, Waltham, MA, USA), and 1% FBS (Gibco, Thermo Fisher Scientific, Waltham, MA, USA). The pH of the collagen solution was adjusted to 7.4 using 1 M NaOH. The plates were incubated at 37 °C for 30 min to allow polymerization. The collagen matrix was then overlaid with 200 µl of DMEM-F12 supplemented with 10% FBS and 20% conditioned medium collected from CT5.3 cells grown alone or in co-culture with either Ec or Fn 71. To prevent evaporation, plates were sealed with ibiseals (ibidi GmbH, Planegg, Germany). Images were captured daily from day 0 (immediately after embedding) until 6 using a Cytation 10/Biospa system (BioTek, Winooski, VT) at ×4 magnification with z-stacking. To assess invasion, the area of the spheroid core was subtracted from the total area (spheroid core + invasive front). Spheroids exhibiting invasive areas at day 0 (before treatment onset) outside the Tukey's fences were considered as outliers and discarded (i.e. area < Q1 − 1.5 × IQR or area > Q3 + 1.5 × IQR, where Q1 and Q3 refer to the first and third quartiles and IQR refers to the interquartile range). Invasion areas were further scaled by experiment (non-centered scaling using the scale function implemented in R) and divided by the average value at day 0 to reflect an area fold increase relative to day 0.

## Stable isotope tracing and metabolite extraction

Stable isotope tracing experiments with [U-$^{13}$C]glutamine (Cambridge Isotope Laboratories, CLM-1822) were performed in customized DMEM-F12 (Thermo Fisher Scientific, lot No. GI240129031). The customized medium was supplemented with the missing metabolites (glucose, tryptophan, serine, glycine, hypoxanthine, aspartate and pyruvate) according to DMEM-F12 formula alongside 10% dialyzed FBS, 1% PS and 2.5 mM of [U-$^{13}$C]glutamine. CT5.3 cells were seeded at 75,000 cells/well in a 24-well plate (in technical triplicates) and co-incubated with $Fn$ 25586, $Fn$ 71 and $Ec$ at a MOI of 50 or control PBS. After 4 h, CT5.3 were washed thrice with DMEM-F12 + 10% FBS + 1% PS for 20 min each, to eliminate any remaining bacteria. Next, we added medium containing [U-$^{13}$C]glutamine for 24 h. At 24 h, cells were washed twice with ice-cold PBS and intracellular metabolites were extracted using a mixture of methanol (Carl Roth, KK44.1):acetonitrile (Carl Roth, AE70.1):water (5:3:2 ratio) and measured by LC-MS as previously described in (Benzarti et al, 2024). Data were analyzed using TraceFinder (Thermo Fisher Scientific, Version 4.1). Natural isotope abundance was corrected for using AutoPlotter, and mass isotopomer distribution was determined and plotted (Pietzke and Vazquez, 2020).

## Dataset analyses

The colon (COAD) and rectal (READ) cases of the Cancer Genome Atlas (TCGA, data release version 36.0) were used. STAR counts for protein-coding genes were extracted from the GDC portal (https://gdc.cancer.gov) using the GenomicDataCommons $R$ package and normalized using DESeq2 (median of ratio normalization or MRN). FFPE or FFPE validation samples (according to the biospecimen aliquot information) were discarded ensuring that each of the 624 CRC patients was linked to a single sample. Gene signature scores were calculated using $R$ package singscore (version 1.28.0; Foroutan et al, 2018). Hallmark gene sets were obtained from MSigDB, and the myCAF/iCAF marker genes were selected from the gene sets described in (Elyada et al, 2019; Chen et al, 2023).

The tumor fibroblasts of the GSE132465 and GSE144735 CRC scRNA-Seq datasets (Lee et al, 2020) were previously integrated and annotated(Koncina et al, 2023). Signature scores were calculated using the R packages Seurat (version 5.3.0) and UCell (version 2.13.1; Andreatta and Carmona, 2021).

The CSI-Microbes analysis of the Pelka2021 dataset (Pelka et al, 2021; Robinson et al, 2024) was obtained from the Zenodo link published in (Robinson et al, 2024).

## PathSeq analysis

PathSeq analysis of TCGA samples to define $Fn$ load was performed as previously described (Ternes et al, 2022; Kostic et al, 2011). RNA-Seq raw files were analyzed and PathSeq scores obtained from two identified TCGA sequencing workflows. The scores were further batch-corrected using the ComBat algorithm implemented in the $R$ package sva (Leek et al, 2012; Johnson et al, 2007). The lowest $Fn$ score of the less sensitive batch was defined as the minimum threshold by further subtracting its value from the corrected scores and setting the remaining negative scores to 0.

## In vivo experiments

Animal experiments have been carried out under the applicable laws and regulations approved by Animal Experimentation Ethics Committee of University of Luxembourg (UL) and the veterinarian service of the Ministry of Agriculture, Viniculture and Rural Development (Luxembourg, LUPA 2020/10 and 2024/14). The care and use of animals for research purposes at UL is performed according to the EU Directive 2010/63/EU, as well as the Grand-Ducal Regulation of 11 January 2013 on the protection of animals used for scientific purposes, including justification of the use of animals, their welfare and the incorporation of the principles of the 3Rs (Replacement, Reduction and Refinement).

In vivo primary CRC model (Grivennikov et al, 2012). Seven to eight-week-old in-house bred male and female germ-free CDX2-CreER$^{T2}$Apc$^{fl/fl}$ (derived from strain #035169, Jackson) were injected intraperitoneally for 5 consecutive days with 75 mg per kg of body weight tamoxifen (Sigma-Aldrich) in corn oil. Mice were then gavaged three times a week orally with 10$^8$ CFU/mouse of $Fn$ 71 or PBS control. After 3 weeks, mice were sacrificed and colons harvested as well as formalin-fixed paraffin-embedded (FFPE) for subsequent analysis.

In vivo tail vein metastasis assay. Seven to eight-week-old in-house bred male and female nod scid gamma (Mus musculus NSG) mice were intravenously injected on the tail with $0.5 \times 10^6$ HT-29 cells in 100 µl volume of PBS for injection. Cancer cells (HT-29) were pretreated with conditioned media (CM) of infected CAFs. To generate the CM, CAFs (CT5.3) were co-cultured with $Ec$ (MG1655, MOI 50) or with $Fn$ 71 (MOI 50) for 4 h. This was followed by washing of the CAFs three times for 20 min with medium and supplementing it with 1% [v/v] penicillin/streptomycin (P/S). The CAF-CM was then collected after a further 24-h incubation of medium with 1% [v/v] P/S. This CAF-CM was then added to the HT-29 cells (100% [v/v]) for 24 h before being injected intravenously. After 30 days, the mice were sacrificed, their lungs were resected, and metastatic nodules were counted blindly using four images from four lobes per mouse based on H&E staining.

NSG mice were housed in AllenTown NexGen Mouse 500 cages ($19.4 \times 13.0 \times 38.1$ cm) with JRS Rehofix Corncob bedding, whereas germ-free CDX2-CreER$^{T2}$Apc$^{fl/fl}$ mice were kept in AllenTown Sentry Sealed Positive Pressure Individually Ventilated (SPP IVC) cages ($19.4 \times 17.8 \times 41.9$ cm). All animals had ad libitum access to food and water and were fed standard irradiated rodent chow (A04, SAFE Lab; 40 kGy). Mice were maintained under standard conditions (humidity: 40–70%, temperature: 22 °C) with a 12-h light/dark cycle.

## Statistical analysis

All statistical analyses as detailed in the figure legends were performed in $R$ using the packages lme4 (v. 1.1.35.4), lmerTest (v. 3.1.3) and rstatix (v. 0.7.2). $P$ values are reported on the figures and in Appendix Table S5.

# Data availability

The RNA-Seq data generated and used in this study have been deposited at the European Genome-phenome Archive (EGA) under the accession

numbers EGAD50000001614 and EGAD00001010322. As required by the European Data Protection Regulation, the data are available under restricted access as it is of human origin. Access can be obtained by contacting the corresponding author at elisabeth.letellier@uni.lu and the procedure is described in https://ega-archive.org/access/data-access.

The source data of this paper are collected in the following database record: biostudies:S-SCDT-10_1038-S44318-025-00542-w.

## Peer review information

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

## Acknowledgements

We thank the patients who kindly donated their samples and made this study possible. We gratefully acknowledge the contributions of the surgeons and nurses (not listed as co-authors) from the Centre Hospitalier Emile Mayrisch and the Hôpitaux Robert Schuman, as well as the teams from the Laboratoire National de Santé and the Clinical and Epidemiological Investigation Centre of the Luxembourg Institute of Health for their dedicated work with the patients. We also thank Prof. Dr. Michel Mittelbronn and the pathologists and macroscopy team from NCP/LNS. The authors would also like to thank their collaborators at the IBBL, Dr. Fay Betsou, Dr. Nikolai Goncharenko, Dr. Christelle Bahlawane, Amélie Gaignaux and Lamia Skhiri for the overall setup of the patient sample collection and the management of the cohort. We would also like to thank the whole team of the Animal Facility at the University of Luxembourg, especially our veterinarian Jennifer Behm and our facility manager Djalil Coowar. We would like to acknowledge the metabolomics platform at Luxembourg Institute of Health and specifically Dr. Nathalie Legrave and Francois Bernardin for their help with sample measurements. Finally, we would like to thank Professor Olivier de Wever for the kind gift of the CT5.3 fibroblasts, Prof. Daniel Slade for the donation of the pan-Fusobacterium membrane antiserum, and Prof. Emma Allen-Vercoe and her lab at the University of Guelph for generously providing the OMP *Fn* antibody. Parts of the data processing presented in this manuscript were carried out using the HPC facility of the University of Luxembourg (https://hpc.uni.lu). The graphical abstract as well as all experimental schematics were created with BioRender.com (https://biorender.com). This work was supported by the Luxembourg National Research Fund [CORE/C16/BM/11282028 (EL), CORE/C15/BM/10404093 (PW), PoC/18/12554295 (EL), CORE/C21/BM/15718879 (JM), AFR 17103240 (CG), PRIDE Doctoral Research in the scope of the Doctoral Teaching Unit - MICROH PRIDE17/11823097 to MT, and CANBIO PRIDE21/16763386 (CCTB). The study was further supported by the FNR and the Fondation Cancer Luxembourg grant CORE/C20/BM/14591557 (to EL), an FNR matched funding schemes (MFP20/15251414/MelCol PFP, EL), an Internal Research Project at the University of Luxembourg (MiDiCa; EL, PW, SH) as well as an FNRS-Télévie funding scheme 7.4565.21 and 7.6603.02 to MM, 7.4560.22 to PS and 7.4552.23 to MG, by the Fondation du Pélican de Mie and Pierre Hippert-Faber under the aegis of the Fondation de Luxembourg (JK, MM, DT, and MT), the Fondation Schumacher (EL), a postdoctoral fellowship from the Swiss National Science Foundation (P500PB_214405, PPE), a postdoctoral fellowship from the Spanish Fundacion Ramon Areces (JAMT), the Fondation Gustave et Simone Prévot (EL). We would like to thank the Fondation Cancer and the University of Luxembourg for their support of the Luxembourgish CRC patient cohort. JV received relevant funding from Deutsche Forschungsgemeinschaft (DFG; SFB 1583/1 DECIDE, Project number: 492620490, Subproject A09). Moreover, this work was supported by Cancer Research UK [grant number C17937/A29070], Fondo de Investigaciones

Sanitarias (FIS) grant (PI20/00889) from the Instituto de Salud Carlos III (ISCIII) and Fundación Mutua Madrileña (MMADRILEÑA/PREMI/2020CCAA_NUCIFORO). The funders had no role in study design, data collection and analysis, decision to publish, or preparation of the manuscript.

## Author contributions

**Jessica Karta**: Conceptualization; Data curation; Formal analysis; Validation; Investigation; Methodology; Writing—original draft. **Marianne Meyers**: Conceptualization; Data curation; Formal analysis; Validation; Investigation; Methodology; Writing—original draft; Writing—review and editing. **Fabien Rodriguez**: Investigation; Methodology. **Eric Koncina**: Conceptualization; Data curation; Formal analysis; Investigation; Visualization; Methodology; Writing—original draft; Writing—review and editing. **Cedric Gilson**: Formal analysis; Investigation; Methodology. **Eliane Klein**: Investigation; Methodology. **Monica Gabola**: Investigation; Methodology. **Mohaned Benzarti**: Methodology. **Pau Pérez Escriva**: Conceptualization; Investigation; Methodology. **Jose Alberto Molina Tijeras**: Investigation; Methodology. **Catarina Correia Tavares Bernardino**: Investigation; Methodology. **Falk Ponath**: Resources; Methodology. **Anais Carpentier**: Methodology. **Mònica Aguilera Pujabet**: Investigation; Methodology. **Maryse Schmoetten**: Methodology. **Mina Tsenkova**: Investigation; Methodology. **Perla Saoud**: Methodology. **Anthoula Gaigneaux**: Methodology. **Dominik Ternes**: Methodology. **Lidia Alonso**: Methodology. **Nikolaus Zügel**: Resources; Methodology. **Eric Willemssen**: Resources; Methodology. **Philippe Koppes**: Resources; Methodology. **Daniel Léonard**: Resources; Methodology. **Luis Perez Casanova**: Methodology. **Serge Haan**: Methodology. **Michel Mittelbronn**: Methodology. **Johannes Meiser**: Funding acquisition; Methodology. **Vitaly I Pozdeev**: Formal analysis; Investigation; Methodology. **Jörg Vogel**: Resources; Funding acquisition; Methodology. **Paolo G Nuciforo**: Resources; Investigation; Methodology. **Paul Wilmes**: Supervision; Funding acquisition. **Elisabeth Letellier**: Conceptualization; Supervision; Funding acquisition; Investigation; Writing—original draft; Project administration; Writing—review and editing.

Source data underlying figure panels in this paper may have individual authorship assigned. Where available, figure panel/source data authorship is listed in the following database record: biostudies:S-SCDT-10_1038-S44318-025-00542-w.

## Disclosure and competing interests statement

The authors declare no competing interests. Jörg Vogel is an editorial advisory board member.

