## [Peer Review File · The EMBO Journal]

***Fusobacterium nucleatum* interacts with cancer-associated fibroblasts to promote colorectal cancer**

Jessica Karta, Marianne Meyers, Fabien Rodriguez, Eric Koncina, Cedric Gilson, Eliane Klein, Monica Gabola, Mohaned Benzarti, Pau Perez Escriva, Jose Molina Tijeras, Catarina Correia Tavares Bernardino, Falk Ponath, Anaïs Carpentier, Monica Pujabet, Maryse Schmoetten, Mina Tsenkova, Perla Saoud, Anthoula Gaigneaux, Dominik Ternes, Lidia Alonso, Nikolaus Zuegel, Eric Willemsen, Philippe Koppes, Daniel Leonard, Luis Casanova, Serge Haan, Michel Mittelbronn, Johannes Meiser, Vitaly Pozdeev, Jörg Vogel, Paolo Nuciforo, Paul Wilmes, and Elisabeth Letellier

Corresponding author: Elisabeth Letellier (elisabeth.letellier@uni.lu)

Review Timeline:

Submission Date:	5th Jul 24
Editorial Decision:	17th Aug 24
Revision Received:	16th May 25
Editorial Decision:	30th Jun 25
Revision Received:	17th Jul 25
Accepted:	29th Jul 25

Editor: Daniel Klimmeck

Transaction Report:

Dear Dr Letellier,

Thank you again for the submission of your manuscript (EMBOJ-2024-118410) to The EMBO Journal. Please accept my sincere apologies for getting back to you with delay due to protracted referee input and detailed discussion in the editorial team. Your study was assessed by four reviewers with expertise in cancer immunity and microbiology, whose comments are enclosed below.

As you will see from the experts' reports, the referees acknowledge the analysis and potential interest of your results. However, they also express major concerns regarding completeness and in vivo relevance of the findings, which need to be addressed thoroughly to make them supportive of publication in the EMBO Journal. The reviewers also raise a number of issues related to the data presentation, additional controls and improved methods annotation required, statistics applied and overall discussion of related literature, that would need to be conclusively addressed to achieve the level of robustness and clarity needed for The EMBO Journal.

Given the overall interest stated and broader angle of your findings, we are able to invite you to revise your manuscript experimentally to address the referees' comments. I need to stress though that we do require strong support from the referees on a revised version of the study in order to move on to publication of the work.

In light of the extensive experimentation requested, I would appreciate if you could contact me during the next weeks for exchange e.g. a video call to discuss your perspective on the comments and potential plan for revisions.

Please feel free to contact me if you have any questions or need further input on the referee comments.

When submitting your revised manuscript, please carefully review the instructions below.

Please feel free to approach me any time should you have additional questions related to this.

Thank you for the opportunity to consider your work for publication.

I look forward to your revision.

Kind regards,

Daniel Klimmeck

Daniel Klimmeck, PhD
Senior Editor
The EMBO Journal

Instruction for the preparation of your revised manuscript:

- 1) a .docx formatted version of the manuscript text (including legends for main figures, EV figures and tables). Please make sure that the changes are highlighted to be clearly visible.
- 2) individual production quality figure files as .eps, .tif, .jpg (one file per figure).
- 3) a .docx formatted letter INCLUDING the reviewers' reports and your detailed point-by-point response to their comments. As part of the EMBO Press transparent editorial process, the point-by-point response is part of the Review Process File (RPF), which will be published alongside your paper.
- 4) a complete author checklist, which you can download from our author guidelines (<https://wol-prod-cdn.literatumonline.com/pb->

assets/embo-site/Author Checklist%20-%20EMBO%20J-1561436015657.xlsx). Please insert information in the checklist that is also reflected in the manuscript. The completed author checklist will also be part of the RPF.

6) It is mandatory to include a 'Data Availability' section after the Materials and Methods. Before submitting your revision, primary datasets produced in this study need to be deposited in an appropriate public database, and the accession numbers and database listed under 'Data Availability'. Please remember to provide a reviewer password if the datasets are not yet public (see <https://www.embopress.org/page/journal/14602075/authorguide#datadeposition>).

7) Our journal encourages inclusion of *data citations in the reference list* to directly cite datasets that were re-used and obtained from public databases. Data citations in the article text are distinct from normal bibliographical citations and should directly link to the database records from which the data can be accessed. In the main text, data citations are formatted as follows: "Data ref: Smith et al, 2001" or "Data ref: NCBI Sequence Read Archive PRJNA342805, 2017". In the Reference list, data citations must be labeled with "[DATASET]". A data reference must provide the database name, accession number/identifiers and a resolvable link to the landing page from which the data can be accessed at the end of the reference. Further instructions are available at .

8) At EMBO Press we ask authors to provide source data for the main and EV figures. Our source data coordinator will contact you to discuss which figure panels we would need source data for and will also provide you with helpful tips on how to upload and organize the files.

Numerical data can be provided as individual .xls or .csv files (including a tab describing the data). For 'blots' or microscopy, uncropped images should be submitted (using a zip archive or a single pdf per main figure if multiple images need to be supplied for one panel). Additional information on source data and instruction on how to label the files are available at .

9) We replaced Supplementary Information with Expanded View (EV) Figures and Tables that are collapsible/expandable online (see examples in <https://www.embopress.org/doi/10.15252/emj.201695874>). A maximum of 5 EV Figures can be typeset. EV Figures should be cited as 'Figure EV1, Figure EV2' etc. in the text and their respective legends should be included in the main text after the legends of regular figures.

11) For data quantification: please specify the name of the statistical test used to generate error bars and P values, the number (n) of independent experiments (specify technical or biological replicates) underlying each data point and the test used to calculate p-values in each figure legend. The figure legends should contain a basic description of n, P and the test applied. Graphs must include a description of the bars and the error bars (s.d., s.e.m.).

The revision must be submitted online within 90 days; please click on the link below to submit the revision online before 15th Nov 2024.

Referee #1:

The manuscript shows how the CRC-associated bacteria Fn interact with cancer-associated fibroblasts (CAFs) in the tumor microenvironment (TME). Although these findings suggest that targeting the crosstalk between CAFs and the microbiome may be a promising new therapeutic avenue for CRC, the work did not give us more information about Fn and fibroblasts. And there are several other concerns.

1. To prove that Fn promotes tumor metastasis by affecting cancer-associated fibroblasts CAF in animal experiments, the subcutaneous tumor model is too simple, and the metastasis animal model needs to be addressed in order to prove that it promotes tumor metastasis.
2. As Fig1D shows Fna 71 were found adhering to the cell surface or residing in the cytoplasm using co-cultured experiment. The author should show the location of Fn in vivo.
3. How about the viability of Fn after they interact with CAF? And How Fn invade CAFs? Is there any specificity?
4. The author should do more experiments, like CCK8\Annexin V-FITC apoptosis detection, to validate the function of Fn on CAF cell viability.
5. As Fig2 shows Fn's adhesin Fap2 binds to CAFs and most likely to their Gal-GalNAc motifs contributing to Fn's binding and invasion capacity. The author should knockout Fap2 to validate the function of Fap2 on Fn's binding and invasion capacity.
6. The author should provide flowchart circle gate strategy (iCAF cell surface labeling).
7. As Fig3 and Fig4 shows Fn polarizes fibroblasts to inflammatory CAFs only by flow chart. The expression of surface membrane proteins on its surface should be proved by WB/IF/IHC, etc.
8. As Fig6 shows this Fn-elicited effect was CAF-specific as the author did not observe a similar effect upon coinjection with Fn-treated NFs. Why Fn elicits an CAF-specific effector?
9. Is there any possible Fn Fap2 more likely to bind to CAF surface rather than tumor cell surface? The author should do competition experiments between CAF and tumor cell binding Fn?
10. The author should also validate the tumor model in vivo by using co-cultured experiment within Fap2KO Fn and CAF.
11. The presence of Fn increases IL6 expression and decreases ACTA2 expression, suggesting that Fn induces iCAF. The author should do more experiments (ELSA\WB, etc.) to prove this.
12. As Fn induced a higher increase in mitochondrial ROS levels in CAFs. How about apoptosis in CAFs induced by Fn?
13. As supplementary fig5B-C showed that Fn 25586 was a more potent CXCL1 inducer than Fn 23726, why?
14. The author should pay more attention on quality of Figures, like Fig6C. Several annotations are not clear to read.

Referee #2:

Karta et al "Fusobacterium nucleatum binds to and invades cancer-associated fibroblasts to enhance colorectal cancer progression"

Authors study how microbes in CRC may interact with CAFs. This is relatively new area of research as previously people focused on cancer and immune cells. Authors specifically look at Fusobacterium (Fn)-CAF interactions and what may be an effect on CRC progression.

They find that that Fn binds to and invades CAFs, just like they do with cancer cells or macrophages. This leads to CAF secretion of pro-inflammatory cytokines and membrane-associated proteases. CAF which have Fn exhibit metabolic shift with an increases in ROS from mitochondria and reduction in methylation substrate availability.

Authors utilize a co-culture model and find that Fn-treated CAFs induce stronger migration of cancer cells. They further find a possible role for CXCL1 (known to be induced by Fn from previous studies). Analysis of human samples shows that high loads of Fn lead to CAFs to be mostly of inflammatory phenotype (iCAF), meanwhile low Fn situation leads primarily to the presence of myofibroblastic CAFs (myCAF).

Overall this is interesting study with most of the experiments well performed, controlled and interpreted. Data on Gal-GalNAc motifs and mode of binding to CAFs is interesting. Few additional experiments, controls and additional explanations are probably required.

- 1) Authors need to more explicitly describe properties/genetic assignment of their Fn strains in view of different clade papers for Fn in CRC by Bullman/Johnson group Nature 2024- which ones are used here?
- 2) For Fig2, it seems to be essential mechanistically to try to disrupt at least some host genes involved into binding and intracellular experiments, in addition to presented experiments with GalNAc treatment
- 3) Fig 3- better characterization of iCAF (or myCAF) fibroblasts with regard to the effect of Fn is needed, past only a few markers such as IL6 etc.
- 4) Fig4 TCGA data on ROS pathways in total tumor are only correlational that these changes happen in CAF and not in cancer cells
- 5) Data on Fig 6 should be reproduce with another bacteria (E.coli). there are publications that bacteria (or even LPS) increase tumor growth. As presented, it does not really illuminate specific and unique role of Fn.
- 6) Mitochondrial ROS data/reprogramming in CAF may be interesting, but what is the functional role?
- 7) Authors imply altered migration but show increased tumor growth in vivo. They should analyze invasion and metastasis probably, as a direct function of migration, In primary model of tumor growth authors should assess proliferation and cell death in vivo in addition to migration. In in vivo model, it would be interesting to use Fn-infected cancer cells as well. Is Fn infection of CAF much more important than Fn infection of cancer cells. Is there an in vivo situation where CAFs are infected but cancer cells are not? Is it enough to infect cancer cells for the "full phenotype"?

Referee #3:

Summary:

Karta et al explore how *Fusobacterium nucleatum* (Fn) affects cancer-associated fibroblasts (CAFs) and the functional consequences for tumour cells. They find that Fn binds to and invades CAFs, leading them to secrete pro-inflammatory cytokines and membrane-associated proteases. They identify CXCL1 as a key cytokine in this process. Using patient data they show that CAFs from patients with high loads of Fn were different to those with low loads. This is an interesting study on an important area, but some additional work is needed before publication.

Major points:

- In Figure 2C binding and invasion seems only minimally reduced with the Fap2 mutant Fn. Could the authors knock down genes linked to N- and O-glycan biosynthesis in the CAFs to see if this reduces Fn binding.
- The findings with respect to mitochondrial ROS are interesting. To properly assess increased activity of the TCA cycle and oxidative phosphorylation it would be ideal to perform either Seahorse Extracellular Flux Analysis or Stable Isotope Tracer Analysis (with ¹³C labelled glucose and glutamine, as total TCA intermediate levels are not very informative in this context.
- The authors state that: "It has previously been reported that cellular demethylases are particularly sensitive to the intracellular levels of the TCA cycle metabolite alpha- ketoglutarate." However, aKG does not vary in their experiments although the S-adenosyl methionine to S-adenosyl homocysteine ratio is reduced. If the authors wish to include these data in the manuscript I feel like more work needs to be done in this regard. Looking at the downstream consequences and methylation patterns for instance.
- In Figure 5, to conclude that reprogrammed CAFs promote the migration of tumor cells, at least one additional CRC line should be used.

Minor points:

- Could the authors explain their choice of control bacteria.
- Figure 1D needs a scale bar.
- Could arrows be added to Figure 1D to indicate what the authors describe?
- How the RNA-seq data set in Figure 2A was generated needs more explanation in the results, including what gene expression is relative to.
- Figure 3A is very confusing, I think this needs to be presented and described much more clearly.
- Legend in Figure 6B is confusing.

Referee #4:

Summary

While the interaction between Fn and tumour cells has been extensively documented in the literature, little is known about the interaction of the bacteria with the tumour microenvironment (TME) and its potential role in the disease.

In this article, the authors document a direct interaction between *Fusobacterium nucleatum* (Fn) and cancer-associated fibroblasts (CAFs) via the bacterial outer membrane protein Fap2 and tumour cell Gal-GalNAc, activating an inflammatory phenotype in the CAFs and elevated ROS levels. Using co-culture assays, evidence is presented for Fn-treated CAF's being able to promote CRC cell-line migration and in vivo tumour growth via enhanced production of pro-inflammatory cytokines such as CXCL1. A correlation between higher Fn loads and elevated inflammatory CAFs and ROS levels in CRC patients is

presented. Overall, the findings are novel and potentially reveal novel opportunities for therapeutic intervention. However, the study is overly reliant on in vitro co-culture models and assays incorporating very limited numbers of CRC cell lines, for which the selection rationale is lacking.

Major critique

Although these findings are potentially highly novel and clinically relevant, there is an over-reliance on relatively simple in vitro models with limited in vivo validation of major findings. Whilst the simple co-culture models employed suffice to document potential interactions and their functional consequences, the major findings must be confirmed in more complex culture (for example human CRC organoid/CAF co-culture) and in vivo CRC models (for example infection of mouse CRC models) to ensure that the documented findings are relevant in more physiologically accurate settings. It should also be possible to document interactions between Fn and CRC stromal populations in CRC tissues via IHC/IF. It is unclear to me why HCT116 was chosen for the tumour migration studies and whether the results are broadly applicable to different sub-types of CRC cell-lines/organoids. Similarly, why was a different CRC cell-line, HT29 chosen for the co-injection studies in vivo? Again, is this result reproduced with other cell-lines representing different CRC sub-types and following orthotopic transplantation (to achieve better physiological relevance)?

The figures were generally challenging to interpret, requiring constant reference to text, legends and materials and methods. Inclusion of illustrative cartoons in Figs 5 and 6 greatly enhanced interpretation and I would encourage the same to be done for other figures.

It was not immediately apparent what the connection is between binding and invasion of Fn in CAFs and the effect of Fn in converting CAFs into iCAF to enhance cell migration. In figure 3, authors should show that there is no increase of iCAF marker expression when they expose Fap2- Fn to CAFs. Following this idea, in figures 5 and 6 they should show the effect of disrupting this interaction on cancer migration and proliferation.

Finally, p values should be indicated by its actual number and not stars. A major proportion of the data do not appear to be statistically significant, reducing overall confidence in the accuracy of some of the major conclusions. For some experiments, the n number is low and doesn't match or exceed the statistical consensus of a minimum of 3 repetitions.

Minor critique

In figure 1A, authors should include immunofluorescence with E. coli control to show that there is no binding to and invasion of the CAFs.

I find figure 3A to be particularly challenging to interpret, lacking requisite annotations. Again, a cartoon would help to clarify what has been done and how. Figure 3B-3D should include a Fap2- Fn mutant and/or a GalNAc treated CAF controls. In figure 3B, how do you explain the decrease of PDGFR expression upon E. coli exposure and PDGFR decrease trend for both Fn and E. coli?

In figure 5A-5C, the authors should show what happens when you treat CAFs with E. coli. In figure 5D-5F, Fap2- and GalNAc treatments should be added.

In figure 6, FAP2-, GalNAc and E. coli treatment should be added. The authors conclude a role for CXCL1 in migration while the results in this figure clearly show an effect on growth. This growth effect has been documented in the literature. When using a blocking antibody against CXCL1, growth is inhibited and IL6 expression is decreased (Miyake M, Furuya H, Onishi S, Hokutan K, Anai S, Chan O, Shi S, Fujimoto K, Goodison S, Cai W, Rosser CJ. Monoclonal Antibody against CXCL1 (HL2401) as a Novel Agent in Suppressing IL6 Expression and Tumoral Growth. *Theranostics*. 2019 Jan 25;9(3):853-867).

In figure 7, the authors do not discuss about the effect of E. coli treatment on CXCL1. Does it have an effect on HCT116 migration? What happens if you block CXCL1 activity in this case?

Following my previous comment on statistics, I highly doubt the statistical significance of these data, especially figure 7D. Again, please add the actual value of the p value instead of stars.

Figure 8: The authors shouldn't ignore the proliferative effect they observe in figure 6. Fn treatment has clearly an effect on proliferation and migration.

We would like to thank the reviewers and the editor for their valuable comments. Below, you will find a point-by-point response to all the comments.

Reviewer #1:

The manuscript shows how the CRC-associated bacteria *Fn* interact with cancer-associated fibroblasts (CAFs) in the tumor microenvironment (TME). Although these findings suggest that targeting the crosstalk between CAFs and the microbiome may be a promising new therapeutic avenue for CRC, the work did not give us more information about *Fn* and fibroblasts. And there are several other concerns.

1.To prove that *Fn* promotes tumor metastasis by affecting cancer-associated fibroblasts CAF in animal experiments, the subcutaneous tumor model is too simple, and the metastasis animal model needs to be addressed in order to prove that it promotes tumor metastasis.

We thank the reviewer for this comment, and we fully agree. Initially, we performed the subcutaneous mouse model which indeed does not adequately address metastatic dissemination; however, upon the reviewers' comment, we have amended our animal protocol to include a metastasis dissemination assay, namely the tail vein assay.

In our experiment, cancer cells were pre-stimulated with conditioned media from either untreated cancer-associated fibroblasts (CAFs) or CAFs treated with a negative control bacterium (*E. coli* MG1655 (*Ec*)), or *F. nucleatum* (strain *Fn* 71). We observed that cancer cells treated with the conditioned media from CAFs pre-stimulated with *Fn* significantly increased metastasis outgrowth at the secondary site (lung) compared to control CAF conditioned media or conditioned media from CAFs treated with *Ec*. We believe that this experiment better supports the claim that *Fn* exposed CAFs promote tumor cell invasion *in vivo* as commented by the reviewer.

*In the revised manuscript we now replaced the data from our previous subcutaneous in vivo experiment by these new findings which we present in **Figure 6A-C**. We have accordingly updated the text in lines 244–249 of the Results section and lines 612–627 of the Materials and Methods section.*

Figure 6. A. Schematic representation of the *in vivo* experimental setup. SPF NSG mice were tail vein injected with 1×10^6 HT-29 cells, which were pre-treated with CM of CT5.3 CAFs cultured alone or CM of CT5.3 co-cultured with *Fn* 71 or *Ec* (4-hour co-culture, followed by a 24-hour incubation, MOI 50). Lungs were harvested after 30 days ($n = 8$ mice for the Ctrl and *Ec* conditions and $n = 7$ for the *Fn* condition). **B.** Quantification of the relative tumor surface area in percentage per imaged lobe (four lobes were analyzed per mouse). **C.** Representative H&E images of tumors (indicated by black arrows) in the lungs on day 30. CM = conditioned medium. Scale bar = 100 μ m. Statistically significant differences were determined using an ANOVA on a robust linear model fit (using the *lmRob* function in the R package *robust*) followed by Tukey's multiple comparison method (using the R package *multcomp*).

2. As Fig1D shows *Fn* 71 were found adhering to the cell surface or residing in the cytoplasm using co-cultured experiment. The author should show the location of *Fn* in vivo.

We fully agree with the reviewer that this is a very important point. To address this and better establish clinical relevance we have now included several new experiments:

1. First, we re-analyzed a publicly available scRNA-Seq-based CSI-Microbes dataset^{1,2} from CRC tumor samples and identified *Fn* within the fibroblast compartment, further suggesting a potential interaction between *Fn* and CAFs in human CRC patients.

We show this new data in Figure 1A of the revised manuscript and accordingly updated the text in lines 86–89 of the Results section and in lines 589–590 of the Materials and Methods section.

2. Furthermore, we conducted an attachment/binding assay directly on human CRC tissue. Using labeled *Fn*, we observed that *Fn* binds significantly to CRC tissue, particularly stromal cell-rich regions, in contrast to *Ec*, which exhibited minimal binding. These findings further support that *Fn* is present and interacts with the stromal compartment in patient CRC tissue.

This finding has been added in Figure 1F and we have updated the text accordingly in lines 102–107 of the Results section and lines 381–389 of the Materials and Methods section.

3. To directly establish human clinical relevance, we investigated the presence of *Fn* in human CRC tissue using in situ hybridization (ISH), alongside immunostaining for α -SMA, and podoplanin (PDPN), two accepted stromal markers. We observed areas where *Fn* staining (red dots) was in close proximity to α SMA-positive cells (brown) and PDPN (purple)-positive cells, suggesting that *Fn* is indeed present *in vivo* in patients in stroma-rich regions. This new data further underlines the co-localization of *Fn* in the stromal compartment in a human physiologically relevant context.

We have included these new analyses in Figure 1B-C and updated the text in lines 89–93 of the Results section and lines 407–432 of the Materials and Methods section.

4. Lastly, we performed an *in vivo* experiment in which germ-free CDX2-CreER^{T2}Apc^{fl/fl} mice were injected with tamoxifen followed by oral gavage with *Fn* 71, three times a week for a span of three weeks. Using a *Fn* reactive antibody (OMP) and co-staining the FFPE colons of these mice with PDPN, an accepted CAF marker (although applied more cautiously in the manuscript as a stromal marker), we were able to localize *Fn* within the stromal compartment of these mice.

We show this new data in **Figure 1D-E** and updated the text in lines 93–97 of the Results section and lines 606–611 in the Materials and Methods section.

Altogether, this new data indicates that *Fn* is present within the stromal compartment *in vivo*. The revised manuscript mainly addresses this point in **Figure 1**, which supports the notion that *Fn* may influence and reshape the stroma, and therefore also CAFs.

Figure 1. A. Re-analysis of the CSI-Microbe Pelka2021 dataset^{1,2} showing *Fusobacterium* infected cells per mille (%) for each cell type in the tumor. B-C. *In situ* hybridization staining of human CRC tissue with α SMA in brown (B, C), PDPN in purple (C), and *Fn* in pink (B, C; indicated by white arrows) in two independent CRC patients. Scale bar = 50 μ m. D. Schematic representation of the germ free CDX2-CreER^{T2}Apc^{fl/fl} experiment. Briefly, tumorigenesis was induced by i.p. injections of tamoxifen (75 mg per kg of body weight) for five daily injections, followed by three times a week oral gavage with 10⁸ CFU/mouse of *Fn* 71 or PBS control. E. Representative immunofluorescent images showing *Fn* 71 (red, white arrows), stained with an OMP *Fn* specific antibody, colocalized with the stromal marker PDPN (green), and DAPI (grey) in a dysplastic region of the colon from the germ free CDX2-CreER^{T2}Apc^{fl/fl} experiment. Images are representative of two mice from one experiment. Scale bar = 50 μ m. F. Attachment and binding assay of *Fn* 71 and *Ec* (red, indicated by white arrows, DAPI in grey) on human CRC tissue. Epithelium (E) and stromal (S) stromal compartments separated by white dashed line. Images are representative of one patient and two independent experiments. Scale bar = 50 μ m.

3. How about the viability of *Fn* after they interact with CAF? And How *Fn* invade CAFs? Is there any specificity?

We thank the reviewer for this comment. To assess the viability of intracellular *Fn* in CAFs, we performed a differential staining to distinguish extracellular from intracellular bacteria. Prior to selective permeabilization with saponin, which targets cholesterol in mammalian membranes, extracellular *Fn* was labelled using a *Fn* reactive antibody (OMP), obtained during revision from the group of Emma Allen-Vercoe through an MTA with the University of Guelph. Following permeabilization of CAF membranes, we applied Hoechst (membrane-permeable DNA dye) and propidium iodide (PI, membrane impermeable) to discriminate viable from dead intracellular bacteria. This approach enabled us to identify both dead and viable bacteria within CAFs. This result validates the viability of *Fn* post interaction with CAFs.

*We now show this new analysis in the **Supplementary figure 1D** and are referring to it in lines 116–118 of the Results section and in lines 464–474 of the Materials and Methods section.*

Regarding the second part of the question - how *Fn* invades CAFs - we believe our original version of the manuscript demonstrates that CAFs express Gal-GalNAc motifs on their surface, which may represent a potential mechanism for *Fn* interaction. To explore the specificity of this binding, we investigated alternative pathways and found that CAFs also express high levels of cadherins.

*We included this new analysis in **Figure 2C** and updated the Results section in line 126 accordingly.*

Most importantly, using a newly established mutants provided during the revisions of the manuscript by our collaborator, Prof. Jörg Vogel, Director of the Helmholtz Institute for RNA-based Infectious Research in Würzburg, we demonstrated that Fap2 mutants—but not FadA mutants—reduce *Fn* binding affinity to CAFs. These findings suggest that *Fn* binding to CAFs is not primarily mediated by cadherins but rather through the Gal-GalNAc motifs, which are highly expressed in CAFs. To validate the Gal-GalNAc axis, the binding affinity and specificity, we performed a binding/adhesion assay in the presence of N-acetyl-D-galactosamine peptide (GalNAc) and observed a reduction in *Fn* binding capacity (previously shown in the initial submitted manuscript, now moved to **Figure 2E**).

*We present this new data in **Figure 2D** and accordingly updated the text in lines 126–129 of the Results section as well as in lines 349–358 of the Materials and Methods section.*

However, as the binding is only partially reversed by the *Fn* mutant and in the presence of the GalNAc, we acknowledge that these results illustrate only one of the possible binding mechanisms between *Fn* and CAFs. We carefully clarified this throughout the

manuscript (*lines 132–134 in the Results section*), emphasizing that this is a potential, but not exclusive, mechanism of interaction. Similar to several studies describing *Fn* interaction partners on epithelial cells^{3,4}, *Fn* might also bind to and interact with CAFs through distinct mechanisms.

To address the reviewer’s question on how specific the interaction of *Fn* is to CAFs in comparison to other players in the tumor microenvironment, in particular the tumor cells themselves, we tried to reanalyze our flow cytometry data. As CAFs and tumor cells differ in size, the measured binding fluorescence needs to be normalized to the cell size. To this end, and as shown in **Figure A** below (*for reviewing purposes only*), we used the forward scatter (FSC-A) measures of our flow cytometry data as a proxy for cell size. As shown in **Figure B** (*for reviewing purposes only*), *Fn* binding values normalized to FSC-A still support that the binding is higher on CAFs than on epithelial cells. However, as we believe that this approach is prone to inherent technical limitations, we decided to not include it in the revised manuscript and didn’t make definitive claims regarding cell-type specificity of *Fn* binding within the TME. Instead, we decided to provide new evidence able to strengthen the observation that *Fn* can interact with CAFs, a finding that broadens our understanding of its potential cellular targets without asserting preferential interaction over other cell types. Nevertheless, we agree with the reviewer that this is of high interest and needs to be investigated in follow-up projects.

We now addressed this point in lines 298–301 of the Discussion.

Figure for reviewing purposes. Mean fluorescence intensity of the forward scatter (FSC-A) as a proxy for cell size (**A**) as well as MFI normalized to size (**B**) in CT5.3, HCT116 and HT29 flow cytometry binding data.

Supplementary figure 1. D. Viability staining of a representative CT5.3 CAF co-cultured with MOI 50 *Fn* 71 for four hours (n = 1 independent experiment). Live extracellular *Fn* are indicated by the asterisk (*) and are OMP⁺ as well as Hoechst⁺. Dead extracellular *Fn* are indicated by the dashed arrow and are OMP⁺, PI⁺ and Hoechst⁺. Dead intracellular bacteria are indicated by a solid arrow and are Hoechst⁺ and PI⁺. And finally, live intracellular bacteria, indicated by the arrowhead, and are only Hoechst⁺. The dashed line surrounding the pink area encloses the nuclei of the resident CAF. Scale bar= 5 μm.

Figure 2. C. Heatmap showing the expression of cadherin genes in a CAFs and tumor spheroids (T) generated RNA-seq dataset. Columns show the expression assessed in independent experiments of patient-derived (T4) and HT-29 tumor spheroids as well as patient-derived CAFs (n = 9 patients: CAF4, CAF12, CAF16, CAF19, CAF20, CAF22, CAF32, CAF41 and CAF42). Expression values are median ratio normalized counts on a log₂ scale. **D.** Binding and invasion of wild type *Fn* 23726, *Fap2* and *FadA* mutant on CT5.3 cells after a four-hour co-culture (MOI 50) assessed using the CFSE dye by flow cytometry (n = 3 independent experiments). Statistically significant differences were determined using a repeated measure ANOVA followed by Tukey's HSD post-hoc test.

4. The author should do more experiments, like CCK8\Annexin V-FITC apoptosis detection, to validate the function of Fn on CAF cell viability.

We agree with the reviewer and have now performed an Annexin V apoptosis assay of the CAFs after co-culture with the three different *Fn* strains. We did not observe any significant difference in apoptosis between control, *Ec*, or *Fn*-treated CAFs.

We now included this additional analysis in **Supplementary figure 2B** and updated the text in lines 152–153 of the Results section.

Supplementary figure 2. B. Cell viability of CT5.3 CAFs and the patient derived CAF180 and CAF181 CAFs treated with *Fn* 25586, *Fn* 23726, *Fn* 71 or *Ec* for four hours at an MOI of 50, determined by the Annexin IV assay via flow cytometry (n = 3 independent experiments).

5. As Fig2 shows Fn's adhesin Fap2 binds to CAFs and most likely to their Gal-GalNAc motifs contributing to Fn's binding and invasion capacity. The author should knockout Fap2 to validate the function of Fap2 on Fn's binding and invasion capacity.

We fully agree with the reviewer. We have now reached out to Prof. Jörg Vogel, Director of the Helmholtz Institute for RNA-based Infectious Research in Würzburg, to ask for his newly generated *Fn* mutants: the Fap2 mutant and the FadA mutant, which prevent binding to Gal-GalNAc motifs and cadherins, respectively. Using these mutants, we now demonstrate that Fap2 mutants, but not FadA mutants, reduce *Fn* binding affinity to CAFs. These findings suggest that *Fn* binding to CAFs is specifically mediated through Gal-GalNAc motifs, which are highly expressed on CAFs. However, and as shown in **Figure 2D**, the binding of *Fn* to CAFs is only partially reversed in the Fap2 mutant, suggesting that, similar to cancer cells^{3,4}, *Fn* interacts with CAFs through several interaction partners.

We present this new data in **Figure 2D** of the manuscript. The text in lines 128–129 of the Results section and in lines 349–358 of the Materials and Methods section has been updated accordingly.

Figure 2. D. Binding and invasion of wild type *Fn* 23726, Fap2 and FadA mutant on CT5.3 cells after a four-hour coculture (MOI 50) assessed using the CFSE dye by flow cytometry (n = 3 independent experiments). Statistically significant differences were determined using a repeated measure ANOVA followed by Tukey's HSD post-hoc test.

6.The author should provide flowchart circle gate strategy (iCAF cell surface labeling).

We have added the gating strategy in Supplementary figure 2A.

Supplementary figure 2. A. Flow cytometry gating strategies of the CAF markers, here podoplanin (PDPN).

7. As Fig3 and Fig4 shows Fn polarizes fibroblasts to inflammatory CAFs only by flow chart. The expression of surface membrane proteins on its surface should be proved by WB/IF/IHC, etc.

We thank the reviewer for this comment. While we agree that additional methods could provide further insights, it is important to note that CAF markers have limitations in defining iCAF and myCAF subtypes, as there is still a lack of highly specific markers for these populations⁵. As an alternative approach we investigated the transcript expression levels of multiple genes able to distinguish between the two subtypes. Therefore, we performed RNA sequencing of *Fn*-infected CAFs and used a selection of published marker genes for iCAFs and myCAFs^{6,7}, which confirmed that *Fn*-exposed CAFs exhibit increased iCAF and decreased myCAF signatures. These findings align with our flow cytometry data, which showed increased iCAF and decreased myCAF markers expression.

*We now show this new data in **Figure 3E-F** and have accordingly updated the Results section text in lines 160–162.*

Additionally, we performed an IL-6 and IL-8 ELISA on spent medium of *Fn*-infected CAFs and observed that *Fn* stimulation led to significantly higher IL-6 and a trend in IL-8 secretion compared to control CAFs, further supporting the induction of an iCAF-like phenotype.

*We show this additional data in **Figure 4C-D** and have accordingly updated the text in lines 183–185 of the Results section and in lines 521–526 of the Materials and Methods section.*

Finally, we have also performed immunofluorescent staining using podoplanin as a generally well accepted iCAF marker in CRC⁸. We observed that the podoplanin expression increased in *Fn* treated CAFs compared to an untreated control.

*We now added this new analysis in **Supplementary figure 2E-F** of the manuscript. We have accordingly updated the text in lines 155–158 of the Results section and in lines 475–487 of the Materials and Methods section.*

In some of the assays, we also observed increased cytokine levels in the *Ec* control. Notably, as during co-culture with CAFs, *Ec* proliferates much faster than *Fn*, resulting in different bacterial loads towards the end of co-culture with CAFs, which may account for some of the observed effects (*reflected in line 190-192*). However, as detailed in the manuscript, particularly in the new results presented in **Figure 1** and the discussion (*lines*

285-286), we have now demonstrated that *Fn* specifically binds to the stromal compartment, and to CAFs, a property not observed with *Ec*.

8. As Fig6 shows this *Fn*-elicited effect was CAF-specific as the author did not observe a similar effect upon coinjection with *Fn*-treated NFs. Why *Fn* elicits an CAF-specific effector?

We appreciate the reviewer's insightful point. CAFs may exhibit heightened sensitivity to bacterial signals, which aligns with their elevated expression of pro-inflammatory factors and signaling molecules, ultimately leading to more pronounced phenotypic changes upon infection. Furthermore, CAFs are known to express higher levels of anti-apoptotic molecules such as MCL1 suggesting enhanced survival and prolonged functionality *in vivo*⁹. However, further investigation is needed to fully elucidate how normal fibroblasts

respond to bacterial interactions and whether bacteria contribute to their transition into CAFs.

As noted in our response to comment one, we have removed the subcutaneous model from the manuscript to better align with the *in vitro* findings. Consequently, the reason why CAFs appear more sensitive than NFs following bacterial infection remains unresolved and will be addressed in future studies.

9. Is there any possible Fn Fap2 more likely to bind to CAF surface rather than tumor cell surface? The author should do competition experiments between CAF and tumor cell binding Fn?

We thank the reviewer for this comment. Please also refer to our response to the third point. Initial experiments indeed suggest a higher binding of *Fn* to CAFs compared to tumor cells. However, this result might be biased due to the difference in size of CAFs compared to epithelial cells. We have opted to not highlight this data in the manuscript and believe that such claims (of which cell type in the tumor microenvironment *Fn* would preferentially interact with) are outside the scope of this manuscript, although highly interesting and important to investigate in follow-up studies (*reflected in lines 298–301*).

10. The author should also validate the tumor model *in vivo* by using co-cultured experiment within Fap2KO Fn and CAF.

We thank the reviewer for this comment. Like several studies identifying *Fn* interaction partners on epithelial cells^{3,4}, *Fn* may also bind to and interact with CAFs through multiple, distinct mechanisms. As outlined in our response to point five, we have revised our claims to clarify that the Fap2-GalNAc axis represents one possible binding mechanism, rather than the sole one, since binding of *Fn* to CAFs is only partially reduced with the Fap2 mutant. Thus, to better address the functional effects of *Fn*-treated CAFs on epithelial cells, we have shifted our focus toward downstream mechanisms. Specifically, we found that *Fn*-stimulated CAFs exhibit increased ROS levels.

To investigate the role of ROS in the *Fn*-CAF-driven phenotype, we employed a newly established 3D invasion model, which better reflects invasion of cancer cells. CRC cells are cultured as 3D complex spheroids as they are co-incubated with CAFs. These complex 3D spheroid models are then treated with conditioned medium of CAFs co-cultured either with a blank medium control, *Ec* (bacterial control) or *Fn* 71. Tumor cell outgrowth from the spheroids into a collagen matrix is then monitored as a proxy for invasion and aggressiveness. Using this model, we demonstrated that supernatants from *Fn*-treated CAFs significantly enhanced the invasive potential of tumor spheroids compared to controls (non-bacterial control and *Ec*-stimulated CAF control).

Additionally, we investigated the effect of the increased ROS levels in CAFs on tumor cells. To this aim, CAFs were treated with *Fn* in the presence or absence of N-acetylcysteine (NAC), a ROS scavenger, and the resulting conditioned media were applied to our CRC spheroid model. We demonstrated that NAC treatment during bacterial co-culture of *Fn* with CAFs significantly reduced the pro-invasive effect of the conditioned media on cancer cells, indicating that ROS production in CAFs is a key driver of *Fn*-induced cancer cell invasion.

We present this new data in **Figure 5G-I** of the revised manuscript and have accordingly updated the text in lines 230–243 of the Results section and in lines 543-562 of the Materials and Methods section.

Figure 5. G. Schematic representation of the *in vitro* complex 3D co-culture experimental setup. mCherry labelled HCT116 CRC cells are co-embedded in collagen with GFP labelled CAFs (CT5.3). These multi-component spheroids are then treated with CM from CT5.3 CAFs cultured alone (blank medium control) or CM of CT5.3 co-cultured with *Fn* 71 or *Ec* (four-hour co-culture, MOI 50), in the presence or absence of NAC (5 mM). **H.** Quantification of invasion in the 3D complex spheroid model measured by HCT116 outgrowth (mCherry) over 6 days ($n = 3$ independent experiments, 2-5 technical replicates included per experiment). **I.** Quantified invasion of HCT116 (mCherry) outgrowth at endpoint (day 6). CM = conditioned medium, NAC = N-acetyl-L-cysteine. The error bars in H show the mean \pm SD and statistically significant differences determined using pairwise 2-way repeated measure ANOVAs (time \times treatment). The interaction term p-values were adjusted using Holm's method. The bar chart and error bars in I show the mean \pm SD. Statistically significant differences were determined using a nested ANOVA followed by Tukey's HSD post-hoc test.

11.The presence of *Fn* increases IL6 expression and decreases ACTA2 expression, suggesting that *Fn* induces iCAF. The author should do more experiments (ELSA\WB, etc.) to prove this.

We thank the reviewer for this comment. We have drastically built upon the notion that *Fn* pushes CAFs towards a more iCAF phenotype. We have now performed additional assays such as ELISAs to better characterize the secretome of *Fn* treated CAFs (**Figure 4C-D**), as well as more downstream assays to assess this iCAF phenotype, including flow cytometry (**Supplementary figure 2D**), immunofluorescent staining (**Supplementary figure 2E-F**) and RNA-seq (**Figure 3E-F**). The results have already been described in point seven's response.

Figure 3. E. Schematic representation of the generated RNA-seq dataset. CT5.3, or the patient derived CAF180 and CAF181 CAFs were exposed to *Fn* 71 (MOI 50) for four hours, washed with penicillin/streptomycin (P/S)-containing medium and incubated for a further 24 hours before RNA-sequencing (n = 3 independent cell lines). **F.** Heat map of the RNA-Seq expression log₂ fold-change in CT5.3 and patient-derived CAFs exposed to *Fn* 71 compared to untreated CAFs. **Figure 4. C-D.** IL-6 (C) and IL-8 (D) levels in CM 24 hours after a four-hour co-culture of CT5.3, or the patient derived CAF180 and CAF181 CAFs with *Fn* 71 and *Ec* (MOI 50), measured by ELISA (n = 3 independent experiments with 3 independent cell lines).

Supplementary figure 2. D. Expression of PDPN in CT5.3 CAFs 24 hours after a four-hour treatment with either *Fn* 25586, *Fn* 23726 or *Ec* (MOI 50) as measured by flow cytometry (n= 3 independent experiments). **E.** Normalized mean fluorescent intensity of PDPN in the cytoplasm of CT5.3 cells treated for four hours with *Fn* 71 or *Ec* (MOI 50, n = 1 independent experiment). **F.** Representative immunofluorescent images of PDPN (red) staining, quantified in E, counter stained with DAPI (blue). Scale bar= 20 μm. PDPN = podoplanin. The bar chart and error bars show the mean ± SD and statistically significant differences were determined using a nested ANOVA followed by Tukey's HSD post-hoc test.

12. As *Fn* induced a higher increase in mitochondrial ROS levels in CAFs. How about apoptosis in CAFs induced by *Fn*?

We thank the reviewer for this comment. We performed an Annexin V apoptosis assay and could not identify any difference in apoptosis rates between Control, *Ec* and *Fn* treated CAFs (**Supplementary figure 2B**). Please refer to point four's response.

Supplementary figure 2. B. Cell viability of CT5.3 CAFs and the patient derived CAF180 and CAF181 CAFs treated with *Fn* 25586, *Fn* 23726, *Fn* 71 or *Ec* for four hours at an MOI of 50, determined by the Annexin IV assay via flow cytometry (n= 3 independent experiments).

13. As supplementary fig5B-C showed that *Fn* 25586 was a more potent CXCL1 inducer than *Fn* 23726, why?

We thank the reviewer for this comment. Different *Fn* strains can exhibit similar phenotypic effects but with varying degrees of intensity, likely due to differences in their genetic composition, virulence factor expression, and adaptation to specific tumor microenvironments. *Fn* strain 71, which has shown the most pronounced effects, is considered the most clinically relevant, suggesting that it may possess enhanced pathogenic traits, such as increased adhesins, immune evasion mechanisms, or a higher capacity to modulate host cell signaling. In contrast, strain *Fn* 23726 consistently exhibits the weakest effects, which could be attributed to differences in its ability to colonize, interact with host cells, or induce inflammatory and pro-tumorigenic pathways. The underlying molecular mechanisms driving these variations remain unclear and warrant further investigation to determine how strain-specific factors contribute to differences in tumor progression and metastatic potential. Throughout the manuscript, we have intentionally included multiple *Fn* strains to demonstrate that our findings are not limited to a single bacterial strain, thereby strengthening the validity of our results.

A sentence has been added in lines 279–280 of the Discussion to address this comment.

14. The author should pay more attention on quality of Figures, like Fig6C. Several annotations are not clear to read.

We have now carefully revised all the Figures and annotations. We hope they are now clearer for the reader.

Reviewer #2:

Karta et al "Fusobacterium nucleatum binds to and invades cancer-associated fibroblasts to enhance colorectal cancer progression"

Authors study how microbes in CRC may interact with CAFs. This is relatively new area of research as previously people focused on cancer and immune cells. Authors specifically look at Fusobacterium (Fn)-CAF interactions and what may be an effect on CRC progression.

They find that that Fn binds to and invades CAFs, just like they do with cancer cells or macrophages. This leads to CAF secretion of pro-inflammatory cytokines and membrane-associated proteases. CAF which have Fn exhibit metabolic shift with an increases in ROS from mitochondria and reduction in methylation substrate availability.

Authors utilize a co-culture model and find that Fn-treated CAFs induce stronger migration of cancer cells. They further find a possible role for CXCL1 (known to be induced by Fn from previous studies). Analysis of human samples shows that high loads of Fn lead to CAFs to be mostly of inflammatory phenotype (iCAF), meanwhile low Fn situation leads primarily to the presence of myofibroblastic CAFs (myCAF).

Overall this is interesting study with most of the experiments well performed, controlled and interpreted. Data on Gal-GalNAc motifs and mode of binding to CAFs is interesting. Few additional experiments, controls and additional explanations are probably required.

1) Authors need to more explicitly describe properties/genetic assignment of their Fn strains in view of different clade papers for Fn in CRC by Bullman/Johnson group Nature 2024- which ones are used here?

We thank the reviewer for this comment. Indeed, there are many discussions arising in the *Fn* field regarding the differences between strains. In this paper we have used several strains of *Fn*, namely *Fusobacterium nucleatum* ssp. *Nucleatum* 25586, *Fusobacterium nucleatum* ssp. *Nucleatum* 23726 and *Fusobacterium nucleatum* ssp. *Animalis* 7_1. We believe that this approach provides a broad perspective of the effect of *Fn* on CAFs, yet we fully acknowledge that further research is necessary to elucidate the unique characteristics and pathogenic potentials of these specific strains in relation to their effect and function on CAFs. We have updated the manuscript text to be able to clarify the strains used.

Notably, we found *Fn animalis* (*Fn* 71), isolated from the human gastrointestinal tract and reported to have pro-tumorigenic functions in CRC¹⁰, to exhibit the highest affinity (**Figure 1I** and **Supplementary figure 1C**). Further studies are required to elucidate the mechanisms underlying the strain-specific variability in *Fn* binding to CAFs and the resulting downstream effects on tumor cells.

A sentence has been added in lines 279–280 of the Discussion to address this comment.

2) For Fig2, it seems to be essential mechanistically to try to disrupt at least some host genes involved into binding and intracellular experiments, in addition to presented experiments with GalNAc treatment

We thank the reviewer for this comment. Deleting individual genes involved in N- and O-glycan biosynthesis without affecting CAF survival is extremely difficult, if not impossible. However, to further validate the binding mechanism and confirm its mediation via Gal-GalNAc motifs, we explored alternative approaches and found that CAFs also express high levels of cadherins.

*We have included this additional analysis in **Figure 2C** and updated the text in line 125-126 of the Results section.*

To this end, we collaborated with Prof. Jörg Vogel, Director of the Helmholtz Institute for RNA-based Infectious Research in Würzburg, who recently developed new *Fn* Fap2 mutants, which we obtained during the manuscript revision phase. Using these newly generated *Fn* Fap2 mutants, as well as FadA mutants (known to bind cadherins), we now demonstrate that the Fap2 mutants significantly reduce *Fn* binding affinity to CAFs, while the FadA mutants do not. Together with our previously presented GalNAc data, these findings suggest that *Fn* binding affinity is mediated through the Fap2–Gal-GalNAc axis.

*We present this new data in **Figure 2D** of the revised manuscript and have accordingly updated the text in lines 128–129 of the Results section and in lines 349–358 of the Materials and Methods section.*

However, as shown in **Figure 2D**, the binding of *Fn* to CAFs is only partially reversed in the Fap2 mutant, suggesting that, similar to cancer cells^{3,4}, *Fn* interacts with CAFs through several interaction partners. Thus, we acknowledge that this is one of the many possible binding and invasion mechanisms of *Fn* with CAFs.

We have revised the manuscript to clearly convey this message in lines 131–134 of the Results section.

Figure 2. C. Heatmap showing the expression of cadherin genes in a CAFs and tumor spheroids (T) generated RNA-seq dataset. Columns show the expression assessed in independent experiments of patient-derived (T4) and HT-29 tumor spheroids as well as patient-derived CAFs (n = 9 patients: CAF4, CAF12, CAF16, CAF19, CAF20, CAF22, CAF32, CAF41 and CAF42). Expression values are median ratio normalized counts on a log2 scale. D. Binding and invasion of wild type *Fn* 23726, *Fap2* and *FadA* mutant on CT5.3 cells after a four-hour co-culture (MOI 50) assessed using the CFSE dye by flow cytometry (n = 3 independent experiments). Statistically significant differences were determined using a repeated measure ANOVA followed by Tukey's HSD post-hoc test.

3) Fig 3- better characterization of iCAF (or myCAF) fibroblasts with regard to the effect of *Fn* is needed, past only a few markers such as IL6 etc.

We thank the reviewer for this comment and we have built upon this claim. While we agree that additional methods could provide further insights, it is important to note that CAF markers have limitations in defining iCAF and myCAF subtypes, as there is still a lack of highly specific markers for these subpopulations (*now reflected in lines 97-100 of the manuscript*). To investigate the expression levels of multiple genes able to distinguish between the two subtypes, we performed RNA sequencing of *Fn*-infected CAFs and used a selection of published marker genes for iCAFs and myCAFs^{2,6}. We were able to confirm that *Fn*-exposed CAFs exhibit increased iCAF and decreased myCAF signatures. These findings align with our flow cytometry data.

We show this new data in **Figure 3E-F** of the revised manuscript and accordingly updated the text in lines 160–162 of the Results section.

Additionally, we performed an IL-6 and IL-8 ELISA on spent medium of *Fn*-infected CAFs and observed that *Fn* stimulation led to significantly higher IL-6 and a trend in IL-8 secretion compared to control-activated CAFs, further supporting the induction of an iCAF-like phenotype.

These findings are now shown in **Figure 4C-D**. The text in lines 183–185 of the Results section and in lines 521–526 of the Materials and Methods section has been updated accordingly.

Finally, we have also performed some immunofluorescent staining using podoplanin as a generally well accepted iCAF marker⁸. We observed that *Fn* increases podoplanin expression in treated CAFs compared to an untreated or bacterial (*Ec*) control.

We present this new data in **Supplementary figure 2E-F** and accordingly updated the text in lines 155–158 of the Results section and in lines 475–487 of the Materials and Methods section.

It is important to note that in some assays, the *Ec* control also induced elevated cytokine levels. This response may be explained by the substantially faster doubling time of *Ec* compared to *Fn* during co-culture with CAFs (reflected in lines 190-192), which could contribute to the observed increase. However, and as detailed in the manuscript, particularly in the new results presented in **Figure 1** and the discussion (lines 285-286), we now provide evidence that *Fn* specifically binds to the stromal compartment, including CAFs, a property not observed with *Ec*. This supports the conclusion that the effects of *Fn* are TME-specific.

Figure 3. **E.** Schematic representation of the generated RNA-seq dataset. CT5.3, or the patient derived CAF180 and CAF181 CAFs were exposed to *Fn* 71 (MOI 50) for four hours, washed with penicillin/streptomycin (P/S)-containing medium and incubated for a further 24 hours before RNA-sequencing (n = 3 independent cell lines). **F.** Heat map of the RNA-Seq expression log₂ fold-change in CT5.3 and patient-derived CAFs exposed to *Fn* 71 compared to untreated CAFs. **Figure 4.** **C-D.** IL-6 (**C**) and IL-8 (**D**) levels in CM 24 hours after a four-hour co-culture of CT5.3, or the patient derived CAF180 and CAF181 CAFs with *Fn* 71 and *Ec* (MOI 50), measured by ELISA (n = 3 independent experiments with 3 independent cell lines).

Supplementary figure 2. **D.** Expression of PDPN in CT5.3 CAFs 24 hours after a four-hour treatment with either *Fn* 25586, *Fn* 23726 or *Ec* (MOI 50) as measured by flow cytometry (n= 3 independent experiments). **E.** Normalized mean fluorescent intensity of PDPN in the cytoplasm of CT5.3 cells treated for four hours with *Fn* 71 or *Ec* (MOI 50, n = 1 independent experiment). **F.** Representative immunofluorescent images of PDPN (red) staining, quantified in E, counter stained with DAPI (blue). Scale bar= 20 μm. PDPN = podoplanin. The bar chart and error bars show the mean ± SD and statistically significant differences were determined using a nested ANOVA followed by Tukey's HSD post-hoc test.

4) Fig4 TCGA data on ROS pathways in total tumor are only correlational that these changes happen in CAF and not in cancer cells

We agree that the data obtained from TCGA is only correlational and, given that TCGA sequencing is performed on bulk tumor tissue, it does not allow for cell type-specific conclusions. Nevertheless, we then specifically used CAFs *in vitro* to validate this observation (**Figure 4G**).

We now clearly present the limitations of these findings in the updated lines 202–203 of the Results section.

5) Data on Fig 6 should be reproduce with another bacteria (E.coli). there are publications that bacteria (or even LPS) increase tumor growth. As presented, it does not really illuminate specific and unique role of Fn.

We thank the reviewer for this comment. As rightly pointed, the subcutaneous model does not accurately reflect invasion nor metastasis, which we believe to be the main effect of *Fn*-exposed CAFs on tumor cells. To better address tumor cell invasiveness and metastasis dissemination, we performed a series of new assays.

First, we established a new *in vitro* assay which better reflects the invasion of cancer cells. CRC cells are cultured as 3D complex spheroids as they are co-incubated with CAFs. These complex 3D spheroid models are then treated with conditioned medium of CAFs co-cultured either with a blank medium control, *Ec* (bacterial control) or *Fn* 71. Tumor cell outgrowth from the spheroids into a collagen matrix is then monitored as a proxy for invasion and aggressiveness. Using this model, we demonstrated that supernatants from *Fn*-treated CAFs significantly enhanced the invasive potential of tumor spheroids compared to controls (non-bacterial control and *Ec*-stimulated CAF control).

*We included this new data in **Figure 5G-I** of the revised manuscript and accordingly updated the text in lines 230–243 of the Results section and in lines 543–562 of the Materials and Methods section.*

Second, to address metastatic dissemination *in vivo*, we amended our animal protocol to include a metastasis dissemination assay. In this assay, cancer cells are injected into the tail vein, and metastatic outgrowth is assessed in the lungs using an experimental tail vein metastasis assay.

In our experiment, cancer cells were pre-stimulated with conditioned media from either CAFs treated with a control bacterium (*Ec*), or *Fn* 71. We observed that cancer cells treated with the conditioned media from CAFs pre-stimulated with *Fn* significantly

increased metastasis outgrowth at the secondary site (lung) compared to control CAF conditioned media or conditioned media from CAFs treated with *Ec*.

We present this new data in **Figure 6A-C** and accordingly updated the text in lines 244–251 of the Results section and lines 612–621 of the Materials and Methods section.

We believe that these new results better support the claim that *Fn*-exposed CAFs promote tumor cell invasion and metastatic outgrowth as highlighted by the reviewer. Furthermore, no effect on metastatic burden was observed following exposure of cancer cells to supernatant from *Ec*-treated CAFs compared to non-treated CAFs, underscoring that these effects are *Fn* specific. We have now revised the manuscript by including these results and have chosen to remove the subcutaneous *in vivo* experiment to focus more clearly on the invasion phenotype.

Figure 5. G. Schematic representation of the *in vitro* complex 3D co-culture experimental setup. mCherry labelled HCT116 CRC cells are co-embedded in collagen with GFP labelled CAFs (CT5.3). These multi-component spheroids are then treated with CM from CT5.3 CAFs cultured alone (blank medium control) or CM of CT5.3 co-cultured with *Fn* 71 or *Ec* (four-hour co-culture, MOI 50), in the presence or absence of NAC (5 mM). **H.** Quantification of invasion in the 3D complex spheroid model measured by HCT116 outgrowth (mCherry) over 6 days ($n = 3$ independent experiments, 2-5 technical replicates included per experiment). **I.** Quantified invasion of HCT116 (mCherry) outgrowth at endpoint (day 6). CM = conditioned medium, NAC = N-acetyl-L-cysteine. The error bars in H show the mean \pm SD and statistically significant differences determined using pairwise 2-way repeated measure ANOVAs (time \times treatment). The interaction term p-values were adjusted using Holm's method. The bar chart and error bars in I show the mean \pm SD. Statistically significant differences were determined using a nested ANOVA followed by Tukey's HSD post-hoc test.

Figure 6. A. Schematic representation of the *in vivo* experimental setup. SPF NSG mice were tail vein injected with 1×10^6 HT-29 cells, which were pre-treated with CM of CT5.3 CAFs cultured alone or CM of CT5.3 co-cultured with *Fn* 71 or *Ec* (4-hour co-culture, followed by a 24-hour incubation, MOI 50). Lungs were harvested after 30 days ($n = 8$ mice for the Ctrl and *Ec* conditions and $n = 7$ for the *Fn* condition). **B.** Quantification of the relative tumor surface area in percentage per imaged lobe (four lobes were analyzed per mouse). **C.** Representative H&E images of tumors (indicated by black arrows) in the lungs at day 30. CM = conditioned medium. Scale bar = 100 μ m. Statistically significant differences were determined using an ANOVA on a robust linear model fit (using the lmRob function in the R package robust) followed by Tukey's multiple comparison method (using the R package multcomp).

6) Mitochondrial ROS data/reprogramming in CAF may be interesting, but what is the functional role?

In this study, we demonstrate that *Fn*-treated CAFs exhibit elevated levels of ROS, which coincides with their ability to enhance the motility of CRC cells. This observation suggests a potential link between ROS production in CAFs and the promotion of cancer cell invasion. Importantly, the increased motility appears to be driven by factors present in the conditioned medium of *Fn*-treated CAFs, highlighting the role of ROS in shaping the tumor microenvironment. Consistent with our findings, previous studies have shown that elevated ROS in CAFs can stimulate the release of cytokines and chemokines, thereby promoting tumor cell motility¹¹. Furthermore, pharmacological inhibition of mitochondrial ROS has been reported to reduce NOX4 expression, a key enzyme in TGFβ-mediated fibroblast reprogramming¹², supporting the functional relevance of ROS in modulating CAF behavior.

To further investigate the role of ROS in the *Fn*-CAF-driven phenotype, we employed a newly established 3D invasion model, as introduced in point five. CAFs were treated with *Fn* in the presence or absence of N-acetylcysteine (NAC), a ROS scavenger, and the resulting conditioned media were applied to our CRC spheroid model. Using this assay, we showed that NAC treatment during bacterial co-culture of *Fn* with CAFs significantly reduced the pro-invasive effect of the conditioned media on cancer cells, indicating that ROS production in CAFs is a key driver of *Fn*-induced cancer cell invasion.

Figure 5. G. Schematic representation of the *in vitro* complex 3D co-culture experimental setup. mCherry labelled HCT116 CRC cells are co-embedded in collagen with GFP labelled CAFs (CT5.3). These multi-component spheroids are then treated with CM from CT5.3 CAFs cultured alone (blank medium control) or CM of CT5.3 co-cultured with *Fn* 71 or *Ec* (four-hour co-culture, MOI 50), in the presence or absence of NAC (5 mM). **H.** Quantification of invasion in the 3D complex spheroid model measured by HCT116 outgrowth (mCherry) over 6 days (n = 3 independent experiments, 2-5 technical replicates included per experiment). **I.** Quantified invasion of HCT116 (mCherry) outgrowth at endpoint (day 6). CM = conditioned medium, NAC = N-acetyl-L-cysteine. The error bars in H show the mean \pm SD and statistically significant differences determined using pairwise 2-way repeated measure ANOVAs (time \times treatment). The interaction term p-values were adjusted using Holm's method. The bar chart and error bars in I show the mean \pm SD. Statistically significant differences were determined using a nested ANOVA followed by Tukey's HSD post-hoc test.

7) Authors imply altered migration but show increased tumor growth *in vivo*. They should analyze invasion and metastasis probably, as a direct function of migration. In primary model of tumor growth authors should assess proliferation and cell death *in vivo* in addition to migration.

We agree with the reviewer and have now replaced the subcutaneous model. To better reflect cancer cell invasion, we have included two new assays: (1) a 3D spheroid invasion assay *in vitro* (**Figure 5G-I**) and (2) *in vivo* tail vein metastasis assay (**Figure 6A-C**). Please refer to point five for the results.

In *in vivo* model, it would be interesting to use *Fn*-infected cancer cells as well.

Is *FN* infection of CAF much more important than *FN* infection of cancer cells. Is there an *in vivo* situation where CAFs are infected but cancer cells are not? Is it enough to infect cancer cells for the “full phenotype”?

We thank the reviewer for this highly relevant comment. Over the last years, several studies have evaluated the effect of *Fn* on cancer cells *in vitro* and *in vivo*^{3,4}. In our current study, we show for the first time that *Fn* can also interact with CAFs within the TME. However, we do not claim in our manuscript that *Fn*-infected CAFs play a superior role in tumor progression compared to *Fn*-infected epithelial cells. Nevertheless, to get some insights into the potential preferential binding of *Fn* to CAFs over cancer cells, we reanalyzed our flow cytometry data. As CAFs and tumor cells differ in size, the measured binding fluorescence needs to be normalized to the cell size. To this end, and as shown in **Figure A** below (*for reviewing purposes only*), we used the forward scatter (FSC-A) measures of our flow cytometry data as a proxy for cell size. As shown in **Figure B** (*for reviewing purposes only*), such *Fn* binding values normalized to FSC-A still support that the binding is higher on CAFs than on epithelial cells. However, as we believe that this approach is prone to inherent technical limitations, we decided to not include it in the revised manuscript and didn't make definitive claims regarding cell-type specificity of *Fn* binding within the TME.

Along this line, we present additional new data (in **Figure 1A-F** of the revised manuscript) demonstrating that *in vivo* *Fn* binds to the stromal as well as epithelial compartments of CRC tissue.

Most importantly, we decided in this manuscript to provide new evidence strengthening the observation that *Fn* can interact with CAFs, a finding that broadens our understanding of its potential cellular targets without asserting preferential interaction over other cell types. Nevertheless, we agree with the reviewer that such a comparative analysis is of high interest and needs to be investigated in follow-up projects (*reflected in lines 298–301 of the Discussion*).

Figure for reviewing purposes. Mean fluorescence intensity of the forward scatter (FSC-A) as a proxy for cell size (**A**) as well as MFI normalized to size (**B**) in CT5.3, HCT116 and HT29 flow cytometry binding data.

Figure 1. **A.** Re-analysis of the CSI-Microbe Pelka2021 dataset^{1,2} showing *Fusobacterium* infected cells per mille (%) for each cell type in the tumor. **B-C.** *In situ* hybridization staining of human CRC tissue with α SMA in brown (**B, C**), PDPN in purple (**C**), and *Fn* in pink (**B, C**; indicated by yellow arrows) in two independent CRC patients. Scale bar = 50 μ m. **D.** Schematic representation of the germ free *CDX2-CreER^{T2}Apc^{fl/fl}* experiment. Briefly, tumorigenesis was induced by i.p. injections of tamoxifen (75 mg per kg of body weight) for five daily injections, followed by three times a week oral gavage with 10^8 CFU/mouse of *Fn* 71 or PBS control. **E.** Representative immunofluorescent images showing *Fn* 71 (red, white arrows), stained with an OMP *Fn* specific antibody, colocalized with the stromal marker PDPN (green), and DAPI (grey) in a dysplastic region of the colon from the germ free *CDX2-CreER^{T2}Apc^{fl/fl}* experiment. Images are representative of two mice from one experiment. Scale bar = 50 μ m. **F.** Attachment and binding assay of *Fn* 71 and *Ec* (red, indicated by white arrows, DAPI in grey) on human CRC tissue. Epithelium (E) and stromal (S) stromal compartments separated by white dashed line. Images are representative of one patient and two independent experiments. Scale bar = 50 μ m.

Of note, our findings are particularly relevant to the scientific community, as they may help explain the poor prognosis observed in CMS4 patients, a subtype of CRC patients with high stromal content, especially when stratifying the cohort based on *Fn* expression levels¹³. In line with our findings, we also observed poorer survival in *Fn*-high patients

within the CMS4 subtype, along with a notable increase in the iCAF compartment and reduction in the myCAF compartment, further supporting our experimental data.

We now present this new data in **Figure 3G-H** and in the text lines 163–168 of the Results section.

Figure 3. G. Kaplan-Meier curve of progression-free survival by *Fn* load in either the complete CRC cohort or by CMS4/CRIS-B classification. **H.** iCAF and myCAF scores (calculated using the R package singscore and the genesets from (F)) by *Fn* load and CMS4/CRIS-B classification. Statistically significant differences were determined in **G** using an ANOVA on the Cox proportional hazard models to assess the interaction significance and in **H** using a two-tailed t-test with Holm’s method adjusted p-values,

Reviewer #3:

Summary:

Karta et al explore how *Fusobacterium nucleatum* (Fn) affects cancer-associated fibroblasts (CAFs) and the functional consequences for tumour cells. They find that Fn binds to and invades CAFs, leading them to secrete pro-inflammatory cytokines and membrane-associated proteases. They identify CXCL1 as a key cytokine in this process. Using patient data they show that CAFs from patients with high loads of Fn were different to those with low loads. This is an interesting study on an important area, but some additional work is needed before publication.

Major points:

In Figure 2C binding and invasion seems only minimally reduced with the Fap2 mutant Fn. Could the authors knock down genes linked to N- and O-glycan biosynthesis in the CAFs to see if this reduces Fn binding.

We thank the reviewer for this comment. It is extremely difficult, if not impossible, to delete individual genes involved in N- and O-glycan biosynthesis without compromising the survival of CAFs. However, we would like to note that we have repeated the experiments using new *Fn* Fap2 and FadA mutants, as we were also puzzled by the initially observed low reduction in *Fn* binding to CAFs using the Fap2 mutants we had obtained from a collaborator in the US.

This time, we reached out to Prof. Jörg Vogel, Director of the Helmholtz Institute for RNA-based Infectious Research in Würzburg, who had recently generated new Fap2 and FadA *Fn* mutants. Using these newly established mutants, we now demonstrate that these Fap2 mutants significantly reduce *Fn* binding affinity to CAFs, with a much clearer effect than in our initial experiments. We have therefore now replaced the old figure with the new figure. Additionally, to explore the specificity of this binding, we investigated alternative pathways and found that CAFs also express high levels of cadherins.

*We have included this additional analysis in **Figure 2C** and have accordingly updated the text in line 126-129 of the Results section.*

We then tested FadA mutants and demonstrated that while Fap2 mutants significantly reduce *Fn* binding affinity to CAFs, FadA mutants do not. These findings suggest that *Fn* binding to CAFs is mediated through Gal-GalNAc motifs, which are highly expressed in CAFs. This is in line with the binding/adhesion assay we previously performed and in which the presence of N-acetyl-D-galactosamine peptide (GalNAc) was able to reduce the binding of *Fn* (this data was already presented in the previous version of the manuscript and now shown in **Figure 2E**).

These new mutant analyses are now presented in **Figure 2D**. We accordingly updated the text in lines 126–129 of the Results section and in lines 349–358 of the Materials and Methods section.

However, as shown in *Figure 2D*, the binding of *Fn* to CAFs is not fully abrogated in the *Fap2*-deficient mutant, indicating that, as with cancer cells^{3,4}, *Fn* likely engages multiple receptors or interaction partners on CAFs. We therefore acknowledge that *Fap2* represents just one of several potential mechanisms involved in *Fn* binding and invasion of CAFs.

We have revised the manuscript to clarify this point in lines 132–134 of the Results section.

Figure 2. **C.** Heatmap showing the expression of cadherin genes in a CAFs and tumor spheroids (T) generated RNA-seq dataset. Columns show the expression assessed in independent experiments of patient-derived (T4) and HT-29 tumor spheroids as well as patient-derived CAFs (n = 9 patients: CAF4, CAF12, CAF16, CAF19, CAF20, CAF22, CAF32, CAF41 and CAF42). Expression values are median ratio normalized counts on a log2 scale. **D.** Binding and invasion of wild type *Fn* 23726, *Fap2* and *FadA* mutant on CT5.3 cells after a four-hour co-culture (MOI 50) assessed using the CFSE dye by flow cytometry (n = 3 independent experiments). Statistically significant differences were determined using a repeated measure ANOVA followed by Tukey's HSD post-hoc test.

The findings with respect to mitochondrial ROS are interesting. To properly assess increased activity of the TCA cycle and oxidative phosphorylation it would be ideal to perform either Seahorse Extracellular Flux Analysis or Stable Isotope Tracer Analysis (with ¹³C labelled glucose and glutamine), as total TCA intermediate levels are not very informative in this context.

We thank the reviewer for their comment and suggestions. We have opted to perform [U-¹³C] glutamine tracing as glutamine is the major substrate for anaplerosis of the TCA cycle. We conducted our [U-¹³C] glutamine tracing for 24 hours on *Fn*-treated CAFs to ensure we are at the metabolic and isotopic steady state at the point of metabolite extraction. We observed a significant increase in the relative flux towards glutamate and

alpha-ketoglutarate from glutamine in CAFs treated with two *Fn* strains. This finding indicates an increase usage of glutamine in the TCA cycle, highlighting increased activity of the cycle.

We present this new data in **Supplementary Figure 3D-E** and accordingly updated the text in lines 208–213 of the Results section and in lines 563–576 of the Materials and Methods section.

Supplementary figure 3. D-E. Fractional enrichment in M+5 glutamate (**D**) and M+5 alpha-ketoglutarate (**E**) from [U-¹³C]glutamine after a four-hour co-culture of CT5.3 with either *Fn* 71, *Fn* 25586, *Ec* or PBS control (MOI 50), followed by a 24-hour incubation with [U-¹³C]glutamine. The bar chart and error bars show the mean \pm SD and statistically significant differences were determined using a nested ANOVA followed by Tukey's HSD post-hoc test.

The authors state that: "It has previously been reported that cellular demethylases are particularly sensitive to the intracellular levels of the TCA cycle metabolite alpha-ketoglutarate." However, α KG does not vary in their experiments although the S-adenosyl methionine to S-adenosyl homocysteine ratio is reduced. If the authors wish to include these data in the manuscript I feel like more work needs to be done in this regard. Looking at the downstream consequences and methylation patterns for instance.

We thank the reviewer for the comment, and we fully agree. As we do not have specific expertise in this area and have not conducted further investigations into methylation patterns, we have decided to remove the sentence from the manuscript. This aspect is not essential to the main message of the paper.

In Figure 5, to conclude that reprogrammed CAFs promote the migration of tumor cells, at least one additional CRC line should be used.

We thank the reviewer for the suggestions. To address this, we have repeated the transwell migration assay with another CRC cell line, namely SW480. Similarly to the results obtained initially with the HCT116 CRC cell line, we observed increased invasion upon *Fn* stimulation of CAFs, compared to blank medium control and *Ec* (bacterial control).

We have included this new data in **Supplementary Figure 4A** and updated the text in lines 227–228 of the Results section and lines 539–542 of the Materials and Methods section accordingly.

Supplementary figure 4. A. Percentage of SW480 cells that invaded upon co-culture with four-hour non- and bacteria (*Fn* 71 and *Ec*)-treated CT5.3 CAFs (MOI 50, n = 3 biological replicates).

Minor points:

Could the authors explain their choice of control bacteria.

We thank the reviewer for this comment. Selecting an appropriate control bacterium in the cancer microbiome field is a significant challenge. We chose a Gram-negative bacterium, as *Fn* is also Gram-negative, and opted for a strain of *E. coli*, consistent with a number of previous studies^{14–16}. However, we acknowledge that this choice is not ideal, as *Ec* has a much faster doubling time compared to *Fn* (now reflected in lines 190–192).

Figure 1D needs a scale bar. Could arrows be added to Figure 1D to indicate what the authors describe?

The scale bar and arrows have been added to **Figure 1D** for increased clarity.

How the RNA-seq data set in Figure 2A was generated needs more explanation in the results, including what gene expression is relative to.

Figure 2A (**Figure 2B** in the revised manuscript) as well as the new **Figure 2C** do not show expression values relative to a condition. Both heatmaps show the log₂ of median ratio normalized expression values. The median ratio normalization (MRN) is the normalization method implemented in the R package DESeq2 and obtained using the “counts” method with the argument “normalized = TRUE”. By doing so, counts are divided by sample-specific size factors determined by median ratio of gene counts relative to geometric mean.

We added this information to the Materials and Methods section of the manuscript in lines 497–504 and now show a small schematic in **Figure 2A** to illustrate how we generated these samples.

Figure 3A is very confusing, I think this needs to be presented and described much more clearly.

The presentation of Figure 3A (now **Figure 3C**) has been changed and the caption and text adjusted for increased clarity.

Figure 3. C. Log₂ fold change in IL6 and ACTA2 expression in *Fn* treated versus untreated CAF05 cells (two-hour co-culture, MOI 500) (n=3 biologically independent experiments) as well as in patient derived CAFs (CAF4 and CAF32, n=1 for each).

Legend in Figure 6B is confusing.

This legend has been edited for increased clarity.

Reviewer #4:

Summary:

While the interaction between Fn and tumour cells has been extensively documented in the literature, little is known about the interaction of the bacteria with the tumour microenvironment (TME) and its potential role in the disease.

In this article, the authors document a direct interaction between *Fusobacterium nucleatum* (Fn) and cancer-associated fibroblasts (CAFs) via the bacterial outer membrane protein Fap2 and tumour cell Gal-GalNAc, activating an inflammatory phenotype in the CAFs and elevated ROS levels. Using co-culture assays, evidence is presented for Fn-treated CAF's being able to promote CRC cell-line migration and in vivo tumour growth via enhanced production of pro-inflammatory cytokines such as CXCL1. A correlation between higher Fn loads and elevated inflammatory CAFs and ROS levels in CRC patients is presented. Overall, the findings are novel and potentially reveal novel opportunities for therapeutic intervention. However, the study is overly reliant on in vitro co-culture models and assays incorporating very limited numbers of CRC cell lines, for which the selection rationale is lacking.

Major critique :

Although these findings are potentially highly novel and clinically relevant, there is an over-reliance on relatively simple in vitro models with limited in vivo validation of major findings. Whilst the simple co-culture models employed suffice to document potential interactions and their functional consequences, the major findings must be confirmed in more complex culture (for example human CRC organoid/CAF co-culture) and in vivo CRC models (for example infection of mouse CRC models) to ensure that the documented findings are relevant in more physiologically accurate settings.

It should also be possible to document interactions between Fn and CRC stromal populations in CRC tissues via IHC/IF.

We fully agree with the reviewer that these are very important points which we now addressed by performing several new experiments:

1. First, we re-analyzed a publicly available scRNA-Seq-based CSI-Microbes data^{1,2} from CRC tumor samples and identified *Fn* within the fibroblast compartment, further suggesting a potential interaction between Fn and CAFs in human CRC patients.

*We included this new analysis in **Figure 1A** and updated the lines 86–89 of the Results section and lines 589–590 of the Materials and Methods section accordingly.*

2. Furthermore, we conducted an attachment/binding assay directly on human CRC tissue. Using labeled *Fn*, we observed that *Fn* binds significantly to CRC tissue, particularly stromal cell-rich regions, in contrast to *Ec*, which exhibited minimal binding.

These findings support that *Fn* is present and interacts with the stromal compartment in patient CRC tissue.

*This additional finding has been added to **Figure 1F**. We have updated the text in lines 102–107 of the Results section and lines 381–389 of the Materials and Methods section accordingly.*

3. To directly establish human clinical relevance, we investigated the presence of *Fn* in human CRC tissue using in situ hybridization (ISH), alongside immunostaining for α -SMA, and podoplanin (PDPN), two accepted stromal markers. We observed areas where *Fn* staining (red dots) was in close proximity to α SMA-positive cells (brown) and PDPN-positive cells (purple), suggesting that *Fn* is indeed present *in vivo* in patients in stroma-rich regions. This new data further supports that *Fn* is present in the stromal compartment in a human physiologically relevant context.

*We present this new analysis in **Figure 1B-C** and accordingly updated the lines 89–93 of the Results section and lines 407–432 of the Materials and Methods section.*

4. Last, we performed an *in vivo* experiment in which germ-free CDX2-CreER^{T2}Apc^{fl/fl} mice were orally gavaged three times a week with *Fn* 71 over a span of three weeks. By co-staining FFPE colon slides of these mice using antibodies recognizing *Fn* (OMP) and PDPN, a known CAF marker (yet used more cautiously in the manuscript as a stromal marker), we were able to localize *Fn* within the stromal compartment of these mice.

*We show this new data in **Figure 1D-E** and have updated lines 93–97 of the Results section and lines 606–611 of the Materials and Methods section accordingly.*

Altogether, this new data indicates that *Fn* is present within the stromal compartment *in vivo*. The revised manuscript mainly addresses this point in **Figure 1**, which supports the notion that *Fn* may influence and reshape the stroma, and therefore also CAFs.

Figure 1. **A.** Re-analysis of the CSI-Microbe Pelka2021 dataset^{1,2} showing *Fusobacterium* infected cells per mille (‰) for each cell type in the tumor. **B-C.** *In situ* hybridization staining of human CRC tissue with α SMA in brown (**B**, **C**), PDPN in purple (**C**), and *Fn* in pink (**B**, **C**; indicated by yellow arrows) in two independent CRC patients. Scale bar = 50 μ m. **D.** Schematic representation of the germ free *CDX2-CreER^{T2}Apc^{fl/fl}* experiment. Briefly, tumorigenesis was induced by i.p. injections of tamoxifen (75 mg per kg of body weight) for five daily injections, followed by three times a week oral gavage with 10⁸ CFU/mouse of *Fn* 71 or PBS control. **E.** Representative immunofluorescent images showing *Fn* 71 (red, white arrows), stained with an OMP *Fn* specific antibody, colocalized with the stromal marker PDPN (green), and DAPI (grey) in a dysplastic region of the colon from the germ free *CDX2-CreER^{T2}Apc^{fl/fl}* experiment. Images are representative of two mice from one experiment. Scale bar = 50 μ m. **F.** Attachment and binding assay of *Fn* 71 and *Ec* (red, indicated by white arrows, DAPI in grey) on human CRC tissue. Epithelium (E) and stromal (S) stromal compartments separated by white dashed line. Images are representative of one patient and two independent experiments. Scale bar = 50 μ m.

Next, to better address the CAF-induced metastatic dissemination, we have performed the following experiments:

5. We have established a new 3D complex *in vitro* assay. For this assay, CRC cells are cultured as 3D spheroids and co-incubated with CAFs. These complex 3D spheroid models are then treated with conditioned medium of CAFs which were either grown alone (blank medium control) or co-cultured with either *Ec* (bacterial control) or *Fn* 71. Tumor cell outgrowth from the spheroids into a collagen matrix is then monitored as a proxy for invasion and aggressiveness. Using this model, we demonstrated that supernatants from *Fn*-treated CAFs significantly enhanced the invasive potential of tumor spheroids compared to controls.

This additional data is shown in Figure 5G-I. We have updated the lines 230–243 of the Results section and lines 543–562 of the Materials and Methods section accordingly.

Figure 5. G. Schematic representation of the *in vitro* complex 3D co-culture experimental setup. mCherry labelled HCT116 CRC cells are co-embedded in collagen with GFP labelled CAFs (CT5.3). These multi-component spheroids are then treated with CM from CT5.3 CAFs cultured alone (blank medium control) or CM of CT5.3 co-cultured with *Fn* 71 or *Ec* (four-hour co-culture, MOI 50), in the presence or absence of NAC (5 mM). **H.** Quantification of invasion in the 3D complex spheroid model measured by HCT116 outgrowth (mCherry) over 6 days ($n = 3$ independent experiments, 2-5 technical replicates included per experiment). **I.** Quantified invasion of HCT116 (mCherry) outgrowth at endpoint (day 6). CM = conditioned medium, NAC = N-acetyl-L-cysteine. The error bars in H show the mean \pm SD and statistically significant differences determined using pairwise 2-way repeated measure ANOVAs (time \times treatment). The interaction term p-values were adjusted using Holm's method. The bar chart and error bars in I show the mean \pm SD. Statistically significant differences were determined using a nested ANOVA followed by Tukey's HSD post-hoc test.

6. Furthermore, to additionally address metastatic spreading *in vivo*, we amended our animal protocol to include a tail vein metastasis assay. In this assay, cancer cells are injected into the tail vein, and metastatic outgrowth is assessed in the lungs.

In our experimental setup, we pre-stimulated cancer cells with conditioned media (CM) collected from CAFs which were treated with either control bacteria (*Ec*), or *Fn* 71. We observed that the pre-stimulation with CM from *Fn*-treated CAFs significantly increased the metastatic outgrowth of injected cancer cells in the lung while the pre-stimulation with the control CM from *Ec*-treated CAFs did not. To draw our conclusion, we analyzed four lung lobes per animal.

We included these data in **Figure 6A-C** and updated the text in lines 244–251 of the Results section and lines 612–627 of the Materials and Methods section.

Figure 6. A. Schematic representation of the *in vivo* experimental setup. SPF NSG mice were tail vein injected with 1×10^6 HT-29 cells, which were pre-treated with CM of CT5.3 CAFs cultured alone or CM of CT5.3 co-cultured with *Fn* 71 or *Ec* (4-hour co-culture, followed by a 24-hour incubation, MOI 50). Lungs were harvested after 30 days ($n = 8$ mice for the Ctrl and *Ec* conditions and $n = 7$ for the *Fn* condition). **B.** Quantification of the relative tumor surface area in percentage per imaged lobe (four lobes were analyzed per mouse). **C.** Representative H&E images of tumors (indicated by black arrows) in the lungs at day 30. CM = conditioned medium. Scale bar = 100 μ m. Statistically significant differences were determined using an ANOVA on a robust linear model fit (using the `lmRob` function in the R package `robust`) followed by Tukey's multiple comparison method (using the R package `multcomp`).

Altogether, we believe that we have successfully implemented more complex and physiologically relevant models to generate study samples. These new results directly address the reviewer's valid concern regarding the *in vivo* relevance of the interaction between *Fn* and CAFs and provide a more detailed characterization of the role this interaction plays in promoting invasion and metastasis.

It is unclear to me why HCT116 was chosen for the tumour migration studies and whether the results are broadly applicable to different sub-types of CRC cell-lines/organoids. Similarly, why was a different CRC cell-line, HT29 chosen for the co-injection studies *in vivo*? Again, is this result reproduced with other cell-lines representing different CRC sub-types and following orthotopic transplantation (to achieve better physiological relevance)?

We thank the reviewer for this comment. We would like to emphasize that our assays involve different bacterial strains and various patient-derived CAF cultures, which already represents a significant challenge. In particular, primary CAF cultures from patients are difficult to maintain across multiple passages. Since the manuscript focuses on CAFs, we have prioritized using CAF cultures obtained from different patients whenever possible. To further minimize bias, we employed three distinct *Fn* strains rather than relying on a single strain.

We also note that three CRC cell lines, HCT116, SW480 and HT-29 were used in the study. HCT116 cells were chosen for *in vitro* assays, while HT-29 cells were selected for *in vivo* experiments, including the new experimental tail vein metastasis assay, as they demonstrate superior tumor engraftment *in vivo*. Moreover, we have now additionally used SW480 CRC cells in the Transwell assay (**Supplementary Figure 4A**). Finally, we were able to validate the invasive behavior of HCT116 cells using the more complex 3D tumor spheroid invasion assay (**Figure 5G-I**).

The figures were generally challenging to interpret, requiring constant reference to text, legends and materials and methods. Inclusion of illustrative cartoons in Figs 5 and 6 greatly enhanced interpretation and I would encourage the same to be done for other figures.

Aiming at enhancing the clarity and improving the overall readability of the manuscript, we carefully reviewed all the figures and added schematics where appropriate.

It was not immediately apparent what the connection is between binding and invasion of Fn in CAFs and the effect of Fn in converting CAFs into iCAF to enhance cell migration.

In figure 3, authors should show that there is no increase of iCAF marker expression when they expose Fap2- Fn to CAFs. Following this idea, in figures 5 and 6 they should show the effect of disrupting this interaction on cancer migration and proliferation.

We thank the reviewer for this comment. We have substantially revised the manuscript to improve clarity on this point. We acknowledge that while we have uncovered some of the interactions between *Fn* and CAFs, as well as their subsequent effects on cancer cells, there are certainly still factors left to be uncovered. Specifically, we were able to identify one mechanism by which *Fn* binds to and invades CAFs. However, since using the Fap2 mutant only partially reduced the binding (**Figure 2D**), we believe that, similarly to what has been shown for CRC cells^{3,4}, *Fn* likely engages via multiple interaction partners.

In the revised manuscript, we now also suggest that an additional interaction—involving LPS and TLR4—participates in the induction of iCAFs and further highlights that *Fn*-CAF interactions, and their resulting phenotypic changes, are multifactorial rather than attributable to a single mechanism (**Supplementary Figure 3B**).

This has been added in lines 186–190 of the Results section.

However, and most importantly, we emphasize further that *Fn* is present in the stromal compartment whereas the control bacteria *Ec* is not. These results are now included in the new **Figure 1F** (see *minor point one*) and further support the observations in our previous and updated binding assays shown in **Figure 1H and 1I**, respectively.

Taking all these considerations into account, we focused on the most downstream effectors in the signaling, namely ROS, and used the ROS scavenger NAC to determine whether it could prevent the cancer cell invasion elicited by *Fn*-stimulated CAFs. We indeed observed a reduced invasion when CAFs were exposed to NAC in addition to *Fn* (**Figure 5G-I**), even though the precise changes in the secretome of *Fn*-exposed CAFs remain to be fully elucidated.

We have substantially revised the Discussion section (*lines 278-292*) to reflect the multifactorial nature of *Fn*-CAF interactions, and we hope that these clarifications will help in explaining to the reviewer the rationale behind our choices to investigate the underlying mechanisms.

Finally, p values should be indicated by its actual number and not stars. A major proportion of the data lot of the data do not appear to be statistically significant, reducing overall confidence in the accuracy of some of the major conclusions. For some experiments, the n number is low and doesn't match or exceed the statistical consensus of a minimum of 3 repetitions.

Most of our experiments are conducted as independent replicates, with each experiment including at least three to four technical replicates. The number of biological replicates is always specified in the figure legends. It is important to note that primary patient-derived CAF cultures are challenging to maintain over multiple passages. Therefore, we often use different CAF cultures to generate biological replicates, which enhances the robustness of our findings by minimizing bias associated with using a single patient-derived culture. Additionally, a biostatistician from our department regularly validates the statistical methods employed in our analyses. We have now indicated the exact p-values down to 0.001 in the figure and indicated smaller p-values using the notation "<0.001".

Minor critique:

In figure 1A, authors should include immunofluorescence with *E. coli* control to show that there is no binding to and invasion of the CAFs.

We thank the reviewer for this insightful comment. It has previously been reported that the *Ec* strain MG1655 is non-invasive¹⁷. Due to the technical requirement for a specific bacterial antibody to label extracellular bacteria, it was not possible to include *Ec* in a directly comparable manner. However, to address the reviewer's comment, we added *Ec* as a control in our attachment/binding assay using *ex vivo* human CRC tissue slides. By fluorescently labeling both *Fn* and *Ec*, we directly compared their binding patterns. Notably, *Fn* exhibited strong binding to CRC tissue, including stromal cell-rich regions, whereas *Ec* showed minimal attachment.

These comparative analyses, including the data involving *Ec*, are now presented in **Figure 1F** of the paper. We have updated the lines 102–107 of the Results section and lines 381–389 of the Materials and Methods section accordingly.

Figure 1. F. Attachment and binding assay of *Fn* 71 and *Ec* (red, indicated by white arrows, DAPI in grey) on human CRC tissue. Epithelium (E) and stromal (S) stromal compartments separated by white dashed line. Images are representative of one patient and two independent experiments. Scale bar = 50 μ m.

I find figure 3A to be particularly challenging to interpret, lacking requisite annotations. Again, a cartoon would help to clarify what has been done and how.

To increase clarity, we now represent Figure 3A (now **Figure 3C** in the revised manuscript) in a different manner and better described it in the text and legend.

Figure 3. C. Log₂ fold change in IL6 and ACTA2 expression in *Fn* treated versus untreated CAF05 cells (two-hour co-culture, MOI 500) (n=3 biologically independent experiments) as well as in patient derived CAFs (CAF4 and CAF32, n=1 for each).

Figure 3B-3D should include a Fap2- *Fn* mutant and/or a GalNAc treated CAF controls.

As shown in **Figure 2D** (previously Figure 3B-D), the binding of *Fn* to CAFs is only partially prevented by the Fap2 mutant, suggesting that, similar to cancer cells, *Fn* interacts with CAFs through several interaction partners. Thus, to get mechanistic insights, we decided to focus on downstream effectors, in particular on ROS, rather than using the Fap2 mutants. Please refer to point four above for further details.

In figure 3B, how do you explain the decrease of PDGFR β expression upon *E. coli* exposure and PDGFR α decrease trend for both *Fn* and *E. coli*?

Indeed, it appears that *Ec* elicits a similar response to *Fn* in a few of the iCAF markers we assessed. However, we did not observe an increase in the ultimate phenotype i.e. binding and CRC cell invasion, suggesting that additional, *Fn*-specific factors are likely to be involved.

We addressed this point in detail in the Discussion section in lines 278–292.

In figure 5A-5C, the authors should show what happens when you treat CAFs with *E. coli*.

The Figure 5A-C shows the RNA-seq data which we generated to elucidate hypothesis on how the secretome of *Fn* treated CAFs may influence gene expression in CRC cells. However, for completeness, we have now included *Ec* as a bacterial control in the downstream invasion assays (transwell migration assay, 3D complex spheroid invasion assay as well as the *in vivo* tail vein assay). These assays allowed us to observe that in contrast to *Fn*, *Ec* wasn't able to lead to the subsequent invasion of tumor cells.

In figure 5D-5F, Fap2- and GalNAc treatments should be added.

In figure 6, FAP2-, GalNAc and *E. coli* treatment should be added. The authors conclude a role for CXCL1 in migration while the results in this figure clearly show an effect on growth. This growth effect has been documented in the literature. When using a blocking antibody against CXCL1, growth is inhibited and IL6 expression is decreased (Miyake M, Furuya H, Onishi S, Hokutan K, Anai S, Chan O, Shi S, Fujimoto K, Goodison S, Cai W, Rosser CJ. Monoclonal Antibody against CXCL1 (HL2401) as a Novel Agent in Suppressing IL6 Expression and Tumoral Growth. *Theranostics*. 2019 Jan 25;9(3):853-867).

We refocused the revised manuscript on CRC invasion rather than on the primary tumor growth and now present ROS as a mechanism for the interaction between *Fn*-activated CAFs and cancer cells. Consequently, we now removed the CXCL1 transwell migration assay and replaced it by results from our newly established 3D spheroid invasion assay where we added NAC treatment. Please refer to point four above for further details. We have updated the graphical abstract accordingly.

In figure 7, the authors do not discuss about the effect of *E. coli* treatment on CXCL1. Does it have an effect on HCT116 migration? What happens if you block CXCL1 activity in this case?

We kindly thank the reviewer for pointing this out. Selecting an appropriate control bacterium in the cancer microbiome field remains a significant challenge. As *Fn* is a Gram-negative bacterium, we decided to use a gram-negative control bacterium and opted for a non-pathogenic strain of *Ec*, consistent with previous studies¹⁴⁻¹⁶. However, as LPS is produced by most gram-negative bacteria including any *Ec* strain, it will contribute to the induced secretion of chemokines such as CXCL1 (already shown in the previous version of the manuscript), or IL-6 and IL-8 (which we show in **Figure 4C-D** and in its extended form including the LPS inhibitor TAK-242 in the **Supplementary figure 3B**).

Additionally, as *Ec* has a faster doubling time than *Fn* (now reflected in lines 190-192), it might further explain why we detected several cytokines in *Ec*-exposed CAFs.

We have included these additional data in **Figure 4C-D** and **Supplementary figure 3B**. The text in lines 183–185 of the Results section and in lines 521–526 of the Materials and Methods section has been updated accordingly.

Nevertheless, as detailed in the revised manuscript, particularly in the new results presented in **Figure 1** and the discussion, we believe that we have now demonstrated that *Fn* specifically binds to the stromal compartment, and to CAFs, a property not observed with *Ec*. Furthermore, in contrast to *Fn*-stimulated CAFs, *Ec* exposed CAFs, do not enhance tumor cell invasion in a 3D spheroid assay and in an *in vivo* metastasis dissemination assay. Using our newly established 3D spheroid assay, we showed that NAC treatment during bacterial co-culture of *Fn* with CAFs significantly reduced the pro-invasive effect of the conditioned media on cancer cells, indicating that ROS production in CAFs is a key driver of *Fn*-induced cancer cell invasion.

This data is presented in **Figure 5G-I**. We have updated the text in lines 230–243 of the Results section and in lines 543–562 of the Materials and Methods section accordingly.

Thus, as we refocused our study on the role of ROS, we prioritized the 3D invasion assay in the presence of the antioxidant NAC and removed the CXCL1 invasion data.

Supplementary figure 3. B. ELISA measured levels of IL-6 and IL-8 in CM 24 hours after a four-hour co-culture of CT5.3, or the patient derived CAF180 and CAF181 CAFs with *Fn* 71 and *Ec* (MOI 50) and pre-treated or not with TAK-242 (a TLR4 inhibitor), (n = 3 independent experiments with 3 independent cell lines).

Figure 5. G. Schematic representation of the *in vitro* complex 3D co-culture experimental setup. mCherry labelled HCT116 CRC cells are co-embedded in collagen with GFP labelled CAFs (CT5.3). These multi-component spheroids are then treated with CM from CT5.3 CAFs cultured alone (blank medium control) or CM of CT5.3 co-cultured with *Fn* 71 or *Ec* (four-hour co-culture, MOI 50), in the presence or absence of NAC (5 mM). **H.** Quantification of invasion in the 3D complex spheroid model measured by HCT116 outgrowth (mCherry) over 6 days (n = 3 independent experiments, 2-5 technical replicates included per experiment). **I.** Quantified invasion of HCT116 (mCherry) outgrowth at endpoint (day 6).

CM = conditioned medium, NAC = N-acetyl-L-cysteine. The error bars in H show the mean \pm SD and statistically significant differences determined using pairwise 2-way repeated measure ANOVAs (time \times treatment). The interaction term p-values were adjusted using Holm's method. The bar chart and error bars in I show the mean \pm SD. Statistically significant differences were determined using a nested ANOVA followed by Tukey's HSD post-hoc test.

Following my previous comment on statistics, I highly doubt the statistical significance of these data, especially figure 7D. Again, please add the actual value of the p value instead of stars.

For all our figures, we conducted several biological replicates (always indicated in the Figure legends), each with a minimum of 3-4 technical replicates, and performed a nested analysis to align the biological experiments. The higher number of independent replicates may explain some of the observed significances. We have now indicated the exact p-values in all the figures. For p-values less than 0.001, we used the notation "<0.001", while all other p-values are presented as exact values.

Noteworthy, due to the revised focus of the manuscript, the specific data shown in Figure 7D is no longer included.

Figure 8: The authors shouldn't ignore the proliferative effect they observe in figure 6. Fn treatment has clearly an effect on proliferation and migration.

We agree with the reviewer. However, due to the revised manuscript's new focus, this figure has been removed and replaced by new results obtained in a tail vein metastasis assay, which we believe better reflects and the function of Fn-exposed CAFs on tumor cells and thus the claims in the manuscript.

References

1. Robinson, W. *et al.* Identification of intracellular bacteria from multiple single-cell RNA-seq platforms using CSI-Microbes. *Science Advances* **10**, eadj7402 (2024).
2. Pelka, K. *et al.* Spatially organized multicellular immune hubs in human colorectal cancer. *Cell* **184**, 4734-4752.e20 (2021).
3. Abed, J. *et al.* Fap2 Mediates *Fusobacterium nucleatum* Colorectal Adenocarcinoma Enrichment by Binding to Tumor-Expressed Gal-GalNAc. *Cell Host & Microbe* **20**, 215–225 (2016).
4. Rubinstein, M. R. *et al.* *Fusobacterium nucleatum* Promotes Colorectal Carcinogenesis by Modulating E-Cadherin/ β -Catenin Signaling via its FadA Adhesin. *Cell Host & Microbe* **14**, 195–206 (2013).
5. Nurmik, M., Ullmann, P., Rodriguez, F., Haan, S. & Letellier, E. In search of definitions: Cancer-associated fibroblasts and their markers. *Intl Journal of Cancer* **146**, 895–905 (2020).
6. Chen, B. *et al.* The molecular classification of cancer-associated fibroblasts on a pan-cancer single-cell transcriptional atlas. *Clinical and Translational Medicine* **13**, e1516 (2023).
7. Elyada, E. *et al.* Cross-Species Single-Cell Analysis of Pancreatic Ductal Adenocarcinoma Reveals Antigen-Presenting Cancer-Associated Fibroblasts. *Cancer Discovery* **9**, 1102–1123 (2019).
8. Koncina, E. *et al.* IL1R1⁺ cancer-associated fibroblasts drive tumor development and immunosuppression in colorectal cancer. *Nat Commun* **14**, 4251 (2023).

9. Bonneaud, T. L. *et al.* Targeting of MCL-1 in breast cancer-associated fibroblasts reverses their myofibroblastic phenotype and pro-invasive properties. *Cell Death Dis* **13**, 787 (2022).
10. Kostic, A. D. *et al.* *Fusobacterium nucleatum* Potentiates Intestinal Tumorigenesis and Modulates the Tumor-Immune Microenvironment. *Cell Host & Microbe* **14**, 207–215 (2013).
11. Li, X. *et al.* A CCL2/ROS autoregulation loop is critical for cancer-associated fibroblasts-enhanced tumor growth of oral squamous cell carcinoma. *Carcinogenesis* **35**, 1362–1370 (2014).
12. Jain, M. *et al.* Mitochondrial reactive oxygen species regulate transforming growth factor- β signaling. *J Biol Chem* **288**, 770–777 (2013).
13. Salvucci, M. *et al.* Patients with mesenchymal tumours and high *Fusobacteriales* prevalence have worse prognosis in colorectal cancer (CRC). Preprint at <https://doi.org/10.1101/2021.05.17.444326> (2021).
14. Sugimura, N. *et al.* *Lactobacillus gallinarum* modulates the gut microbiota and produces anti-cancer metabolites to protect against colorectal tumourigenesis. *Gut* **71**, 2011–2021 (2021).
15. Huang, P. *et al.* *Peptostreptococcus stomatis* promotes colonic tumorigenesis and receptor tyrosine kinase inhibitor resistance by activating ERBB2-MAPK. *Cell Host Microbe* **32**, 1365-1379.e10 (2024).
16. Yu, J. *et al.* An interplay between human genetics and intratumoral microbiota in the progression of colorectal cancer. *Cell Host Microbe* S1931-3128(25)00136–2 (2025) doi:10.1016/j.chom.2025.04.003.

17. de Sousa Figueiredo, M. B. *et al.* Adherent-Invasive and Non-Invasive *Escherichia coli* Isolates Differ in Their Effects on *Caenorhabditis elegans*' Lifespan. *Microorganisms* **9**, 1823 (2021).

Dear Dr Letellier,

Thank you for submitting your revised manuscript (EMBOJ-2024-118410R) to The EMBO Journal, as well for your patience with our feedback. Your amended study was sent back to the referees for their scientific reassessment, and we have received re-reports from three of them, which I enclose below. Please note that while referee #3 was not able at this time to reassess your work, we have evaluated your response to the issues raised by this expert editorially and found them to be addressed satisfactorily. As you will see, the other referees state that the work has been substantially enhanced by the revisions and they are now broadly in favour of publication, pending minor amendments.

Thus, we are pleased to inform you that your manuscript has been accepted in principle for publication in The EMBO Journal.

Please carefully consider the remaining minor points raised by referees #1 and #4 by adjusting the discussion of the findings and introducing additional data or relativising claims in the manuscript where appropriate.

Also, we now need you to take care of a number of issues related to formatting and data presentation as detailed below, which should be addressed at re-submission.

Please contact me at any time if you have additional questions related to below points.

Thank you for giving us the chance to consider your manuscript for The EMBO Journal. I look forward to your final revision.

Again, please contact me at any time if you need any help or have further questions.

Best regards,

Daniel Klimmeck

>> Authors: please revisit name discrepancies for F. P. and E.W. in the manuscript vs. in our online system.

>> Please add up to five keywords to your study.

>> Author Contributions: Remove the author contributions information from the manuscript text. Note that CRediT has replaced the traditional author contributions section as of now because it offers a systematic machine-readable author contributions format that allows for more effective research assessment. and use the free text boxes beneath each contributing author's name to add specific details on the author's contribution.

More information is available in our guide to authors.
<https://www.embopress.org/page/journal/14602075/authorguide>

>> Please provide a 'Disclosure and Competing Interests Statement' section after the Acknowledgements.

>> The manuscript sections should be in the following order: Title page - Abstract & Keywords - Introduction - Results - Discussion - Methods - Data Availability - Acknowledgments - Disclosure Statement & Competing Interests - References - Figure Legends - (Main Tables with legends if applicable) - Expanded View Figure Legends.

>> References: adjust reference format to EMBO Journal format, 10 authors et al, and place References after the Discussion, before figure legends.

>> Please cite your earlier study (Koncina et al, 2024; PMID: 37460545) in the Methods section.

>> Figures in separate files: Figures should be removed from the manuscript text and uploaded as individual, high resolution figure files. Legends should be placed after the References.

>> Funding: Please remove funders and grants pasted in the Comments box and enter them as separate funder via the 'More Funders' option.

>> Appendix file with ToC: there is a PDF with Supplementary tables that should be turned into an Appendix by renaming the tables to Appendix Table S1-S4; some of the suppl. figures could be included here as Appendix Figure S1, etc. . The Appendix file needs to have a ToC with page numbers on its first page.

>> Supplemental Figure Legends: if the suppl. figures will be EV figures then the nomenclature of the individual Figure files and the figure legends in the manuscript needs to be Figure EV1-EV4

>> Add a Reagents and Tools table to the Methods section, as a separate file using the existing template in the Guide For Authors, listing key reagents, experimental models, software and relevant equipment.

>> Data availability section: please update the ID for the first EGAS dataset. Adjust the Author Checklist accordingly.

>> Consider additional changes and comments from our production team as indicated below:

DATA CHECK: FAIL

- Please note that the specific URLs for EGA dataset EGAS00001007205 is not provided in the data availability statement.
- Please note that the accession ID for the "EGA(EGAS0000XXXXXXX)" is not provided in the data availability statement.

Figure Legends - Comments

- Please note that the exact p values are not provided in the legends of figures 1H, I; 2D, E; 3D, H; 4B, C, D, G; 5E, I; S2 C, E
- Please indicate the statistical test used for data analysis in the legends of figures 5B, C
- Please note that information related to n is missing in the legends of figures 5B, I

Referee #1:

The process of demonstration in the article is logically coherent and easily readable. However, I have concerns regarding the validation of the observed results and their interpretation. There are still some questions and concerns about this article.

1. The author should add a description in the introduction section of the link between Fn's known roles in CRC (e.g., immune evasion) and the need to study its interaction with CAFs. Currently, the transition from Fn's pathogenicity to CAF research is abrupt.
2. The author should cite recent literature, update references to include 2023-2025 studies on CAF heterogeneity (e.g., recent single-cell RNA-seq analyses) to enhance timeliness.
3. Regarding the experiments on the combination of bacteria and CAFs, Fap2 and FadA mutants were used, but was it verified whether the other phenotypes of these mutants were normal? For instance, does mutation affect the survival of bacteria or other virulence factors?
4. Increased mitochondrial ROS and TCA cycle activity are observed, but were mitochondrial oxygen consumption rate (OCR) or ATP production measured to directly assess metabolic reprogramming in Fn-exposed CAFs?
5. The study uses Fn 71 and Fn 25586, which belong to different clades. Do these strains differ in Fap2/FadA expression or CAF invasion efficiency? A comparison of virulence factor expression would strengthen conclusions.
6. TAK-242 partially reduces cytokine secretion, but does TLR4 knockout in primary CAFs abolish the iCAF phenotype? This would confirm TLR4 as the primary signaling mediator.
7. The study uses PDGFR α and Lamin A/C as iCAF markers, but recent studies (e.g., Elyada et al., 2019) recommend adding CXCL12 or IL1R1 for specificity. Include additional markers in flow cytometry analyses.
8. The study focuses on Fn but does not address other CRC-associated bacteria (e.g., *Bacteroides fragilis*). The author should discuss how microbial community dynamics might influence CAF phenotypes.
9. Beyond marker expression, validate iCAF functionality via wound healing or ECM remodeling assays to confirm their pro-tumorigenic role.

Referee #2:

Revisions look strong and this is a very detailed and important manuscript. I would accept it now

Referee #4:

The authors have extensively modified figure 1, adding in vivo data with both human and mouse tissue. The new data presented are convincing and strongly reinforce their initial findings. Minor remaining critiques are:

- 1) In figure 1C, the contrast between each marker is very low making it hard to distinguish clearly that Fn is in close proximity to Pdpn and α SMA fibroblasts.
- 2) Authors have analysed CRC tumours, but they give very little details about what kind of tumour they are. Colorectal cancer is a very heterogenous disease. The authors should provide more details for the CRC samples (localisation, stage, ...).
- 3) In figure 1D, a control mouse with no tumour should be added.
- 4) The authors have used a Cdx2-Cre, Apcfl/fl model to drive tumorigenesis. This model has been previously published and should be referenced.
- 5) Supplementary figure 1A & B lack labels.

In line 99-100 of the manuscript, it is premature to conclude a role for bacteria in stromal remodelling at this point. Whilst an in vivo presence of the bacteria has been shown, its functional relevance remains to be demonstrated.

Again, the authors should give more details about the CRC tissue they obtained (at least in the methods section).

Finally, the authors have included a tail vein metastasis assay (Figure 6). While the Fn-treated CAF conditioned medium clearly influences metastasis formation, the author should accurately quantify the number of metastases in the lung. They show an effect on the expansion of these mets but it could be due to a higher proliferation rate, an effect they showed in the initial manuscript. A staining showing the proliferation status of these different mets (Ki67, PCNA) should be included and the authors should at least discuss the Fn influence on proliferation.

The figures have been greatly improved.

The authors have added a significant amount of new data and discuss more extensively their strategy to focus on FAP2 and potential other interactions. The data are in general satisfactory.

The focus on the ROS is well executed but raise the question again on a potential effect on the proliferation. Indeed, ROS and inflammation are known to enhance proliferation. The author should discuss this potential effect..

All other critique have been addressed.

Overall, the inclusion of the new data strongly enhances the focus of this article. While the new focus of the article is understandable (migration and invasion effect), the initial finding about the effect on proliferation should be discussed by the authors. This is particularly important because ROS and inflammation have a known effect on proliferation.

We would like to thank the reviewers and the editor for their valuable comments. Below, you will find a point-by-point response to all the comments. All changes are highlighted in green in the revised manuscript.

Referee #1:

The process of demonstration in the article is logically coherent and easily readable. However, I have concerns regarding the validation of the observed results and their interpretation. There are still some questions and concerns about this article.

1. The author should add a description in the introduction section of the link between *Fn*'s known roles in CRC (e.g., immune evasion) and the need to study its interaction with CAFs. Currently, the transition from *Fn*'s pathogenicity to CAF research is abrupt.

We thank the reviewer for this suggestion. We have now revised the introduction accordingly and added the following text:

“While interactions between *Fn* and epithelial or immune cells have been explored, elucidating its engagement with the stromal compartment, particularly CAFs, is critical to fully understanding *Fn*'s impact on the TME and its potential as a therapeutic target.”

2. The author should cite recent literature, update references to include 2023-2025 studies on CAF heterogeneity (e.g., recent single-cell RNA-seq analyses) to enhance timeliness.

We thank the reviewer for this suggestion. We have added additional recent references regarding CAF heterogeneity to the introduction.

3. Regarding the experiments on the combination of bacteria and CAFs, *Fap2* and *FadA* mutants were used, but was it verified whether the other phenotypes of these mutants were normal? For instance, does mutation affect the survival of bacteria or other virulence factors?

We thank the reviewer for this important comment. Both the viability and morphology of the mutants were assessed, and no obvious differences were observed compared to the control *Fn*. Below (*for reviewing purposes only*), we present endpoint OD measurements from three independent cultures of *Fn* 23726 (wild-type control, as it is the background for the mutants), along with the *Fap2* and *FadA* mutants. No differences in overall growth were observed.

Additionally, as we hope is now clearly conveyed in the revised manuscript, attachment and binding of *Fn* via Gal-GalNAc is only one of several factors regulating the *Fn*-induced CAF phenotype.

Figure for reviewing purposes. Optical density (600 nm) of three independent cultures of *Fn* 23726 wild type control, along with the *Fap2* and *FadA* mutants.

4. Increased mitochondrial ROS and TCA cycle activity are observed, but were mitochondrial oxygen consumption rate (OCR) or ATP production measured to directly assess metabolic reprogramming in *Fn*-exposed CAFs?

We thank the reviewer for this comment. While we have observed an increased relative utilization of glutamine in the TCA cycle in *Fn*-exposed CAFs, it is true that we have not used OCR to measure absolute levels of oxidative phosphorylation (OXPHOS). However, we believe that the central finding of our study is the elevated mitochondrial ROS in *Fn*-exposed CAFs, which we demonstrate as a key driver of the cancer cell invasion phenotype. As such, whether OCR is increased in *Fn*-exposed CAFs does not impact the main conclusions of our manuscript.

5. The study uses *Fn* 71 and *Fn* 25586, which belong to different clades. Do these strains differ in *Fap2*/*FadA* expression or CAF invasion efficiency? A comparison of virulence factor expression would strengthen conclusions.

We thank the reviewer for this comment and agree that this consideration is important when studying different *Fusobacterium nucleatum* clades and subspecies. In an attempt to address this point:

- We compared the binding capacity of both species and presented the findings in Figure 1, with the following description in the text:

“Moreover, we observed *Fn* subsp. *animalis* (*Fn* 71), isolated from the human gastrointestinal tract (Strauss *et al*, 2011) and a representative of the *Fna* C2 (Zepeda-Rivera *et al*, 2024) reported to have pro-tumorigenic functions in CRC (Kostic *et al*, 2013), to exhibit the highest affinity (Figure 1I and Appendix Figure S1C).”

- We also addressed this point in the discussion to emphasize the need for further investigations in the following sentence:

“Further studies are required to elucidate the mechanisms underlying the strain-specific variability in *Fn* binding to CAFs and the resulting downstream effects on tumor cells.”

- Additionally, we now refer to two recent studies exploring different virulence genes across *Fusobacterium* species and revised the discussion accordingly:

“Additionally, we observed differences in CAF binding affinity among *Fusobacterium nucleatum* subspecies and strains, which may be attributed to variation in the expression of virulence factors, particularly *Fap2*, as recently reported (Ma *et al*, 2023; Ponath *et al*, 2021).”

We acknowledge the importance of species- or strain-specific pathogenicity as well as the existing knowledge gap in this domain. However, while this represents a promising avenue for future investigations, we believe that a comprehensive analysis of this aspect falls outside the scope of the present study.

6. TAK-242 partially reduces cytokine secretion, but does TLR4 knockout in primary CAFs abolish the iCAF phenotype? This would confirm TLR4 as the primary signaling mediator.

As already mentioned in the revised version of the manuscript, we do not believe that the observed phenotype—namely, the invasion and metastatic dissemination of cancer cells—is linked solely to TLR signaling, as this phenotype is not observed in *Ec*-treated CAFs. For this reason, while a TLR4 knockout in CAFs might reveal a link to the iCAF signature, it would not explain the functional phenotype, which we show to be associated with ROS. Therefore, we consider that this approach is unlikely to provide further insight into the underlying mechanism.

7. The study uses PDGFR α and Lamin A/C as iCAF markers, but recent studies (e.g., Elyada *et al*, 2019) recommend adding CXCL12 or IL1R1 for specificity. Include additional markers in flow cytometry analyses.

The Elyada *et al* (2019) paper defines the following iCAF markers: IL6, PDGFRA, CFD, PLA2G2A, HAS1, CXCL2, CCL2, CLU, EMP1, LMNA. We decided to use three markers out

of this list, namely IL6, PDGFRA and LMNA. Additionally, we have assessed further well-established and widely accepted CRC iCAF markers. However, we believe that the most accurate and reliable way to define the iCAF phenotype is through a comprehensive gene expression signature, which is the approach we have taken in the revised manuscript (please refer to Figure 3E and F).

8. The study focuses on *Fn* but does not address other CRC-associated bacteria (e.g., *Bacteroides fragilis*). The author should discuss how microbial community dynamics might influence CAF phenotypes.

We thank the reviewer for this comment and have now revised the discussion accordingly.

“Finally, as *Fn* exists within a broader microbial community in the TME, future studies should explore how microbial interactions influence CAF phenotypes and collectively shape the stromal compartment.”

9. Beyond marker expression, validate iCAF functionality via wound healing or ECM remodeling assays to confirm their pro-tumorigenic role.

In the revised version of the manuscript, we not only used additional iCAF markers in both, flow cytometry and ELISA but also propose that ROS, which is an established hallmark of iCAFs, is a key mechanism linking *Fn*-exposed CAFs to cancer cell invasion. While we agree that this may not be the only mechanism involved, we believe that follow-up studies will provide further insights into how *Fn* reprograms CAFs. We have revised the discussion accordingly:

“While ROS appears to be a key driver, it is unlikely to be the only mechanism involved. Future work should investigate additional pathways by which *Fn*-exposed CAFs influence tumor cell invasion.”

Referee #4:

The authors have extensively modified figure 1, adding in vivo data with both human and mouse tissue. The new data presented are convincing and strongly reinforce their initial findings. Minor remaining critiques are:

1) In figure 1C, the contrast between each marker is very low making it hard to distinguish clearly that Fn is in close proximity to Pdpn and α SMA fibroblasts.

To improve clarity and distinguishability between markers, we have updated the image in Figure 1C using a gamma correction of 0.5 and noted this information in the corresponding figure legend.

Amended to the figure legend:

The image in C was visualized using a gamma correction of 0.5 to enhance visibility.

2) Authors have analysed CRC tumours, but they give very little details about what kind of tumour they are. Colorectal cancer is a very heterogenous disease. The authors should provide more details for the CRC samples (localisation, stage, ...).

We have now added clinical data (stage of the tumor and microsatellite stability) in the respective figure legend.

B-C. *In situ* hybridization staining of human CRC tissue with α SMA in brown (**B, C**), PDPN in purple (**C**), and *Fn* in pink (**B, C**; indicated by white arrows) in two independent stage III, microsatellite stable CRC patients. Scale bar = 50 μ m.

3) In figure 1D, a control mouse with no tumour should be added.

We believe the reviewer is referring to a control mouse in the experiment, specifically a CDX2-CreER^{T2}Apc^{fl/fl} mouse orally gavaged with PBS. We have now included this control in Appendix Figure S1C.

Newly added figure legend for Appendix Figure S1C: **C.** Representative immunofluorescent image showing no colocalization of *Fn* (red, stained with an OMP *Fn* specific antibody) with the stromal marker PDPN (green), and DAPI (grey) in a PBS-gavaged CDX2-CreER^{T2}Apc^{fl/fl} mouse (as a control for the main Figure 1E). Scale bar = 50 μ m.

4) The authors have used a Cdx2-Cre, Apcfl/fl model to drive tumorigenesis. This model has been previously published and should be referenced.

Thanks for pointing it out. The reference has been added in the materials and methods section of the manuscript.

5) Supplementary figure 1A & B lack labels.

We updated the Appendix Figure S1A to provide the same image field as shown in Figure 1B and corrected the figure legend to refer to the appropriate Figure 1B as well as describing that the black outlined region is intended to highlight the same region of interest as outlined in Figure 1B.

In line 99-100 of the manuscript, it is premature to conclude a role for bacteria in stromal remodelling at this point. Whilst an in vivo presence of the bacteria has been shown, its functional relevance remains to be demonstrated.

We thank the referee for this comment, we have edited the text accordingly:

“Thereby, *Fn*-exposed CAFs contribute to driving cancer cell migration and invasion. Altogether, this study **suggests** a multifaceted role for *Fn* in modulating the TME, ultimately enhancing cancer cell invasiveness and revealing a previously unrecognized function of *Fn*, particularly through its interactions with CAFs.”

Again, the authors should give more details about the CRC tissue they obtained (at least in the methods section).

We have now added clinical data (stage of the tumor and microsatellite stability) in the respective Figure legend:

B-C. *In situ* hybridization staining of human CRC tissue with α SMA in brown (**B, C**), PDPN in purple (**C**), and *Fn* in pink (**B, C**; indicated by white arrows) in two independent stage III, microsatellite stable CRC patients. Scale bar = 50 μ m.

Finally, the authors have included a tail vein metastasis assay (Figure 6). While the Fn-treated CAF conditioned medium clearly influences metastasis formation, the author should accurately quantify the number of metastases in the lung. They show an effect on the expansion of these mets but it could be due to a higher proliferation rate, an effect they showed in the initial manuscript. A staining showing the proliferation status of these different mets (Ki67, PCNA) should be included and the authors should at least discuss the Fn influence on proliferation.

The figures have been greatly improved.

The authors have added a significant amount of new data and discuss more extensively their strategy to focus on FAP2 and potential other interactions. The data are in general satisfactory.

The focus on the ROS is well executed but raise the question again on a potential effect on the proliferation. Indeed, ROS and inflammation are known to enhance proliferation. The author should discuss this potential effect.

All other critique have been addressed.

Overall, the inclusion of the new data strongly enhances the focus of this article. While the new focus of the article is understandable (migration and invasion effect), the initial finding about the effect on proliferation should be discussed by the authors. This is particularly important because ROS and inflammation have a known effect on proliferation.

We thank the reviewer for the positive feedback on the improvements made during the revision process, and we also appreciate the insightful comment regarding the underlying mechanism. In response to the reviewer's suggestion, we performed Ki67 stainings on lung metastases from mice intravenously injected with cancer cells pre-stimulated with control, Ec-, or Fn-treated CAFs (figure provided for reviewing purposes only). We were

not able to observe a difference in Ki67 staining in the metastases between the different conditions. However, we agree with the reviewer that elevated ROS levels may contribute to both proliferation and invasion. We have now acknowledged this potential limitation in the revised manuscript:

“Further studies are needed to determine whether the observed metastatic phenotype is driven exclusively by ROS-induced invasion, or whether enhanced proliferation also contributes to metastatic outgrowth.”

Figure for reviewing purposes only: Ki67 staining on lung tissues from mice treated intravenously with cancer cells pre-treated with control, *E.coli* or Fn exposed CAFs. One representative section is shown from one mouse out of three mice per group. Scale bar = 500 μ m.

References

- Elyada E, Bolisetty M, Laise P, Flynn WF, Courtois ET, Burkhart RA, Teinor JA, Belleau P, Biffi G, Lucito MS, *et al* (2019) Cross-Species Single-Cell Analysis of Pancreatic Ductal Adenocarcinoma Reveals Antigen-Presenting Cancer-Associated Fibroblasts. *Cancer Discovery* 9: 1102–1123
- Kostic AD, Chun E, Robertson L, Glickman JN, Gallini CA, Michaud M, Clancy TE, Chung DC, Lochhead P, Hold GL, *et al* (2013) *Fusobacterium nucleatum* Potentiates Intestinal Tumorigenesis and Modulates the Tumor-Immune Microenvironment. *Cell Host & Microbe* 14: 207–215
- Ma X, Sun T, Zhou J, Zhi M, Shen S, Wang Y, Gu X, Li Z, Gao H, Wang P, *et al* (2023) Pangenomic Study of *Fusobacterium nucleatum* Reveals the Distribution of Pathogenic Genes and Functional Clusters at the Subspecies and Strain Levels. *Microbiology Spectrum* 11: e05184-22
- Ponath F, Tawk C, Zhu Y, Barquist L, Faber F & Vogel J (2021) RNA landscape of the emerging cancer-associated microbe *Fusobacterium nucleatum*. *Nat Microbiol* 6: 1007–1020
- Strauss J, Kaplan GG, Beck PL, Rioux K, Panaccione R, DeVinney R, Lynch T & Allen-Vercoe E (2011) Invasive potential of gut mucosa-derived *Fusobacterium nucleatum* positively correlates with IBD status of the host. *Inflammatory Bowel Diseases* 17: 1971–1978
- Zepeda-Rivera M, Minot SS, Bouzek H, Wu H, Blanco-Míguez A, Manghi P, Jones DS, LaCourse KD, Wu Y, McMahon EF, *et al* (2024) A distinct *Fusobacterium nucleatum* clade dominates the colorectal cancer niche. *Nature* 628: 424–432

Dear Dr Letellier,

Thank you for submitting the revised version of your manuscript. I have now evaluated your amended manuscript and concluded that the remaining minor concerns have been sufficiently addressed.

I am thus pleased to inform you that your manuscript has been accepted for publication in the EMBO Journal.

On a different note, I would like to alert you that EMBO Press offers a format for a video-synopsis of work published with us, which essentially is a short, author-generated film explaining the core findings in hand drawings, and, as we believe, can be very useful to increase visibility of the work. Please see the following link for representative examples and their integration into the article web page:

<https://www.embopress.org/doi/full/10.15252/emj.2019103932>

Best regards,

Daniel Klimmeck

Daniel Klimmeck, PhD
Senior Editor
The EMBO Journal
EMBO
Postfach 1022-40
Meyerhofstrasse 1
D-69117 Heidelberg
contact@embojournal.org